



# Quantification of primary and secondary organic aerosol sources by combined factor analysis of extractive electrospray ionisation and aerosol mass spectrometer measurements (EESI-TOF and AMS)

Yandong Tong[1], Lu Qi[1], Giulia Stefenelli[1], Dongyu S. Wang[1], Francesco Canonaco[1], Urs Baltensperger[1], André S.H. Prévôt[1], Jay G. Slowik[1],*

[1]Laboratory of Atmospheric Chemistry, Paul Scherrer Institute (PSI), 5232 Villigen, Switzerland

*Correspondence to Jay Slowik (jay.slowik@psi.ch)

**Abstract:**

**Source apportionment studies have struggled to quantitatively link secondary organic aerosol (SOA) to its precursor sources, due largely to instrument limitations. For example, aerosol mass spectrometers (AMS) provide quantitative measurements of the total SOA fraction, but lack the chemical resolution to resolve most SOA sources. In contrast, instruments based on soft ionisation techniques, such as extractive electrospray ionisation mass spectrometry (EESI, e.g., the EESI time of flight mass spectrometer, EESI-TOF), have demonstrated the resolution to identify specific SOA sources but provide only a semi-quantitative apportionment due to uncertainties in the dependence of instrument sensitivity on molecular identity. We address this challenge by presenting a method for positive matrix factorisation (PMF) analysis on a single dataset which includes measurements from both AMS and EESI-TOF instruments, denoted "combined PMF" (cPMF). Because each factor profile includes both AMS and EESI-TOF components, the cPMF analysis maintains the source resolution capability of the EESI-TOF, while also providing quantitative factor mass concentrations. Therefore, the bulk EESI-TOF sensitivity to each factor can also be directly determined from the analysis. We present metrics for ensuring both instruments are well-represented in the solution, a method for optionally constraining factor profile contributions for one or both instruments, and a protocol for uncertainty analysis.**

**As a proof of concept, the cPMF analysis was applied to summer and winter measurements in Zurich, Switzerland. Factors related to biogenic and wood burning-derived SOA are quantified, as well as POA sources such as wood burning, cigarette smoke, cooking, and traffic. The retrieved EESI-TOF factor-dependent sensitivities are consistent with both laboratory measurements of SOA from model precursors and bulk sensitivity parameterisations based on ion chemical formulae. The cPMF analysis shows that with the standalone EESI-TOF PMF, in which factor-dependent sensitivities are not accounted for, some factors are significantly under/overestimated. For example, when factor-dependent sensitivities are not considered in the winter dataset, the SOA fraction is underestimated by ~25% due to the high EESI-TOF sensitivity to components of primary biomass burning such as levoglucosan. In the summer dataset, where both SOA and total OA are dominated by monoterpene oxidation products, the uncorrected EESI-TOF underestimates the fraction of daytime SOA relative to nighttime SOA (in which organonitrates and less oxygenated $C_xH_yO_z$ molecules are enhanced). Although applied here to an AMS/EESI-TOF pairing, cPMF is suitable for the general case of a multi-instrument dataset, thereby**



**providing a framework for exploiting semi-quantitative, high-resolution instrumentation for**
**quantitative source apportionment.**
## 1. Introduction
Atmospheric aerosols negatively affect visibility (Chow et al., 2002), human health (Beelen et al., 2014;
Laden et al., 2006; Pope et al., 2002), and urban air quality (Fenger, 1999; Mayer, 1999) on local and
regional scales. Aerosols also provide the largest uncertainties for global radiation balance and climate
change (Myhre et al., 2014; Penner et al., 2011; Forster et al., 2007; Lohmann and Feichter, 2005).
Therefore, to develop appropriate mitigation policies, it is of vital importance to understand aerosol
chemical composition, sources, and evolution. Organic aerosol (OA) is a major component of
atmospheric aerosol and accounts for 20 to 90 % of the submicron aerosol mass (Jimenez et al., 2009).
OA is typically classified as either primary organic aerosol (POA), which is directly emitted to the
atmosphere, or secondary organic aerosol (SOA), which is produced by atmospheric reactions of
emitted volatile organic compounds (VOCs). Both POA and SOA can exert serious health effects,
including protein and DNA damage caused by reactive oxygen species (ROS), which can be either
contained in the particles or induced by oxidation reactions following inhalation (Fuller et al., 2014;
Kelly and Fussell, 2012; Reuter et al., 2010; Li et al., 2003; Halliwell and Cross, 1994). Recent studies
indicate that the oxidation potential of SOA is source-dependent. Therefore, different sources likely
carry different health risks, highlighting the importance of OA source identification and quantification
(Daellenbach et al., 2020; Zhou et al., 2018). Previous studies have been relatively successful in
quantitatively linking POA to its sources. However, quantification of SOA sources and/or formation
pathways is more challenging due to 1) the chemical complexity of SOA, which can consist of
thousands of unique oxidation products, including highly oxygenated molecules and high molecular
weight organic oligomers, and 2) limitations of traditional instrumentation for characterising OA
chemical composition, especially the SOA fraction. Therefore, the effects of individual SOA sources
on health and climate remain poorly constrained.
Positive matrix factorisation (PMF) is a widely used source apportionment technique. PMF is a bilinear
receptor model which represents the measured mass spectral time series as a linear combination of
factor mass spectra and their corresponding time-dependent concentrations (Paatero and Tapper, 1994).
These factors may then be related to emission sources, and/or atmospheric processes, depending on
their chemical and temporal characteristics. PMF has been implemented in extensive online and offline
studies worldwide to quantify OA sources. The Aerodyne aerosol mass spectrometer (AMS) is widely
used in OA source apportionment studies because it provides online, quantitative measurements of non-
refractory $PM_1$ or $PM_{2.5}$ (particulate matter with aerodynamic diameter smaller than 1 or 2.5 μm,
respectively) chemical composition with high time resolution. Source apportionment studies using PMF
based on AMS data have successfully separated and quantified POA sources based on different
chemical signatures, e.g., hydrocarbon-like OA (HOA) (Zhao et al., 2019; Xu et al., 2019; Sun et al.,
2016a; Elser et al., 2016; Zhang et al., 2014; Ng et al., 2011b), cooking-related OA (COA) (Xu et al.,
2019; Zhao et al., 2019; Hu et al., 2016; Sun et al., 2016a; Sun et al., 2016b; Crippa et al., 2013a; Mohr
et al., 2012), biomass burning OA (BBOA) (Alfarra et al., 2007; Lanz et al., 2007; Sun et al., 2011),
and coal combustion OA (CCOA) (Elser et al., 2016; Hu et al., 2016; Sun et al., 2016a; Zhang et al.,
2014; Zhang et al., 2008). However, SOA is typically reported as either a single SOA factor (denoted
oxygenated organic aerosol, OOA), or as two factors distinguished by degree of oxygenation (i.e., less
oxygenated OOA, LO-OOA, and more oxygenated OOA, MO-OOA) or by volatility (i.e., semi-volatile
OOA, SV-OOA, and low-volatility OOA, LV-OOA) (Xu et al., 2019; Elser et al., 2016; Sun et al.,
2016a; Sun et al., 2013; Jimenez et al., 2009; Zhang et al., 2011; Crippa et al., 2013a) rather than in





terms of sources and/or formation processes. This limitation is due to the vaporisation/ionisation scheme
in the AMS, which causes significant thermal decomposition and ionisation-induced fragmentation
(Decarlo et al., 2006). The corresponding decrease in chemical resolution, particularly for the
multifunctional and/or highly oxygenated SOA components molecules of which SOA is comprised (e.g.,
multifunctional acids, peroxides, organonitrates, organosulfates, oligomers), limits the resolution of
SOA source apportionment.
The development of continuous or semi-continuous instruments with softer vaporisation/ionisation
schemes has provided new insights into SOA composition, and is thus of considerable interest for source
apportionment. Recent examples include the (semi-continuous) Filter Inlet for Gases and AEROsols
chemical ionisation time-of-flight mass spectrometer (FIGAERO-CIMS) (Lopez-Hilfiker et al., 2014),
and the (continuous) extractive electrospray ionisation time-of-flight mass spectrometer (EESI-TOF)
(Lopez-Hilfiker et al., 2019), which implement soft ionisation schemes at lower temperatures than the
AMS, thereby reducing thermal decomposition and increasing chemical resolution (i.e., providing
chemical formulae of molecular ions). A recent source apportionment study using FIGAERO-CIMS at
a rural site in the southeastern USA successfully resolved three SOA factors, characterised by isoprene-
derived species such as carboxylic acids from aqueous phase processes, highlighting the chemistry of
biogenic species (Chen et al., 2020). Source apportionment studies in Zurich using an EESI-TOF
identified SOA factors from monoterpene oxidation in summer (Stefenelli et al., 2019) and oxidation
of biomass burning emissions in winter (Qi et al., 2019). EESI-TOF measurements identified SOA
factors related to solid fuel combustion and aqueous-phase processes in Beijing (Tong et al., 2021) and
SOA factors with aromatic and biogenic origins in Delhi (Kumar et al., 2021). However, to date the
factor concentrations returned by PMF analyses using these instruments are not quantitative.
Quantification of the measurements by instruments such as EESI-TOF and CIMS is challenging,
because the instrument sensitivity varies strongly with molecular identity. For CIMS, the sensitivity to
different compounds is determined by the frequency of collisions between reagent ions and analytes,
the ion–molecule reaction time, and the transmission efficiency of product ions to the detector, which
depends on ion-molecule binding energy. Lopez-Hilfiker et al. (2016) developed methods to estimate
the binding energy of iodide ($I^-$) adduct ions of multifunctional organic compounds for species whose
formation is collision-limited, providing a lower limit to their mass concentrations. Another method to
explore the sensitivity is to measure single-compound aerosols or SOA generated from different
precursors simultaneously by an EESI-TOF and a scanning mobility particle sizer (SMPS) to determine
the mass concentration (Lopez-Hilfiker et al., 2016). Lopez-Hilfiker et al. (2019) explored EESI-TOF
sensitivities to selected reference compounds with different functional groups (including saccharides,
polyols and carboxylic acids) and bulk SOA generated from oxidation of a single precursor VOC. For
pure compounds, relative sensitivities vary by two orders of magnitude, with some composition-
dependent trends evident (e.g., increasing sensitivity of saccharides with decreasing molecular weight,
and high sensitivities for polyols relative to other functionalities). In addition, a trend of decreasing
sensitivity with decreasing molecular weight of the precursors was found for bulk SOA. While
calibration with standard compounds is straightforward, the quantification of individual species within
SOA is extremely challenging, due to its complex composition, the lack of chemical standards for most
molecules, and the potential for structural isomers to have significantly different sensitivities. These
issues were investigated recently for the EESI-TOF by generating SOA in the presence of a variable
seed surface area, and comparing the difference in SOA ion concentrations measured by the EESI-TOF
and the corresponding gas-phase concentrations measured by a Vocus proton transfer reaction-mass
spectrometer (Vocus-PTR-MS) (Wang et al., 2021). The observed sensitivities for different SOA
components produced from the oxidation of limonene, *o*-cresol, or 1,3,5-trimethylbenzene ranged from





$10^3$ to $10^5$ ion s$^{-1}$ ppb$^{-1}$. A regression model was developed that was able to predict the ion-by-ion
sensitivities to within a factor of 5 of the experimental value when the precursor VOC is known *a priori*.
However, the study also showed significantly different sensitivities (up to a factor of 20) for structural
isomers derived from different VOC precursors. Similar isomer sensitivity differences for I⁻-CIMS was
also reported by (Bi et al., 2021). The fact that these isomers cannot be distinguished by 1-dimensional
mass spectrometry, represents a fundamental limitation of calibration/parameterisation-based
quantification and complicates interpretation of the binding energy-based approach (Lopez-Hilfiker et
al., 2016), because ambient SOA may derive from unknown or complex mixtures of VOCs. Therefore,
for source apportionment purposes, source-based sensitivities are preferred and essential to quantify
SOA sources and formation processes.
Here we present a new approach for quantification of SOA sources retrieved from source apportionment.
This is achieved by PMF analysis of a single input matrix consisting of data from both a quantitative
instrument with lower chemical resolution (i.e., AMS) and an instrument with high chemical resolution
and a linear but molecule-dependent response (i.e., EESI-TOF). This method is based on the combined
PMF (cPMF) analysis previously performed on combined OA/VOC data from AMS and PTR-MS,
respectively (Crippa et al., 2013b; Slowik et al., 2010), but utilises a more robust metric for ensuring
adequate representation of both instruments in the model solution, optionally allows constraints to be
placed on the factor profile contributions for one or both instruments, and provides a method for
uncertainty analysis. The cPMF method is applied to AMS/EESI-TOF datasets collected during summer
and winter campaigns in Zurich, Switzerland, for which single-instrument PMF analyses were
previously reported (Qi et al., 2019; Stefenelli et al., 2019). The present study is the first application of
cPMF to a joint EESI-TOF/AMS dataset, and the first attempt at quantitative EESI-TOF-driven source
apportionment.
**2. Methodologies**

27       **2.1 The measurement site and field campaigns**

Field campaigns were conducted at the Swiss National Air Pollution Monitoring Network (NABEL)
station, an urban background site located in the Alte Kaserne, central Zurich (47º22' N, 8º33' E, 410 m
above sea level), previously described in detail (Canonaco et al., 2013; Lanz et al., 2007). The
measurements used in the current analysis are from 20 June to 26 June 2016 and 25 January to 4
February 2017. These periods are excerpted from longer campaigns, and correspond to the times during
which both the AMS and EESI-TOF achieved stable operation. The measurement site is located in a
courtyard, although influences from nearby restaurants, local minor roads, and human activities (e.g.,
cigarette smoking) are often observed (Lanz et al., 2007; Daellenbach et al., 2017; Stefenelli et al., 2019;
Qi et al., 2019; Qi et al., 2020). Gas-phase species, e.g., nitrogen dioxide (NO$_2$), nitrogen oxide (NO)
and sulfur dioxide (SO$_2$) and meteorological data, e.g., temperature (T), relative humidity (RH),
radiation, wind speed (WD) and wind direction (WD) are recorded by the monitoring station.
During the intensive campaigns, a separate trailer was deployed to house an additional suite of gas and
particle instrumentation. A PM$_{2.5}$ cyclone was installed ~75 cm above the trailer roof (~5 m above
ground) to remove coarse particles. After passing through the cyclone, the sampled air passed through
a stainless steel (~6 mm outer diameter, O.D.) tube to the particle instrumentation, which included a
high-resolution time-of-flight aerosol mass spectrometer (HR-TOF-AMS, Aerodyne Research Inc.) and
an extractive electrospray ionisation time-of-flight mass spectrometer (EESI-TOF) to measure the OA
composition, and a scanning mobility particle sizer (SMPS) to measure the particle concentration and
size distribution. The summer and winter campaign results, including OA source apportionment from
the standalone AMS and EESI-TOF datasets, were previously presented in detail (Qi et al., 2019;





Stefenelli et al., 2019). In this study, we focus on the OA source apportionment using positive matrix factorisation (PMF) on the combined dataset from AMS and EESI-TOF, collected during the two campaigns.

### 2.2 Instrumentation
#### 2.2.1 High-resolution time-of-flight aerosol mass spectrometer (HR-TOF-AMS)

The AMS (Aerodyne Research, Inc.) provides fast, online, quantitative measurements of the size-resolved composition of non-refractory $PM_1$ (NR-$PM_1$). A detailed description of the instrument can be found elsewhere (Decarlo et al., 2006; Canagaratna et al., 2007), while operational details and data treatment are documented in Stefenelli et al. (2019) and Qi et al. (2019). Briefly, in both campaigns, the organic composition of NR-$PM_1$ was measured by AMS with a time resolution of 1 min. At the beginning and at the end of the both campaigns, the instrument was calibrated for ionisation efficiency (IE) using 400 nm $NH_4NO_3$ particles by the mass-based method (Canagaratna et al., 2007; Jimenez et al., 2003). The HR-TOF-AMS data was analysed using the SQUIRREL (v.1.57) and PIKA (v.1.16) software packages in IGOR Pro 6.37 (Wavemetrics, Inc., Portland, OR, USA). Before further single-instrument and cPMF analysis, a composition-dependent collection efficiency (CDCE) was implemented to correct the measured aerosol mass (Middlebrook et al., 2012). For both single-instrument PMF and cPMF analysis, the input matrices consisted of the time series of fitted OA ions from highresolution mass spectral analysis, together with their corresponding uncertainties estimated from ion counting statistics and detector variability according to Allan et al. (2003). Following Ulbrich et al. (2009), a minimum error value was applied to the error matrix.

The AMS PMF input matrices are identical to those used by Stefenelli et al. (2019) and Qi et al. (2019), with the exception that they include not only the OA ions retrieved from spectral analysis, but also $NO^+$ and $NO_2^+$. These ions are added because they represent the major products measured from organonitrate fragmentation (Farmer et al., 2010), and standalone EESI-TOF PMF suggested a significant role for organonitrates and other nitrogen-containing species during both the summer and winter campaigns (Qi et al., 2019; Stefenelli et al., 2019). Detailed descriptions of the final input matrices from AMS (e.g., number of measurements, number of ions and time resolution) in summer and in winter are presented in Table 1.

#### 2.2.2 Extractive electrospray ionisation time-of-flight mass spectrometer (EESI-TOF)

The EESI-TOF provides online, fast, near-molecular-level measurement (i.e., chemical formulae of molecular ions) of OA composition, without thermal decomposition or ionisation-induced fragmentation. A detailed description can be found elsewhere (Lopez-Hilfiker et al., 2019) and the operational details for the summer and winter campaigns are documented in Stefenelli et al. (2019) and Qi et al. (2019), respectively. Briefly, aerosol particles were continuously sampled through a 6 mm O.D., 5 cm long multi-channel extruded carbon denuder. Particles then intersected a spray of charged droplets generated by a conventional electrospray probe and the soluble fraction was extracted into the droplets. The droplets passed through a heated stainless-steel capillary (~250 °C), wherein the electrospray solvent evaporated and ions were ejected into the mass spectrometer. Due to the short residence time (~1 ms) in the capillary, no thermal decomposition was observed. The analyte ions were detected by a high-resolution time-of-flight mass spectrometer with an atmospheric pressure interface (API-TOF) (Junninen et al., 2010). In the summer campaign, the electrospray consisted of a 1:1 water/methanol (MeOH, UHPLC-MS grade, LiChrosolv) mixture doped with 100 ppm NaI (>99 %, Sigma-Aldrich). In the winter campaign, a 1:1 water/acetonitrile mixture (> 99.9 %, Sigma-Aldrich)



mixture with 100 ppm NaI (99 %, Sigma-Aldrich) was utilised, which reduced background signal. In both campaigns, the mass spectrometer was configured to detect positive ions. Because of NaI use, analyte ions were detected almost exclusively as [M]Na$^+$ and other ionisation pathways were suppressed (the only notable exception being nicotine, which was detected as [C$_{10}$H$_{14}$N$_2$]H$^+$). This yields a linear response to mass, avoids matrix effects, and simplifies spectral interpretation (Lopez-Hilfiker et al., 2019). Adducts of an analyte with acetonitrile or methanol molecule(s) may also be detected by the instrument, depending on the voltage settings in the ion transfer optics (i.e., collision energy), but these adducts were observed to have negligible signals with our voltage configurations in both campaigns. The EESI-TOF alternates between direct sampling and sampling through a particle filter to provide a measurement of instrument background (including spray).

Data analysis, including high-resolution peak fitting, was performed using Tofware version 2.5.7 (Tofwerk AG, Thun, Switzerland). Detailed data treatment processes can be found in Stefenelli et al. (2019) and Qi et al. (2019). The EESI-TOF alternates between periods of direct ambient sampling ($M_{amb}$) and filter sampling ($M_{bkgd}$), with the filter periods interpolated to yield an estimated background spectrum during ambient measurements ($M_{bkgd,est}$). The spectra corresponding to aerosol composition ($M_{diff}$) are determined by the difference of $M_{amb}$ and $M_{bkgd,est}$ as shown in Eq. (1a). The corresponding error matrix was estimated by adding in quadrature the uncertainties of the total sampling measurement $s_{amb}$ ($i,j$) and the filter sampling measurement $s_{bkdg,est}$ ($i,j$) as shown in Eq. (1b), which are in turn calculated from ion counting statistics and detector variability (Allan et al., 2003):

$$M_{diff}(i.j) = M_{amb}(i,j) - M_{bkgd,est}(i,j) \qquad (1a)$$

$$s_{diff}(i,j) = \sqrt{s_{amb}^2(i,j) + s_{bkgd,est}^2(i,j)} \qquad (1b)$$

where the unit of all quantities in both equations is counts per second (cps). Ions with a mean SNR smaller than 2 were removed from both matrices, because the signals of these ions were predominantly caused by electrospray and/or instrumental background. Input matrix dimensions are summarised in Table 1.

In theory, EESI-TOF signal for an ion $x$ can be converted from ion flux (cps) to mass concentration (µg m$^{-3}$), according to Eq. (2):

$$Mass_x = I_x \cdot \frac{MW_x}{EE_x + CE_x + IE_x + TE_{m/z}} \cdot \frac{1}{F} \qquad (2)$$

where $Mass_x$ and $I_x$ are the mass concentration (in µg m$^{-3}$), and the ion flux (cps) reaching the detector for an ion $x$, respectively. MW$_x$ represents the molecular weight of the measured ion (e.g., [M]Na$^+$) (Lopez-Hilfiker et al., 2019; Qi et al., 2019; Stefenelli et al., 2019). EE$_x$, CE$_x$, IE$_x$ and TE$_{m/z}$ denote EESI extraction efficiency (the probability that a molecule dissolves in the spray), EESI collection efficiency (the probability that the analyte-laden droplet enters the inlet capillary), ionisation efficiency (the probability that an ion forms and subsequently survives declustering forces induced by evaporation and electric fields), and ion transmission efficiency (the probability that a generated ion is transmitted to the detector, which is independent from chemical identity but depends only on $m/z$), respectively. $F$ indicates the flow rate. In practice, several of these parameters are ion-dependent and remain uncharacterised, and therefore conversion to mass concentration on an ion-by-ion basis cannot currently be achieved (Lopez-Hilfiker et al., 2019). Instead, to facilitate comparison with bulk quantities, we define an "apparent sensitivity ($AS$)" to describe the EESI-TOF response to a measured concentration of species $x$, as shown in Eq. (3):





$$AS_x = \frac{MW_x}{EE_x \cdot CE_x \cdot IE_x \cdot TE_{m/z}} \cdot \frac{1}{F} = \frac{I_x}{Mass_x} \qquad (3)$$

where $I_x$ is the measured ion flux (counts per second, cps) for the ion or factor $x$ detected by EESI-TOF, $Mass_x$ is measured mass concentration ($\mu$g m$^{-3}$) from a reference instrument for the same ion or factor $x$, thus the $AS$ is in the unit of cps ($\mu$g m$^{-3}$)$^{-1}$. In this study, we calculated the apparent sensitivities for different factors from the cPMF results by utilising AMS contribution to the factor profile ($\mu$g m$^{-3}$) as $Mass_x$ and EESI-TOF contribution (cps) as $I_x$. Calculation of these contributions is discussed later in in Sect. 4 using these factor-dependent sensitivities.

### 2.2.3 Estimation of EESI-TOF sensitivities from a multi-variate model

For comparison to the factor-dependent sensitivities determined by the cPMF analysis (see Sect. 3.3), we also estimated sensitivities for SOA factors from molecular formulae of individual analyte ions using parameterisations developed from laboratory measurements of SOA generated from oxidation of limonene (LMN) by ozone and $o$-cresol (cresol) and 1,3,5-trimethylbenzene (TMB) by OH radicals (Wang et al., 2021). As discussed in Sect. 1, the parameterisation can predict the relative sensitivities of ions measured by the EESI-TOF to within a factor of 5, provided that the SOA is derived from a single, known VOC. However, for ambient data, SOA derives from multiple precursor VOCs, increasing uncertainties. For example, SOA isomers generated from different precursors can differ by up to a factor of 20 in relative sensitivity (Wang et al., 2021). This represents a significant source of uncertainty for calibration/parameterisation-based approaches for quantifying SOA factors from source apportionment, but is nonetheless a useful point of comparison.

In the present study, we utilise two well-performing models from Wang et al. (2021), namely the gradient boosting regression and linear ridge regression models, denoted GBR and LRR, respectively, developed in scikit-learn packages in Spyder 4.1.4 and Python 3.8.3. The SOA parameterisation derived from LMN was used to predict the sensitivities for summer SOAs (which are predominantly terpene-derived SOAs), and SOA systems derived from cresol and TMB were used to predict the sensitivities for winter SOAs (which are characterised by aromatics from biomass burning activities). The regression models provide compound-dependent relative sensitivities ($AS_x$) based only on molecular formulae. Then, the EESI-TOF signals for factor are calculated as a signal-weighted average from the respective factor profiles, as shown in Eq. (4):

$$AS_{factor} = \frac{\sum_x I_x}{\sum_x (I_x / AS_x)} \qquad (4)$$

Here $I_x$ denotes the contribution to the factor profile of each ion $x$. Because the model parameterisations are based on laboratory SOA that contained only the CHO group, while the resolved OA sources in this study include both CHO and CHON, we approximate the total factor sensitivity by assuming the average EESI-TOF sensitivity to CHON ions is equal to the average sensitivity of CHO ions (on a factor-by-factor basis). Note that the ions from the CHO group contribute a major fraction in SOA mass for each factor, comprising 85.2 %, 78.1 %, 57.3 % and 76.3 % for DaySOA1, DaySOA2, NightSOA1 and NightSOA2 for summer and 77.9 % and 75.0 % to SOA1 and SOA2 for winter, reducing the uncertainties introduced by this assumption (these factors will be discussed in Sect. 3.2).

### 2.3 Source Apportionment Method



In this paper, source apportionment was performed using the positive matrix factorisation (PMF) model on a single dataset containing both AMS and EESI-TOF data. We denote the overall method governing analysis of such a merged dataset as "combined PMF" (cPMF), while "PMF" denotes both the general PMF model and single-run executions by the Multilinear Engine solver (see Sect. 2.3.1), which are identical for PMF and cPMF. This section presents an overview of the cPMF method, with detailed descriptions of each step in the referenced sub-sections. Section 3.1 then presents its application to the test datasets, including dataset-specific decisions (e.g., which factors to constrain, criteria for accepting/rejecting solutions) required during certain steps. The overall procedure is outlined in Fig. 1, with the main steps as follows:

1)  Conventional PMF analyses are conducted on the standalone EESI-TOF and AMS datasets with synchronised time resolution, including constraints on factor profiles as necessary. Residuals from the optimised solutions are used as a reference to retrieve a balanced solution (step 4).

2)  The EESI-TOF and AMS datasets with synchronised time resolution are combined into a single input matrix. This input matrix contains OA spectra from EESI-TOF and AMS, as well as the $NO^+$ and $NO_2^+$ ions measured by the AMS due to the contributions of organonitrates to these ions (Sect. 2.3.1).

3)  For any factors that are to be constrained, joint AMS/EESI-TOF profiles are constructed (Sect. 2.3.2 and 3.1.2).

4)  An exploratory PMF analysis is conducted on the joint AMS/EESI-TOF matrix. This consists of a 2-D exploration of the solution space defined by the number of factors ($p$) and relative instrument weight ($C$) (Sect. 2.3.3). The instrument weight ensures that both instruments are well-represented in the solution and is assessed by comparing residuals from cPMF and standalone PMF. For computational efficiency, the profiles of all constrained factors are not allowed to deviate from their reference profiles. Solutions in which both instruments receive approximately equal weight are evaluated for environmental interpretability, with the most interpretable solution utilised as the base case for further analysis. Note that the base case is fully defined by $C$, $p$, and the set of constrained factor profiles.

5)  From the selected base case, 1000 PMF runs are conducted, which combine bootstrap analysis with random selection of $a$-values (i.e., tightness of constraint) for the constrained factors within predetermined limits that are defined on a factor-by-factor basis (Sect. 2.3.4). This requires the following as prerequisites:

    a.  Definition of dataset-specific criteria for acceptance/rejection of individual runs (Sect. 3.1.4).

    b.  Determination of the $a$-value range on a factor-by-factor basis giving a reasonable acceptance probability, i.e., sufficient rejection rate to ensure adequate exploration while maintaining computational efficiency (Sect. 3.1.4).

6)  The final cPMF result is taken as the mean of all accepted solutions from the bootstrap/$a$-value analysis (step 5), with uncertainties represented by the standard deviation. From this mean solution, quantitative time series and EESI-TOF factor-specific sensitivities are calculated.



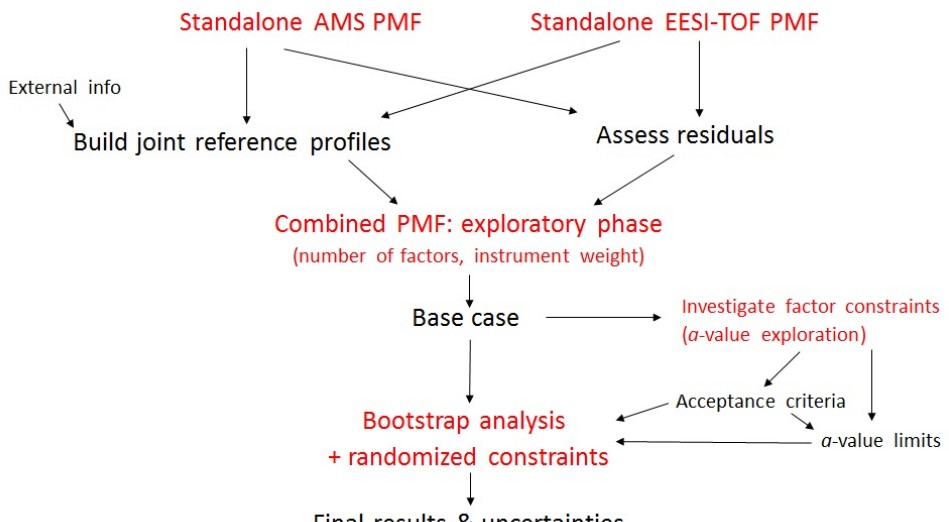

2 Figure 1. Flow chart summary of cPMF analysis workflow. Red text denotes PMF model operations
3 while black text denotes inputs, outputs, and/or analysis decisions.

### 2.3.1 Positive matrix factorisation (PMF)

Positive matrix factorisation (PMF) is implemented using the Multilinear Engine (ME-2) (Paatero, 1999), with model configuration and post-analysis performed with the Source Finder (SoFi, version 6B) (Canonaco et al., 2013), programmed in Igor Pro 6.39 (Wavemetrics, Inc.). PMF is a bilinear receptor model, which operates on an input data matrix $\mathbf{X}$ (here the mass spectral time series collected by EESI-TOF and/or AMS) and uncertainty matrix $\mathbf{S}$, which corresponds point-by-point to $\mathbf{X}$. PMF describes $\mathbf{X}$ as a linear combination of static factor profiles (in this case characteristic mass spectra, representing specific sources and/or atmospheric processes) and their corresponding time-dependent source contributions, as described in Eq. (5):

$$\mathbf{X} = \mathbf{G} \times \mathbf{F} + \mathbf{E} \tag{5}$$

Here $\mathbf{X}$ has dimensions of $m{\times}n$, representing $m$ measurements of $n$ variables (here ions), $\mathbf{G}$ and $\mathbf{F}$ are respectively the factor time series with the dimension of $m{\times}p$, and factor profiles with the dimension of $p{\times}n$, where $p$ is the number of factors in the PMF solution, and is determined by the user. $\mathbf{E}$ is the residual matrix and defined by Eq. (5). Figure 2 shows a conceptual representation of the combined EESI-TOF and AMS input data matrix $\mathbf{X}$. The corresponding uncertainty matrix $\mathbf{S}$ and residual matrix $\mathbf{E}$ are constructed in the same way (Slowik et al., 2010). Note that the AMS component of $\mathbf{X}, \mathbf{S}$ and $\mathbf{E}$ is in µg m$^{-3}$, and the EESI-TOF component is in cps. Also, $\mathbf{X}$ includes not only organic ions from the AMS, but also $NO^+$ and $NO_2^+$, which contain a large fraction of the AMS signal derived from organonitrates (Farmer et al., 2010).





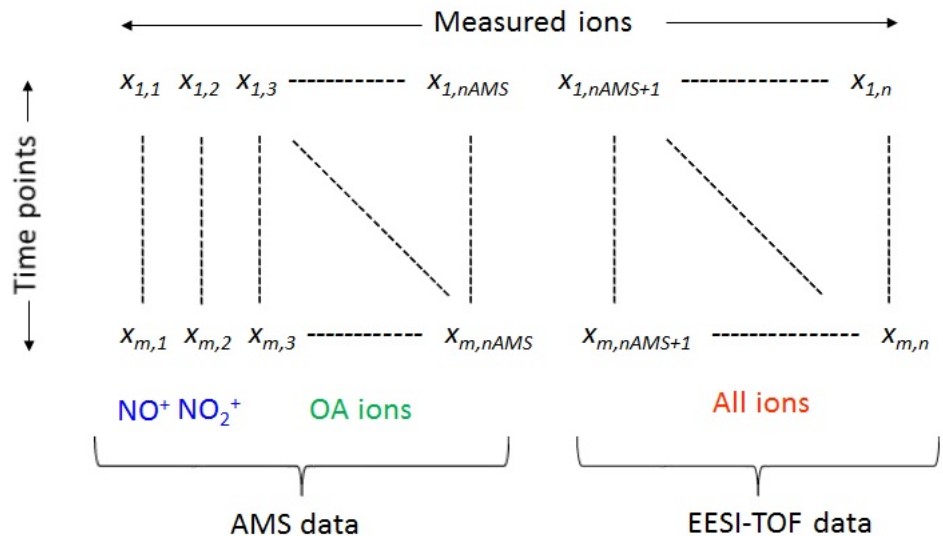

Figure 2. Schematic of the combined EESI-TOF and AMS input data matrix (**X**) for cPMF. Matrix
dimensions for the summer and winter datasets are provided in Table 1.
Equation 5 is solved by a least-squares algorithm that iteratively minimises the quantity $Q$, which is
defined in Eq. (6) as the sum of the squares of the uncertainty-weighted residuals:

$$Q = \sum_i \sum_j \left(\frac{e_{ij}}{s_{ij}}\right)^2 \qquad (6)$$

Here $e_{ij}$ is an element in the residual matrix **E**, and $s_{ij}$ is the corresponding element in the uncertainty
matrix, where $i$ and $j$ are the indices representing time and ion (or $m/z$), respectively. The theoretical $Q$,
denoted as $Q_{\text{expected}}$, is estimated by Eq. (7):

$$Q_{\text{expected}} \cong mn - p(m + n) \qquad (7)$$

where $m$ and $n$ denote the number of measurements (here time points) and the number of variables (ions
or $m/z$), respectively, and $p$ denotes the number of factors in this PMF solution.
Due to the nature of the cPMF **X** matrix, each retrieved factor has a single time series, which can be
expressed in the concentration units of either instrument, and the factor profile contains both an AMS
and an EESI-TOF component. The factor time series for a single factor $k$ is calculated as follows:

$$\left(g_{i,k}\right)_{inst} = g_{i,k} \cdot \sum_{j=inst} f_{k,j} \qquad (8)$$



Here $(g_{i,k})_{inst}$ refers generally to the time series in the measurement units of a given instrument, which
we denote $(g_{i,k})_{AMS}$ or $(g_{i,k})_{EESI}$, and the $j=inst$ formalism denotes the set of ions measured by the
respective instrument. For ease of interpretation, we report the instrument contribution to each factor
profile as the mass spectrum (in the respective instrument units) that would be obtained for a factor
mass concentration of 1 µg m$^{-3}$. This is expressed as follows, for a single factor $k$:

$$(f_{k,j})_{inst} = \left( \frac{f_{k,j} \overline{(g_{i,k})_{AMS}}}{g_0} \right)_{j=inst} \tag{9}$$

Here $\overline{(g_{i,k})_{AMS}}$ denotes the mean of the factor time series in AMS units (µg m$^{-3}$), $g_0$ is a reference mass
concentration (chosen here as 1 µg m$^{-3}$), the $j=inst$ formulation again refers to all ions measured by a
given instrument. We refer to the organic fraction of AMS profile components and EESI-TOF profile
components as $(f_{k,j})_{AMS}$ and $(f_{k,j})_{EESI}$ , respectively. The EESI-TOF apparent sensitivity ($AS_x$,
defined in Eq. (3)) can then be calculated for a single factor $k$ as:

$$AS_k = \left( \frac{\overline{(g_{i,k})_{EESI}}}{\overline{(g_{i,k})_{AMS}}} \right)_{j=inst} \tag{10}$$

Evaluation of factor interpretability for PMF analysis of the data from a single instrument typically
includes: 1) correlation of the time series with external data; 2) comparison of factor diurnal cycles with
known source activity and previous measurements; 3) identification of source-specific spectral features.
In addition to these three points, factors from cPMF were also interpreted by considering the consistency
of spectral features between the AMS and EESI-TOF, e.g., factors originated from fresh biomass
burning activities are characterised by elevated signal from $C_2H_4O_2^+$ in the AMS spectrum and
levoglucosan in the EESI-TOF spectrum.
23        2.3.2    Constraints on factor profiles
Different combinations of the **G** and **F** matrices may result in solutions with the same or similar $Q$
(rotational ambiguity), which in practice leads to mixed or unresolvable factors. Here we explore a
subset of the possible PMF/cPMF solutions in which one or more factor profiles are constrained using
the $a$-value approach to direct solutions towards environmentally meaningful rotations. These factors
are constrained using reference profiles, with the scalar $a$ ($0 \leq a \leq 1$) determining the tightness of
constraint as follows:
$$(f_{k,j})_{sol} = (f_{k,j})_{ref} \pm a \times (f_{k,j})_{ref} \tag{11}$$
Here $(f_{k,j})_{ref}$ represents the reference profile and $(f_{k,j})_{sol}$ the final profile returned by the model. Due
to the renormalisation of matrices after PMF runs, the final values in $(f_{k,j})_{sol}$ may slightly exceed the
prescribed range. This approach has been shown to significantly improve the model performance
relative to unconstrained PMF (Crippa et al., 2014; Canonaco et al., 2013; Daellenbach et al., 2016;
Stefenelli et al., 2019; Qi et al., 2019).
As shown in Eqs. (8-10), the EESI-TOF factor sensitivity is intrinsic to a given factor (via its profile).
However, in the cPMF, it may be desirable to constrain a factor for which a single reference profile
incorporating both AMS and EESI-TOF mass spectra is not available. For example, a factor may be
detectable by only one instrument, or reference profiles may have been retrieved independently for each





1  instrument (e.g., from different studies). In such cases, the cPMF reference profile, $\left(f_{k,j}\right)_{j=all,ref}$ is

2  constructed from merged individual profiles as follows:

$$
\frac{\left(f_{k,j}\right)_{j=all,ref}}{1\ \mu g\ m^{-3}} = \begin{cases} \dfrac{\left(f_{k,j}\right)_j}{\sum_j \left(f_{k,j}\right)_j}, & j \in AMS, ref \\[3mm] AS_k \cdot \dfrac{\left(f_{k,j}\right)_j}{\sum_j \left(f_{k,j}\right)_j}, & j \in EESI, ref \end{cases} \tag{12}
$$

Here $\left(f_{k,j}\right)_j$ denotes standalone reference profiles for the AMS and EESI-TOF, respectively. Note that
although Eq. (12) requires an initial value of $AS_k$ to be assumed prior to PMF execution and utilised
during the exploratory phase of cPMF (Sect. 2.3, step 4), selection of a non-zero *a*-value during
bootstrap analysis (Sect. 2.3, step 5) allows the final $AS_k$ to be determined by the algorithm within the
designated boundaries. Therefore, only a reasonable *a priori* estimate is required. In the case that a
factor is undetectable by the EESI-TOF (e.g., non-oxygenated hydrocarbons comprising traffic-related
factors), a value of $AS_k$ is assumed that fixes the EESI-TOF contribution near zero. In the present study,
we utilised $AS_k = 0.01$ cps $(\mu g\ m^{-3})^{-1}$ when this situation arose. For contrast, $AS_k$ for factors detectable
by both instruments ranged from approximately 100 to 1000 cps $(\mu g\ m^{-3})^{-1}$.
2.3.3   Instrument weighting
For both factor interpretation and quantitative analysis, it is important that both instruments be well-
represented in any accepted PMF solution. In principle, the extent to which PMF can explain a variable
$x_{i,j}$ is limited by the measurement uncertainty, $s_{i,j}$; that is, the expectation value of the scaled residual
$(e_{i,j}/s_{i,j})$ is 1 (i.e., $Q/Q_{expect} \sim 1$). In practice, $e_{i,j}/s_{i,j}$ may be systematically above or below 1, and differ
between instruments, for several reasons. First, the accuracy of the error calculation may be
systematically different between instruments, leading to systematic differences in the effect of residuals
from a given instrument on $Q$. Second, the extent of internal correlations in the dataset may differ
between instruments. For example, fragmentation/thermal decomposition in the AMS can lead to
sequences of correlated ions (e.g., $C_nH_{2n+1}^+$ for alkanes). In contrast, for the EESI-TOF measurement of
individual molecular ions, ion-to-ion correlations depend solely on particle composition. Finally, even
for a case where ion-by-ion signal-to-noise and the extent of internal correlations is equal between
instruments, the relative number of variables (ions) included in the dataset may affect the weight due to
small drifts in instrument performance, modelling errors in PMF, and the prevalence of
transient/variable sources not fully captured by PMF. Therefore, it is important to assess the relative
weight of the two instruments and rebalance if necessary. We define a balanced solution as one in which
there are no systematic differences between quality of fit for different instruments (Crippa et al., 2013b;
Slowik et al., 2010). However, note that variable-to-variable differences in the $e_{i,j}/s_{i,j}$ within the dataset
of a single instrument are permitted (as in standalone PMF).
The instrument weighting process follows the method previously proposed by Slowik et al., (2010), in
which weighting is performed by applying a weighting factor *C* to the uncertainties and evaluated by
comparison of the AMS vs. EESI-TOF residuals. Here we utilise the same weighting method, but
propose an improved evaluation metric. Instrument weighting is performed by applying a weighting





factor $C$ to the components of the uncertainty matrix $\mathbf{S}$ corresponding to one of the two instruments.
This increases/decreases the contribution of that instrument's residuals to $Q$, thereby changing its
weight within the PMF solver. In this paper, we applied the weighting factor, denoted $C_{\text{EESI}}$, to the
columns of $\mathbf{S}$ corresponding to ions measured by the EESI-TOF, according to Eq. (13):

$$\begin{cases} (s'_{i,j})_{j=EESI} = \dfrac{(s_{i,j})_{j=EESI}}{C_{\text{EESI}}} \\ (s'_{i,j})_{j=AMS} = (s_{i,j})_{j=AMS} \end{cases} \tag{13}$$

Note that $C_{\text{EESI}} = 1$ is equivalent to an unweighted solution; and $C_{\text{EESI}} > 1$ means the uncertainty matrix
of EESI-TOF decreases, which upweights the EESI-TOF.
As noted above, a balanced solution is defined as one in which the quality of fit to a given ion (assessed
via scaled residuals, $e_{ij}/s_{ij}$) is independent of the instrument performing the measurement. In previous
work (Slowik et al., 2010; Crippa et al., 2013), the metric used to assess this was the mean of the
absolute scaled residuals. This metric assumes that the optimised solution for each individual instrument
yields approximately the same $Q/Q_{exp}$. In practice, this may vary between instruments for the reasons
described above. Further, this metric can be unduly influenced by a few large outliers. Therefore, we
employ a new approach which references the residuals from the combined dataset to those obtained
from the final solutions from single-instrument PMF, which having been selected as the optimal
representation of environmental data are assumed to likewise provide the optimised distributions of
single-instrument residuals. The new method is as follows:
20        1) From the result of each single instrument PMF (here AMS PMF, EESI-TOF PMF), calculate
the scaled residual ($e_{ij}/s_{ij}$) probability distribution over the entire (single instrument) dataset. Here we
denote the scaled residual probability distribution function in the scaled residual ($e_{ij}/s_{ij}$) space for EESI-
TOF and AMS as $P_{\text{EESI}}(e_{ij}/s_{ij})$ and $P_{\text{AMS}}(e_{ij}/s_{ij})$, respectively.
24        2) Calculate the overlap fraction $F_{\text{overlap}}$ between the AMS and EESI-TOF scaled residual
probability distributions from the single instrument solutions, according to Eq. (14):

$$F_{\text{overlap}} = \int \min \left( P_{\text{EESI}}\left(\frac{e_{ij}}{s_{ij}}\right), P_{\text{AMS}}\left(\frac{e_{ij}}{s_{ij}}\right) \right) \tag{14}$$

where $P_{\text{EESI}}(e_{ij}/s_{ij})$ and $P_{\text{AMS}}(e_{ij}/s_{ij})$ indicates the probability of occurrence of AMS and EESI-TOF at the
point $e_{ij}/s_{ij}$ in scaled residual space, respectively. Given the previously mentioned assumption that the
single-instrument solutions represent the optimal representation of the data for the individual
instruments, the $F_{\text{overlap}}$ calculated at this step is the value that should likewise be obtained from a
balanced solution to the combined dataset. Therefore, we define the quantity $F_{\text{overlap}}^*$ as the $F_{\text{overlap}}$ of
the final single-instrument PMF solutions.
34        3) For the combined dataset, calculate $F_{\text{overlap}}$ as a function of a two-dimensional exploration
of the space defined by weighing factor ($C_{\text{EESI}}$) and the number of factors ($p$). This exploration is
necessary because the scaled residuals have been empirically observed to depend not only on $C$ but also
$p$ (Crippa et al., 2013b; Slowik et al., 2010), likely because $p$ affects the degrees of freedom in the
solution. We select for further analysis the set of solutions in which $F_{\text{overlap}}$ does not greatly differ from
$F_{\text{overlap}}^*$, as given by Eq. (15):

$$\left| F_{\text{overlap}}(C, p) - F_{\text{overlap}}^* \right| < \beta \tag{15}$$

where the threshold of absolute difference is defined as $\beta$. Here $\beta$ is a subjective parameter chosen to
allow a manageable number of solutions to be selected for detailed inspection. For computational





efficiency, if one or more factors are constrained, we choose $a = 0$ for all constrained factors at this
preliminary exploration stage and will explore the $a$-value range(s) for constraint(s) for further
bootstrapping analysis once the $C$ and $p$ are determined.
The balanced solutions satisfying Eq. (15) are then evaluated using the same metrics as in standard
PMF analysis to select the solution with the greatest explanatory power. This solution is used as the
base case for bootstrap analysis and, if one or more factors are constrained, simultaneous randomised
$a$-value trials.
10                2.3.4    Bootstrap/constraint sensitivity analysis on the combined dataset
Bootstrap analysis (Davison and Hinkley, 1997) is frequently used to characterise solution stability,
reproducibility and estimate uncertainties. In typical bootstrap analysis, a set of new input and error
matrices are created by random resampling of rows from the original input data and error matrices. The
resulting resampled matrices preserve the original dimensions of the input data matrix, but randomly
duplicate some time points while excluding others (Paatero et al., 2014). In the present analysis, we
combined bootstrap analysis with randomised selection of $a$-values for all constrained factors within
predetermined limits defined on a factor-by-factor basis. Since the constrained factors use reference
profiles constructed with an estimated $AS_k$ (see Eq. (12)), this combined bootstrap/constraint analysis
allows recalculation of $AS_k$ within PMF. As a result, the final reported solution is the average of all
accepted bootstrap runs, with uncertainties in factor profiles and time series taken as the standard
deviation.
Within this analysis, the range of $a$-values explored for a given factor may have a significant effect on
the acceptance probability. A very low acceptance probability is undesirable because it is
computationally inefficient, while a very high acceptance probability is also undesirable because it
implies the solution space is inadequately explored due to excessively restrictive $a$-values (Canonaco
et al., 2021). Therefore, we conduct pre-tests to estimate the $a$-value range leading to a reasonable
acceptance probability. This is done by a set of 2-dimensional $a$-value ("multi-2D") scans in which the
$a$-values of two constrained factors are varied stepwise from 0 to 1 with a step size of 0.1 (i.e., 121 runs),
while the $a$-values of other constrained factors are held at 0. The results of all multi-2D runs for a given
factor are combined to determine the acceptance probability as a function of $a$-value, and upper and
lower $a$-value boundaries are assessed. The acceptance criteria are dataset-specific and discussed later
(Sect. 3.1.4). When the number of constrained factors ($p_{ref}$) = 2, the multi-2D algorithm is equivalent to
an explicit exploration of all possible $a$-value combinations. However, for $p_{ref}$ >2, multi-2D is much
more computationally efficient, because it increases as $p_{ref}$ ($p_{ref}$ -1)/2, whereas the explicit method
increases as the factorial of $p_{ref}$ . For the datasets used here, in which $p_{ref}$ is 3 (summer) and 4 (winter),
the multi-2D approach decreases the number of runs required for $a$-value pre-scans by factors of ~4 and
~20, respectively.
Acceptance criteria consist of both the assessment of specific features of selected factor profiles/time
series (see Sect. 3.1.4), as well as a general evaluation of whether the solution is qualitatively similar to
the base case. That is, we require that the time series of each factor from a PMF run to be unambiguously
related to the corresponding base case factor (Stefenelli et al., 2019; Vlachou et al., 2019; Tong et al.,
2021). The key steps of this method are summarised below: 1) identify a base case, which as discussed
above is defined by a weighting factor $C$, number of factors $p$, and set of constrained factors with the $a$-
value set to 0; 2) calculate the Spearman correlation between the time series of base case and the multi-
2D scans, which yields a correlation matrix with the highest correlation values on the diagonal; 3) each
correlation coefficient on the matrix diagonal must be by a statistically significant margin (using





different confidence levels from a $t$ test) than any value on the intersecting row or column. In the current
study, we selected a confidence level of 0 for this base case/bootstrap correlation test, representing the
most permissive application of this criterion. That is, we require only that the diagonal matrix mentioned
above can be constructed, i.e., that there is a unique 1:1 correspondence between base case factors and
factors from the bootstrap/$a$-value analysis.
The final set of PMF runs consisted of 1000 bootstrap runs, conducted at a single combination of $C_{EESI}$
and $p$, with $a$-values randomly selected with a step size of 0.05 for summer and 0.1 for winter within
the factor-specific limits determined via the multi-2D pre-scans. The same acceptance criteria utilised
for the multi-2D pre-scans were also used for the bootstrap runs. As a final solution, we report the mean
factor profiles and time series determined from all accepted bootstrap runs, with the standard deviation
taken to represent the uncertainty of the analysis procedure. Although not currently implemented within
the analysis software used, we note that in theory it would be possible to additionally include random
$C_{EESI}$ selection (within a predefined range corresponding to balanced solutions) and randomised $AS_k$ for
constrained profiles (within a user-defined range) in this stage of the analysis and in calculation of the
final model outputs.

## 3. Results

Due to the complexity of the analysed datasets (2 seasons × 3 PMF methods), we use the following
convention for identifying factors: factorName$_{season,method}$, where "factorName" is the name of the factor
(e.g., COA for cooking-related organic aerosol), "season" denotes either the summer ("S") or winter
("W") dataset, and "method" refers to PMF on standalone AMS dataset ("A"), standalone EESI-TOF
dataset ("E"), or combined dataset ("C"). For example, COA$_{S,C}$ stands for the cooking-related factor
retrieved from cPMF applied to the summer dataset.

### 3.1 Method validation and solution selection

#### 3.1.1    PMF analysis of single-instrument datasets

Single-instrument AMS and EESI-TOF PMF analysis was previously conducted and validated for both
the summer and winter datasets (Qi et al., 2019; Stefenelli et al., 2019). To determine the $F^*_{overlap}$, the
EESI-TOF-only PMF was re-run on only the period when both AMS and EESI-TOF were operating.
In addition, the AMS PMF analysis was re-run on the same period, but with the $NO^+$ and $NO_2^+$ ions
included. As discussed above, these ions contain a large fraction of the AMS signal deriving from
organonitrates. For EESI-TOF-only PMF analysis in both datasets, we used the same constraints as in
the referenced studies, that is, cooking-influenced OA (COA$_{S,E}$) was constrained for the summer dataset
and cigarette-smoking OA (CSOA$_{W,E}$) was constrained for the winter dataset. For AMS-only PMF
analysis, the only constrained factor in the original studies was hydrocarbon-like OA during winter
(HOA$_{W,A}$). We additionally constrained inorganic nitrate (InorgNit) in both the summer and winter
datasets, by including 1) the $CO_2^+/(NO^+ + NO_2^+)$ ratio, where the $CO_2^+$ signal was produced by reaction
of nitrate on the vaporiser (Pieber et al., 2016), as well as minor organic contaminants, and 2) $NO^+/NO_2^+$
ratio. In summer, we took the mass spectrum acquired from the $NH_4NO_3$ calibration period during the





campaign to calculate the ratios in 1) and 2), whereas in winter, we constructed the reference using the
two ratios from the ambient measurements (2.54) during periods of high nitrate to organic ratios.
Fig. S1 and Fig. S2 show the results from these single-instrument AMS and EESI-TOF PMF analyses
for summer and winter, respectively, as well as a comparison with the factor time series from the
original studies. Because the results are very similar to the single-instrument studies, they are discussed
only briefly here. The AMS-only PMF yielded five OA factors consistent with those of Stefenelli et al.
(2019), namely hydrocarbon-like OA (HOA$_{S,A}$), cooking-influenced OA (COA$_{S,A}$), cigarette-smoking
OA (CSOA$_{S,A}$), more oxygenated OA, MO-OOA$_{S,A}$, and less oxygenated OA (LO-OOA$_{S,A}$), and
additionally a factor dominated by NO$^+$ and NO$_2^+$ in a ratio consistent with that of ammonium nitrate,
denoted InorgNit$_{S,A}$. The main difference between these results and those reported by Stefenelli et al.
(2019) is some exchange of signal between MO-OOA$_{S,A}$ and LO-OOA$_{S,A}$. In addition, the contribution
from NO$^+$ and NO$_2^+$ is not solely apportioned to InorgNit$_{S,A}$ but also to factors such as LO-OOA$_{S,A}$;
however, this does not affect the identity and interpretation of these factors.
Similarly, for the winter dataset, seven factors were resolved consistent with the OA factors determined
by Qi et al. (2019), namely HOA$_{W,A}$, COA$_{W,A}$, LO-OOA$_{W,A}$, MO-OOA$_{W,A}$, biomass burning OA
(BBOA$_{W,A}$), event-specific OA (EVENT$_{W,A}$) and nitrogen-rich OA (NitrogenOA$_{W,A}$), as well as a new
factor consistent with InorgNit$_{W,A}$. Apart from being apportioned to InorgNit, NO$^+$ and NO$_2^+$ were also
apportioned to non-InorgNit factors, indicating organonitrate content and/or imperfect attribution of
inorganic NO$^+$ and NO$_2^+$ to these factors. Although the NO$^+$ and NO$_2^+$ contributions in some non-
InorgNit factors are significant, causing some changes in the factor time series compared to those in Qi
et al. (2019), the main features of the spectra from other OA components (i.e., ions other than NO$^+$ and
NO$_2^+$) in these factors are retained.
As discussed in Sect. 2.3.3, scaled residual probability distributions, i.e., $P$ ($e_{ij}/s_{ij}$), for the selected
single-instrument solutions were calculated and are shown in Fig. 3. As discussed in Eq. (14), this yields
values for $F_{\text{overlap}}^*$ , which are calculated to be 0.769 in summer and 0.899 in winter.

a)

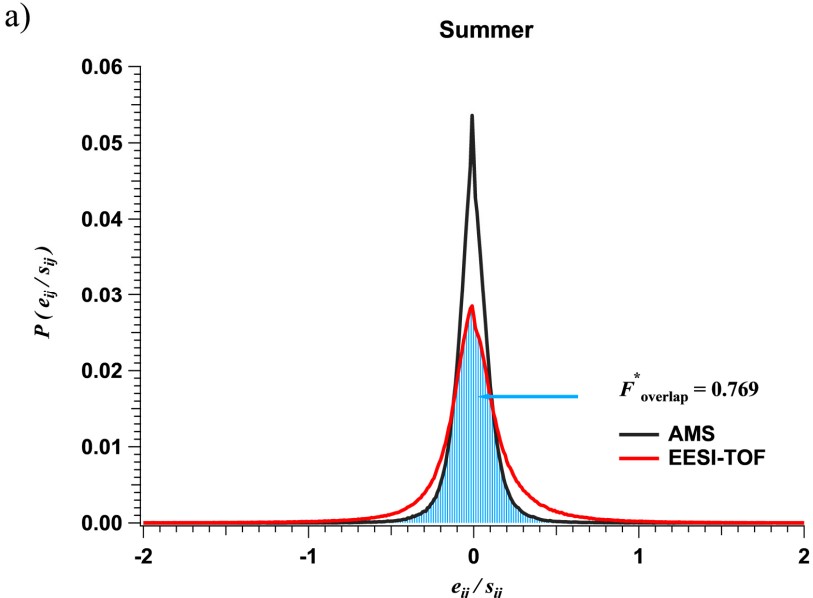

b)

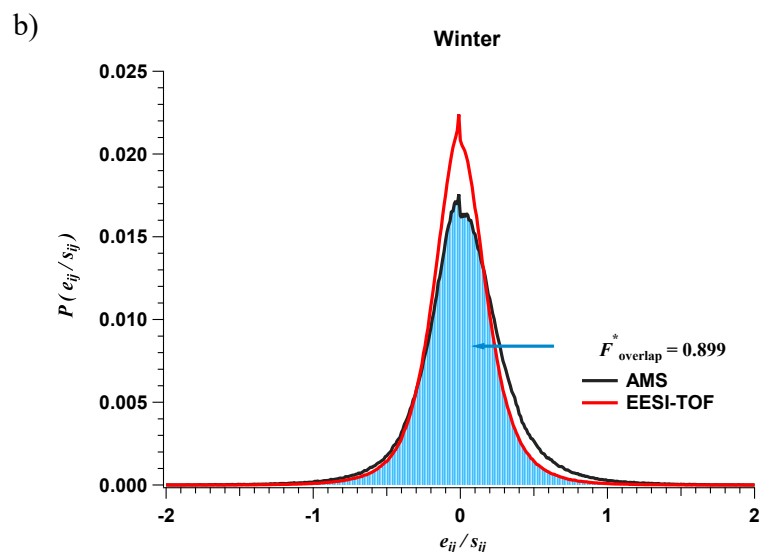

Figure 3. Scaled residual probability distributions and region of overlap from individual AMS PMF solution and EESI-TOF PMF solutions for the summer (a) and winter (b) datasets. Red and black lines show the residual distributions for the EESI-TOF and AMS, respectively; shading denotes the region of overlap.



**Table. 1** Summary of parameters for the PMF analysis of re-analysed summer and winter datasets, and
the combined dataset. All datasets include AMS measurements of $NO^+$ and $NO_2^+$.

| | | EESI-TOF | AMS | Combined |
|---|---|---|---|---|
| **Summer** | Matrix dimensions (time points × $m/z$) | $1779 \times 507$ | $1779 \times 287$ | $1779 \times 794$ |
| | Time period | 20 to 26 June 2016 | 20 to 26 June 2016 | 20 to 26 June 2016 |
| | Time resolution (min) | 5 | 5 | 5 |
| | Range of $p$ analysed | 6 | 6 | 5-10 |
| **Winter** | Matrix dimensions (time points × $m/z$) | $6142 \times 892$ | $6142 \times 258$ | $6142 \times 1150$ |
| | Time period | 25 Jan to 4 Feb 2017 | 25 Jan to 4 Feb 2017 | 25 Jan to 4 Feb 2017 |
| | Time resolution (min) | 1 | 1 | 1 |
| | Range of $p$ analysed | 12 | 8 | 7-14 |

4        **3.1.2**    **Construction of reference profiles**

In the cPMF analysis, the factor profiles for HOA, COA, and InorgNit were constrained in both the
summer and winter datasets, while CSOA was constrained in winter only. All reference profiles were
constructed according to Eq. (12). Here we discuss the methods used to determine $\left(f_{k,j}\right)_{j=AMS,ref}$,
$\left(f_{k,j}\right)_{j=EESI,ref}$, and the estimated $AS_k$ used to synthesise the reference profile. Note that COA and
CSOA are retrieved by both AMS and EESI-TOF, while HOA and InorgNit are not retrieved by the
EESI-TOF in the configuration used for these campaigns. Specifically, no HOA-sensitive EESI-TOF
extraction/ionisation scheme has yet been developed, while the measurable ion corresponding to
inorganic nitrate, $[NaNO_3]Na^+$, has been detected in other studies (Tong et al., 2021) but falls below the
$m/z$ transmission window used here.
For summer $COA_{S,C}$, $\left(f_{k,j}\right)_{j=AMS,ref}$ and $\left(f_{k,j}\right)_{j=EESI,ref}$ were taken from the factor profiles for
$COA_{S,A}$ and $COA_{S,E}$, respectively. $AS_{COA_S}$ was calculated as the ratio of the mean signals of $COA_{S,E}$
(cps) to $COA_{S,A}$ (µg m⁻³). For $HOA_{S,C}$, $\left(f_{k,j}\right)_{j=AMS,ref}$ the HOA profile of Crippa et al. (2013b) was
used, and for $InorgNit_{S,C}$, it was taken to be the mass spectrum acquired from the $NH_4NO_3$ calibration
period during the campaign. The latter included the $CO_2^+$ signal produced by reaction of nitrate on the
vaporiser (Pieber et al., 2016), here observed with a $CO_2^+/(NO^+ + NO_2^+)$ ratio of 0.0345, as well as
minor organic contaminants. For both $HOA_{S,C}$ and $InorgNit_{S,C}$ all ions in $\left(f_{k,j}\right)_{j=EESI,ref}$ were set at
the same intensity, and $AS_k$ was selected to be 0.01 cps (µg m⁻³)⁻¹.
The $COA_{W,C}$ reference profile was constructed using the identical method as for $COA_{S,C}$, with $COA_{W,A}$
and $COA_{W,E}$ as references. For $CSOA_{W,C}$, $\left(f_{k,j}\right)_{j=EESI,ref}$ was taken to be the $CSOA_{W,E}$ profile.
However, because the AMS did not resolve CSOA in the winter, we used the $CSOA_{S,A}$ profile for
$\left(f_{k,j}\right)_{j=AMS,ref}$ and estimated $AS_{CSOA,w}$ as follows:

$$AS_{CSOA,w} = \frac{AS_{COA,w}}{AS_{COA,s}} \cdot AS_{CSOA,s} \tag{16}$$





where $AS_{COA,s}$, $AS_{CSOA,s}$, and $AS_{COA,w}$ are the EESI-TOF apparent sensitivities of the corresponding
factors, calculated assuming direct correspondence between the AMS and EESI-TOF factors sharing
the same name (Stefenelli et al., 2019; Qi et al., 2019).
The reference profile for $HOA_{W,C}$ is identical to $HOA_{S,C}$, and constructed in the same way using the
same profile as in the summer dataset. Unlike summer, the calibration mass spectrum of $NH_4NO_3$ was
not used as the reference profile for $InorgNit_{W,C}$, because the $NO^+/NO_2^+$ in the $NH_4NO_3$ calibration
period (1.58) was not consistent with that observed from ambient measurements (2.54) during periods
of high nitrate to organic ratios, possibly indicating contributions from non-$NH_4^+$ cations. Instead, the
InorgNit reference profile of AMS ions was constructed based on these features: 1) the $NO^+/NO_2^+$ ratio
(2.54) from 26 Jan 2016 to 31 Jan 2016, when the instrument remained stable and the ratio of nitrate to
OA was high, suggesting the contribution from organonitrates to $NO^+$ and $NO_2^+$ was low, 2) the
$CO_2^+/(NO^+ + NO_2^+)$ ratio (0.00026) was assumed to be the same as during the calibration period in the
Zurich winter campaign and 3) the ratio of intensity of each organic ion to $CO_2^+$ was kept the same as
during the calibration period in the Zurich winter campaign. Then $\left(f_{k,j}\right)_{j=EESI,ref}$ and $AS_{InorgNit,w}$
were determined using the same method as in summer.

### 3.1.3    Determination of $C_{EESI}$ and number of solutions

Because $F_{overlap}$ depends on both the weighting factor $C_{EESI}$ and the number of factors $p$, an exploration
of this two-dimensional space is required. As discussed earlier, for computational efficiency the $a$-
values of all constrained factor profiles were set to zero during this initial exploration. For the summer
dataset, in which both the AMS-only and EESI-TOF-only PMF analyses yielded 6 factors, the cPMF
was explored from 5 to 12 factors with $HOA_{S,C}$, $COA_{S,C}$ and $InorgNit_{S,C}$ constrained. For the winter
dataset, in which the AMS-only and EESI-TOF-only PMF analyses yielded 8 and 11 factors,
respectively, the cPMF was explored from 7 to 15 factors with $HOA_{W,C}$, $COA_{W,C}$, $CSOA_{W,C}$ and
$InorgNit_{W,C}$ constrained. For the summer dataset, $C_{EESI}$ was explored from 0.1 to 100, and in winter
from 0.001 to 50. The results of this exploration are shown in Fig. 4a and Fig. 4b, which present
$|F_{overlap} - F^*_{overlap}|$ as a function of $C_{EESI}$ and $p$ for the summer and winter datasets, respectively.
The Zurich summer dataset displays the expected trend of $|F_{overlap} - F^*_{overlap}|$ with respect to $C_{EESI}$.
Balanced solutions are found at intermediate values of $C_{EESI}$, with lower and higher values yielding
solutions in which the AMS and EESI-TOF, respectively, are overweighted. Examples of scaled
residual distributions for these three cases (AMS overweighted, balanced, and EESI-TOF overweighted)
are shown in Fig. S3. The black box in Fig. 4a denotes a set of solutions satisfying the criterion in Eq.
(15), which are selected for further inspection. The value of $\beta$ is selected empirically to yield a practical
number of solutions for manual inspection, with 0.02 chosen for summer and 0.005 for winter. Factor
profiles and time series for solutions satisfying the $\beta$ criterion, comprising solutions with 6 to 9 factors
(black box in the figure) are shown in Figs S4 to Fig. S13. An 8-factor solution was chosen as the best
representation of the data, and included $HOA_{S,C}$, $COA_{S,C}$, $CSOA_{S,C}$, $InorgNit_{S,C}$, two daytime SOAs
($DaySOA1_{S,C}$ and $DaySOA2_{S,C}$) and two nighttime SOAs ($NightSOA1_{S,C}$ and $NightSOA2_{S,C}$), discussed
in detail in Sect. 3.2.1. Solutions with higher numbers of factors yielded uninterpretable splits in the
SOA or CSOA factors. Among the balanced 8-factor solutions, we selected the solution with $C_{EESI} = 2$,
which has the minimum value of $|F_{overlap} - F^*_{overlap}|$. This solution serves as the base case for further
analysis. The other 8-factor solutions exhibit time series and profiles that are similar to the selected
solutions. Therefore, we simply select the 8-factor solution with minimum $|F_{overlap} - F^*_{overlap}|$.
For the winter dataset, solutions with 12 or more factors are similar to the summer in which balanced
solutions (i.e., $\beta < 0.005$) are clustered narrowly around a single value of $C_{EESI}$ (in this case 0.05), as
shown in the right black box in Fig. 4b. However, in addition, solutions with 10 to 11 factors show



balanced solutions over a relatively broad range, $C_{EESI}$ = 0.001 to 0.01, as shown in the left black box
in Fig. 4b. This complex behaviour highlights the importance of fully exploring the two-dimensional
space. Solutions from the left black box (e.g., a 10-factor solution with $C_{EESI}$ = 0.01, and 11-factor
solutions with $C_{EESI}$ = 0.001, 0.005, and 0.001 which are shown in Fig. S14 to Fig S17) exhibited mixed
factors, in which biomass burning was not clearly separable from other sources. In contrast, the 12-
factor solution (see Fig. S18) and 13-factor solution (see Fig. S19) in the narrow band successfully
resolves these factors. The 12-factor and 13-factor solutions differ in that the 13-factor solution includes
uninterpretable splitting of biomass-burning-related factors. Similarly, higher-order solutions also result
in uninterpretable factor splitting. Therefore, the 12-factor solution with $C_{EESI}$ of 0.05 is selected as the
best representation of the combined dataset.

a)

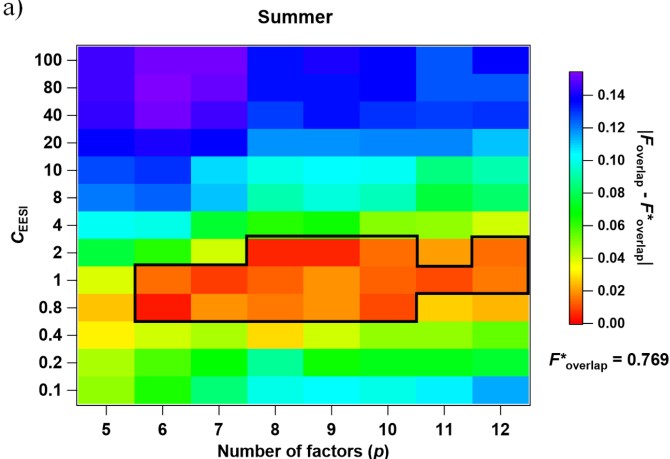

b)

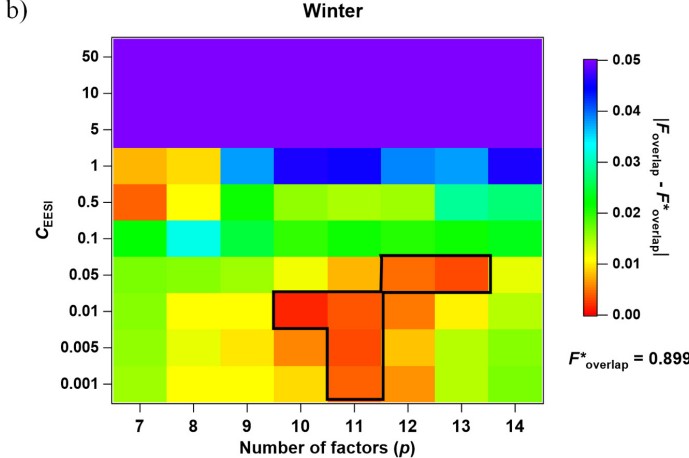

**Figure 4.** Identification of balanced solutions in the combined dataset (i.e., $| F_{overlap} - F^*_{overlap} |$ as a
function of $C_{EESI}$ and $p$) for summer (a) and winter (b) datasets. Note that $| F_{overlap} - F^*_{overlap} | = 0$ defines
a balanced solution. Solutions within the black box satisfied the $| F_{overlap} - F^*_{overlap} | < \beta$ criterion
defined in Eq. (15) ($\beta$ is set to be 0.02 and 0.005 for summer and winter, respectively) and were selected
as base case candidates, from which the base case that can best represent the combined data was selected
by manual inspection.





### 3.1.4   Acceptance criteria and factor-specific $a$-value boundaries

As discussed in Sect. 2.3.4, the combined bootstrap/$a$-value randomisation analysis requires (1) a set of criteria for solution acceptance/rejection and (2) factor-specific boundaries for randomised $a$-value selection to maintain computational efficiency. The final set of acceptance criteria and $a$-value boundaries are presented in Table 2. Here we discuss their selection, which is determined synergistically by consideration of 1) unique correlations of factor time series with the base case (see Sect. 2.3.4), 2) factor-based acceptance criteria, which are here based on selected key mass spectral features (see Sect. 3.2.1 and Sect. 3.2.2 for a complete discussion of factor characteristics). Both (1) and (2) are evaluated as a function of changing $a$-values within the multi-2D scanning algorithm (see Sect. 2.3.4). For assessing the solution/base case correlations, we utilise a confidence level of 0, meaning that the only requirement is the ability to construct a correlation matrix with the values on the diagonal being higher than any vertical or horizontal transect. This accepts the largest possible number of solutions while requiring an unambiguous relationship between base case and bootstrapped factors. Recall that the multi-2D algorithm consists of two-dimensional $a$-value scans in which the $a$-values of constrained factors are scanned from 0 to 1 with a step size of 0.1, the $a$-values of other constrained factors are set to zero, and the remaining factors are left free.

Here we describe the general steps to determine acceptance criteria and $a$-value boundaries. A factor-based acceptance criterion is defined by the combination of a diagnostic quantity relating to one or more factors and a corresponding acceptance/rejection threshold ($\theta$). Solutions that fulfil all criteria simultaneously are classified as accepted solutions. We calculate the acceptance probability as a function of $a$-value for a given factor (this is calculated independently for each factor). For a given factor, the acceptance probability is defined as the ratio of the number of accepted solutions to the total number of solutions, for which the factor has the selected $a$-value and the $a$-value of at most one other constrained factor is non-zero (that is, we consider only multi-2D runs where the factor in question is being scanned against a single other factor, while discarding runs for which the factor in question is fixed at $a$=0 while two other factors are scanned; this is relevant only for analyses with at least 3 constrained factors). The acceptance probability is not only a function of the $a$-value of the target constraint but also a function of the threshold $\theta$. When an appropriate value of $\theta$ cannot be defined *a priori*, it is selected via sensitivity tests. The final selection of the threshold $\theta$ and $a$-value ranges is a compromise between (1) maintaining a reasonably high acceptance probability, thereby providing sufficient statistics without an excessive number of bootstrap runs; and (2) ensuring a sufficiently broad exploration of the solution space to encompass most environmentally reasonable solutions and thus accurately assess errors. Therefore, we determine the threshold $\theta$ and $a$-value upper limit for each constrained factor at which a steep drop-off from high to low probability of acceptance occurs.

For the summer dataset, three factors are constrained: $HOA_{S,C}$, $COA_{S,C}$, and $InorgNit_{S,C}$, yielding three pairs ($C(3,2) = 3$) of two-dimensional $a$-value scans. Two factor-based diagnostic quantities with acceptance/rejection thresholds ($\theta$) were selected: 1) the ratio of $C_3H_3O^+$ to $C_3H_5O^+$ for $COA_{S,C}$ should be higher than the threshold $\theta_{COA_{S,C}}$ (Mohr et al., 2012), and 2) the ratio of $CO_2^+/(NO^+ + NO_2^+)$ for $InorgNit_{S,C}$ should not be higher than $\theta_{InorgNit_{S,C}}$, because the $CO_2^+$ signal in $InorgNit_{S,C}$ should not greatly exceed the $CO_2^+$ signal produced by reaction of nitrate on the vaporiser (Pieber et al., 2016); excessively high values would indicate mixing with OA. To explore the sensitivity of the acceptance probability to the threshold $\theta$, we varied $\theta_{COA_{S,C}}$ from 4.5 to 5.1 with a step of 0.1 (note that 5.0 is the ratio of $C_3H_3O^+ / C_3H_5O^+$ in the reference profile) and $\theta_{InorgNit_{S,C}}$ from 0.034 to 0.040 with a step of 0.01, (note that 0.0345 is the ratio of $CO_2^+/(NO^+ + NO_2^+)$ in the reference profile).



The acceptance probability as a function of $a$-value and the various thresholds ($\theta$'s) for $COA_{S,C}$,
$InorgNit_{S,C}$, and $HOA_{S,C}$ are shown in Fig. S22. Vertical dashed lines denote the final selected $a$-values,
while the thicker traces denote the selected $\theta$ values (both of which are also given in Table 2). For
$\theta_{COA_{S,C}} > 5.0$, very few runs are accepted. Within the range $4.5 \leq \theta_{COA_{S,C}} \leq 5.0$, $\theta_{COA_{S,C}}$ does not affect
the relationship between acceptance probability and $a$-value for $InorgNit_{S,C}$ (Fig. S22b), but has a
considerable effect for $COA_{S,C}$ and $HOA_{S,C}$, with a decreasing $\theta_{COA_{S,C}}$ leading to the acceptance
probability remaining high at larger $a$-values. Visual inspection of the solutions suggests that this is due
to increased mixing, mostly between $COA_{S,C}$ and $HOA_{S,C}$. Therefore, we select a value of $\theta_{COA_{S,C}} = 5.0$,
corresponding to the $C_3H_3O_3^+/C_3H_5O_3^+$ in the factor profile. For $\theta_{InorgNit_{S,C}}$, values smaller than 0.0345
(i.e., reference profile) result in a very low acceptance probability, whereas choice of $\theta_{InorgNit_{S,C}}$ results
in similar acceptance probabilities as a function of $a$-value. Therefore, we select 0.0345, as the
acceptance probability for $\theta_{InorgNit_{S,C}}$ of 0.035 is not substantially different from 0.0345.Having
selected these $\theta$ values, we set $a$-value limits at the point where an incremental increase/decrease in $a$
yields a large change in acceptance probability (i.e. transition from high probability to low probability).
For the current dataset, constrained factors, and selected $\theta$'s, there is no such transition at low $a$-values,
and we therefore select only an upper limit for the $a$-values. For $COA_{S,C}$, there is a clear decrease for
both criteria between $a_{COA_{S,C}} = 0.1$ and $a_{COA_{S,C}} = 0.2$, and we therefore set the $a$-value boundaries as 0
$\leq a_{COA_{S,C}} \leq 0.2$. $InorgNit_{S,C}$ maintains an acceptance probability of ~50 % for $a_{InorgNit_{S,C}} \leq 0.4$, before
decreasing to <20 % at $a_{InorgNit_{S,C}} = 0.5$ and ~0 for $a_{InorgNit_{S,C}} > 0.5$; therefore the range $0 \leq$
$a_{InorgNit_{S,C}} \leq 0.5$ is chosen. Finally, for $HOA_{S,C}$, the acceptance probability decreases from ~55 % at
$a_{HOA_{S,C}} \leq 0.1$ to ~35 % at $a_{HOA_{S,C}} \leq 0.2$, so the a-value range for $HOA_{S,C}$ is selected as $0 \leq a_{HOA_{S,C}} \leq$
0.2. The $a$-values selected for constraints for the further summer bootstrap analysis are summarised in
Table 2. However, we also see that for $HOA_{S,C}$ the acceptance probability increases and stays high again
for the $a$-value of 0.4 to 0.8. Therefore, we made an additional bootstrap analysis to explore the result
when the $a$-value of $HOA_{S,C}$ randomises from 0 to 0.8, as discussed in the last paragraph in this section.
In the winter dataset, four factors ($HOA_{W,C}$, $COA_{W,C}$, $CSOA_{W,C}$, and $InorgNit_{W,C}$) are constrained,
yielding six pairs ($C(4,2) = 6$) of two-dimensional $a$-value scans. Compared to the summer dataset, the
unique base case/bootstrap correlation requirement yields a much smaller number of accepted solutions,
probably due to the more complicated aerosol sources and/or evolution conditions in winter (e.g.,
multiple biomass burning-related factors). Three factor-based diagnostic quantities were selected: 1)
the fraction of the nicotine signal ($[C_{10}H_{14}N_2]H^+$) apportioned to $CSOA_{W,C}$, 2) the relative intensity of
the AMS primary biomass burning tracer $C_2H_4O_2^+$ (Alfarra et al., 2007; Cubison and Jimenez, 2015) in
the factor profiles (AMS part) of less-aged biomass burning ($LABB_{W,C}$) vs. more-aged biomass burning
($MABB_{W,C}$), and 3) the relative intensity of the EESI-TOF primary biomass burning tracer levoglucosan
($[C_6H_{10}O_5]Na^+$) (Qi et al., 2019; Stefenelli et al., 2019; Lopez-Hilfiker et al., 2019) in the factor profiles
of $LABB_{W,C}$ vs. $MABB_{W,C}$. For 2) and 3), we require that the contribution of the primary tracer is higher
for the profile of $LABB_{W,C}$ than $MABB_{W,C}$ as follows:

$$\frac{LABB_{W,C,ion} - MABB_{W,C,ion}}{(LABB_{W,C,ion} + MABB_{W,C,ion})/2} > \theta_{ion} \qquad (17)$$

where $LABB_{W,C,ion}$ and $MABB_{W,C,ion}$ are the "ion" intensity in the $LABB_{W,C}$ and $MABB_{W,C}$ factor
profiles, and "ion" in Eq. (17) denotes either AMS $C_2H_4O_2^+$ (criterion 2) or EESI-TOF levoglucosan
($[C_6H_{10}O_5]Na^+$) (criterion 3), and $\theta_{ion}$ denotes the acceptance threshold.





For criterion 1), we select the threshold $\theta_{CSOA_{W,C}}$ from investigation of Fig. S23, which shows the
frequency distribution of the fraction of total nicotine signal apportioned to $CSOA_{W,C}$, derived from the
multi-2D scans used to assess criteria 2 and 3 (see below). The figure shows that for nearly all runs, the
fraction of total nicotine mass apportioned to this factor is higher than 0.96. The exceptions are clear
outliers, and we therefore select $\theta_{CSOA_{W,C}} = 0.96$ which was therefore chosen as the criterion threshold.
The acceptance probability as a function of $a$-value is shown in Fig. S24 for $HOA_{W,C}$, $COA_{W,C}$,
$InorgNit_{W,C}$, and $CSOA_{W,C}$. For criteria 2 and 3, sensitivity tests are conducted using $\theta_{C_2H_4O_2^+}$ and
$\theta_{levoglucosan}$, which were varied from 0 to 1 with a step of 0.1, and the final selected values are shown
as a thicker line. The acceptance probability decreases to near-zero for $\theta_{C_2H_4O_2^+} \geq 0.1$ and $\theta_{levoglucosan}$
$\geq 0.2$. We select 0 for both thresholds, which is the most permissive value, requiring only that $MABB_{W,C}$
appear more aged than $LABB_{W,C}$ (i.e., reduced contribution from POA tracers). Similar to the summer
dataset, there is no major decrease in acceptance probability at low $a$-values, and we therefore impose
only an upper limit. For $HOA_{W,C}$, we set the upper $a$-value boundary at 0.9, due to the large decrease in
acceptance probability at $a_{HOA_{W,C}} = 1.0$. However, for the other constrained factors, the acceptance
probability decreases steadily without a steep drop-off. We target an acceptance probability of ~0.4 (by
considering the unmixing status) as a subjective compromise between exploration and computational
efficiency, and select as an upper boundary the largest $a$-value that achieves this. This results in upper
$a$-value limits of 0.3 for $COA_{W,C}$, and 0.5 for $InorgNit_{W,C}$. For $CSOA_{W,C}$. the high acceptance probability
is kept high from the $a$-value of 0 to 0.6. Therefore, we chose the $a$-value range of $CSOA_{W,C}$ to be 0 to
0.6. However, it is also observed that the acceptance probability for this factor dips at 0.7 and stays high
again at $a$-values of 0.8 and 0.9, so we made an addition bootstrap analysis with the $a$-value range for
$CSOA_{W,C}$ of 0 to 0.9 to explore the influence of the $a$-value of this factor on overall result, as discussed
in the following paragraph. The $a$-values selected for the four constraints for the further winter bootstrap
analysis are summarised in Table 2.
After $a$-value selection, 1000 bootstrap runs were performed for summer and winter, respectively, and
in each bootstrap run, an $a$-value was randomly selected for each constrained factor, with a step size of
0.05 for summer and 0.1 for winter within the corresponding range. The criteria for accepted solutions
in the bootstrap analysis are exactly the same as the criteria and $\theta$ in Sect. 3.1.4, and are given in Table
2. As noted above, accepted solutions must simultaneously satisfy all criteria including the time-series-
based mixing status exploration and mass-spectral-based criteria. Note that we also did an additional
bootstrap analysis for summer and winter, respectively, as mentioned in previous paragraphs, to explore
the bootstrap result with larger $a$-value range of $HOA_{S,C}$ and $CSOA_{W,C}$. In the additional bootstrap
analysis for summer, $a$-value range for $HOA_{S,C}$ was set to be $0 \leq a_{HOA_{S,C}} \leq 0.8$, while the $a$-value ranges
of the other two constraints were kept the same as indicated in Table 2. Likewise, we only changed the
$a$-value range of $CSOA_{W,C}$ to be $0 \leq a_{CSOA_{W,C}} \leq 0.9$, while keeping the $a$-value ranges of the other
three constraints the same as in Table 2. Since the results of these additional bootstrap analysis are not
qualitatively different from the bootstrap analysis with $a$-value ranges in Table 2, we only present the
bootstrap results with $a$-value ranges in Table 2.
**Table. 2** Summary of $a$-value range for constrained factors, criteria for $a$-value range and accepted
bootstrap run selection and the number of accepted runs from the final combined bootstrap/$a$-value
analysis for the summer and winter datasets.



| Dataset | Constrained factor | $a$-value range | Criteria | Accepted runs |
|---------|-------------------|-----------------|----------|---------------|
| Zurich summer | $HOA_{S,C}$ | $0 \leq a \leq 0.2$ | 1). $COA_{S,C}$: $\frac{C_3H_3O^+}{C_3H_5O^+} \geq 5$ <br> 2). $InorgNit_{S,C}$: $\frac{CO_2^+}{NO^+ + NO_2^+} \leq 0.035$ <br> 3). Base case vs. Bootstrap correlation test at confidence level = 0 | 764 (76.4 %) |
| | $COA_{S,C}$ | $0 \leq a \leq 0.2$ | | |
| | Inorganic nitrate ($InorgNit_{S,C}$) | $0 \leq a \leq 0.5$ | | |
| Zurich winter | $HOA_{W,C}$ | $0 \leq a \leq 0.9$ | 1). $CSOA_{W,C}$: $f_{mass}$(nicotine) $\geq 0.96$ <br> 2). $C_2H_4O_2^+$ intensity: $LABB_{W,C} - MABB_{W,C} > 0$ <br> 3). $C_6H_{10}O_5$ intensity: $LABB_{W,C} - MABB_{W,C} > 0$ <br> 4). Base case vs. Bootstrap correlation test at confidence level = 0 | 308 (30.8 %) |
| | $COA_{W,C}$ | $0 \leq a \leq 0.3$ | | |
| | Inorganic nitrate ($InorgNit_{W,C}$) | $0 \leq a \leq 0.5$ | | |
| | $CSOA_{W,C}$ | $0 \leq a \leq 0.6$ | | |

### 3.2 cPMF results

Here we present final results from the cPMF analysis of the summer and winter campaigns. The final
solutions are reported as the average of all accepted bootstrap/$a$-value randomisation runs (764 for
summer, 308 for winter), with uncertainties corresponding to the standard deviation. We compare the
cPMF factors to their counterparts from the standalone AMS and EESI-TOF solutions, for cases where
a clear factor-to-factor correspondence exists.
A complication in this analysis is that the $NO^+$ and $NO_2^+$ signal can result from either organic or
inorganic nitrate. Ideally, all inorganic $NO^+$ and $NO_2^+$ would apportion to the $InorgNit_{S,C}$ and
$InorgNit_{W,C}$ factors, however inspection of the solutions reveals that this is not the case, as discussed in
the factor presentations (Sect. 3.2.1 and Sect. 3.2.2). Therefore, we estimate the organic and inorganic
contributions to these ions by the method of Kiendler-Scharr et al. (2016), as follows:

$$frac_{ON,k} = \frac{(1 + R_{ON})(R_k - R_{cal})}{(1 + R_k)(R_{ON} - R_{cal})} \qquad (18)$$

Here we apply this analysis on a factor-by-factor basis, where $frac_{ON,k}$, defined in Eq. (19a), represents
the fraction of ON apportioned to the $k$th factor, and $R_k$ denotes the intensity ratio of $NO_2^+$ to $NO^+$ in
the factor profile. $R_{cal}$ is the reference $NO_2^+/NO^+$ ratio for inorganic nitrate, taken as that of the
$InorgNit_{W,C}$ and $InorgNit_{S,C}$ reference profiles for their respective datasets. $R_{ON}$, defined in Eq. (19b),
is the intensity ratio of $NO_2^+$ to $NO^+$ for organonitrate, which ranges from 0.08 to 0.20 (Boyd et al.,
2015; Bruns et al., 2010; Fry et al., 2011; Fry et al., 2009; Rollins et al., 2009).

$$frac_{ON,k} = \frac{(f_{ON,k,NO^+} + f_{ON,k,NO_2^+})}{(f_{k,NO^+} + f_{k,NO_2^+})} \qquad (19a)$$

$$R_{ON} = \frac{f_{ON,k,NO_2^+}}{f_{ON,k,NO^+}} \qquad (19b)$$

28 Here $f_{k,NO^+}$ and $f_{k,NO_2^+}$ denote the total $NO^+$ and $NO_2^+$ signal, respectively in the $k$th factor profile,

29 while $f_{ON,k,NO^+}$ and $f_{ON,k,NO_2^+}$ denote the organonitrate contribution to these ions. Because $f_{k,NO^+}$ and

30 $f_{k,NO_2^+}$ are directly available from the factor profile, $frac_{ON,k}$ is independently calculated via Eq. (18),





and $R_{ON}$ is assumed, Eqs. (19a) and (19b) constitute a system of 2 equations with 2 unknowns, which
can be solved algebraically for $f_{ON,k,NO^+}$ and $f_{ON,k,NO_2^+}$, yielding:

$$f_{ON,k,NO^+} = \frac{(R_k - R_{cal})(f_{k,NO^+} + f_{k,NO_2^+})}{(1 + R_k)(R_{ON} - R_{cal})} \quad (20a)$$

$$f_{ON,k,NO_2^+} = \frac{(R_k - R_{cal})(f_{k,NO^+} + f_{k,NO_2^+})}{(1 + R_k)(R_{ON} - R_{cal})} \cdot R_{ON} \quad (20b)$$

These calculations are important not only for profile interpretation, but also for quantitative
apportionment of OA. Specifically, as noted earlier, calculations of the OA contribution to the factor
time series, $(g_{i,k})_{AMS}$, and the EESI-TOF sensitivity to a given factor, $AS_k$, should consider only the
organic contribution to $NO^+$ and $NO_2^+$. In this study, we estimated the contribution from organonitrates
for all factors in summer and winter assuming the midpoint of the $R_{ON}$ range ($R_{ON} = 0.14$).
Organonitrate contributions ($frac_{ON,k}$) to the total nitrate signal for each factor and the corresponding
OA fraction $\sum_j (f_{k,j})_{AMS}$ are shown in Table S1. We also include the same calculations performed
assuming an $R_{ON}$ of 0.08 or 0.20, which as discussed above consitute the lower and upper estimates
from previous studies. For $R_{ON} = 0.14$, the $frac_{ON,k}$ for all SOAs in summer are higher than 75 %, and
for winter, this fraction $frac_{ON,k}$ varies by factor from 0 to 100 %, with four factors having $frac_{ON,k}$
= 100 % (SOA1$_{W,C}$, MABB$_{W,C}$, LABB$_{W,C}$ and NitOA1$_{W,C}$), suggesting the $NO^+$ and $NO_2^+$ signals are
strongly influenced by ON. If $R_{ON} = 0.08$ is assumed, the estimated $frac_{ON,k}$ decreases by ~12 % for
the summer SOA factors and by 10 % to 20 % for the winter SOA factors, whereas assuming $R_{ON} =$
0.20 increases $frac_{ON,k}$ by ~15 % in the summer and 16% in the winter OA factors. The effect of this
assumption on the factor OA concentration and thus $AS_k$ is much smaller, with all factors below ±2 %
except for one wintertime SOA factor (SOA1$_{W,C}$, ±6 %).
22 .
**3.2.1    cPMF analysis: Zurich summer**
Eight factors were resolved from the Zurich summer campaign: HOA$_{S,C}$, COA$_{S,C}$, CSOA$_{S,C}$, InorgNit$_{S,C}$,
two daytime SOA factors (DaySOA1$_{S,C}$ and DaySOA2$_{S,C}$), and two nighttime SOA factors
(NightSOA1$_{S,C}$ and NightSOA2$_{S,C}$). The mean time series, diurnal cycle and the mass spectra of these
factors over 764 accepted runs are shown in Fig. 5, together with the time series from AMS-only PMF
and/or EESI-TOF-only PMF when the corresponding standalone factor(s) exist. An estimate of
campaign-average percent uncertainty in the mass concentration of each factor, calculated as the median
of the standard deviation across all accepted runs, is given in Table S2. Many factor characteristics from
cPMF resemble those previously discussed in detail for single-instrument AMS PMF and/or EESI-TOF
PMF (Stefenelli et al., 2019). Therefore, only a summary discussion of these characteristics are
presented here, and we focus on new information and/or differences obtained by the cPMF analysis.
Recall that factor profiles for HOA$_{S,C}$, COA$_{S,C}$, and InorgNit$_{S,C}$ are constrained as discussed above.
**HOA$_{S,C}$** --- The AMS mass spectrum is dominated by the $C_nH_{2n+1}^+$, and $C_nH_{2n-1}^+$ series, consistent with
*n*-alkanes and branched alkanes (Ng et al., 2011a; Ulbrich et al., 2009; Lanz et al., 2007; Zhang et al.,
2005; Qi et al., 2019; Stefenelli et al., 2019). The diurnal cycle of HOA$_{S,C}$ has three clear peaks (see
Fig. 5b), however, compared to HOA$_{S,A}$ from Stefenelli et al. (2019), their intensities are weaker.
Specifically, the morning peak intensity ratio to the evening peak intensity is almost 1 in the HOA$_{S,A}$
factor, whereas in HOA$_{S,C}$, the morning peak is ~1/3 of the evening peak. In terms of contribution to
total OA, the HOA$_{S,A}$ factor contributes 5.8 % (0.177 µg m$^{-3}$) of the total OA, whereas in the cPMF
analysis, this factor only contributes 3.1 % (0.092 µg m$^{-3}$) of the total OA.





**COA$_{S,C}$** --- This factor is characterised by long-chain fatty acids and alcohols, e.g., coronaric acid and/or its isomers at *m/z* 319.2 ([C$_{18}$H$_{32}$O$_3$]Na$^+$), oleic acid and/or its isomers at *m/z* 305.2 ([C$_{18}$H$_{34}$O$_2$]Na$^+$), and 2-oxo-tetredecanoic acid and/or its isomers at *m/z* 293.2 ([C$_{16}$H$_{30}$O$_3$]Na$^+$). Similar to previous work, the AMS profile shows both alkyl fragments and slightly oxygenated ions, consistent with aliphatic acids from cooking oils (Hu et al., 2016). The AMS profile is characterised by a high ratio of C$_3$H$_3$O$^+$ to C$_3$H$_5$O$^+$ (~5 here), slightly higher than in other studies (Xu et al., 2019; Zhao et al., 2019; Sun et al., 2016a; Sun et al., 2016b), as well as high contributions from C$_5$H$_8$O$^+$, C$_6$H$_{10}$O$^+$ and C$_7$H$_{12}$O$^+$. Both cPMF and single instrument PMF analyses yield peaks during lunch and dinner. The time series of COA$_{S,C}$ is strongly correlated with those of the single instrument solutions, with Pearson's $r^2$ of 0.846 and 0.634 against COA$_{S,A}$ and COA$_{S,E}$, respectively.

**CSOA$_{S,C}$** --- The EESI-TOF factor profile is dominated by nicotine (detected as [C$_{10}$H$_{13}$N$_2$]H$^+$) at *m/z* 163.12 and levoglucosan at *m/z* 185.042 ([C$_6$H$_{10}$O$_5$]Na$^+$), which derives from pyrolysis of the cellulose present in tobacco (Talhout et al., 2006). In the AMS profile, this factor accounts for 79.3 % of the signal from C$_5$H$_{10}$N$^+$ at *m/z* 84.081, which is attributed to a fragment of n-methyl pyrrolidine and previously identified as a tracer for cigarette smoke (Struckmeier et al., 2016). The time series of CSOA$_{S,C}$ correlates with that of the AMS-only and EESI-TOF solutions, with $r^2$ of 0.922 and 0.965, respectively. The diurnal cycles from the combined and single-instrument solutions are likewise correlated, showing high concentrations at night and low concentration during daytime.

**InorgNit$_{S,C}$** --- Among the accepted bootstrap runs, the mean CO$_2$$^+$/(NO$^+$+NO$_2$$^+$) ratio is 0.0346, slightly higher than the ratio of 0.0345 observed during the NH$_4$NO$_3$ calibration period, probably due to 1) uncertainties in the constrained profile, and/or 2) a small amount of OA apportioned to this factor. The time series of this factor correlates with AMS nitrate (NO$_3$$^-$), NO$^+$ and NO$_2$$^+$ time series, with $r^2$ of 0.654, 0.645 and 0.956, respectively. Regarding the mass fraction, approximately 48.5 % of the NO$^+$ signal and 78.0 % of the NO$_2$$^+$ signal are apportioned to this factor, followed by the two NightSOA$_{S,C}$ factors. This is consistent with the overall NO$^+$ and NO$_2$$^+$ signals deriving not only from inorganic nitrate, but also from organonitrates (in other factors).

**DaySOA1$_{S,C}$ and DaySOA2$_{S,C}$** --- The cPMF analysis yields two SOA factors elevated during daytime, denoted DaySOA1$_{S,C}$ and DaySOA2$_{S,C}$. The EESI-TOF spectra are similar to two factors retrieved from EESI-TOF-only PMF analysis by Stefenelli et al. (2019), but were not resolved in AMS-only PMF, where only more- and less-oxygenated SOA factors (MO-OOA$_{S,A}$ and LO-OOA$_{S,A}$) were obtained. These factors contain strong signatures from terpene oxidation products, e.g., monoterpene-derived ions (C$_{10}$H$_{16}$O$_x$, x=5, 6, 7) and sesquiterpene oxidation products (C$_{15}$H$_{24}$O$_x$, x=3, 4, 5). A detailed comparison of the two DaySOA factors from the cPMF analysis to the LO-OOA$_{S,A}$ and MO-OOA$_{S,A}$ factors from AMS-only PMF is shown in Fig. S28, and a comparison between the two DaySOA$_{S,C}$ factors and DaySOA$_{S,E}$ factors are shown in Figs. S29 a) and b), respectively. The AMS ions in these two factors are characterised by a strong CO$_2$$^+$ signal, similar to the LO-OOA$_{S,A}$ and MO-OOA$_{S,A}$ factors, indicating they largely consist of oxygenated OA, consistent with the EESI-TOF spectra. We calculate $frac_{ON}$ for DaySOA1$_{S,C}$ and DaySOA2$_{S,C}$ to be 0.869 and 1.000, respectively, demonstrating that the NO$^+$ and NO$_2$$^+$ signal in these factors is dominated by organonitrates. Regarding the time series, DaySOA1$_{S,C}$ and DaySOA2$_{S,C}$ correlate strongly with DaySOA1$_{S,E}$ and DaySOA2$_{S,E}$, with $r^2$ of 0.883 and 0.977, respectively. The diurnal patterns of DaySOA1$_{S,C}$ and DaySOA2$_{S,C}$ are consistent with the diurnal patterns of DaySOA1$_{S,E}$ and DaySOA2$_{S,E}$. The diurnal patterns of both factors show an enhancement in the afternoon and the evening, which distinguish these SOAs from other SOAs: DaySOA1$_{S,C}$ exhibits



almost a factor of 2 enhancement in signal between 15:00 and 21:00 compared to the morning, whereas
the DaySOA2$_{S,C}$ exhibits the same magnitude of enhancement in signal around 12:00 to 17:00.
**NightSOA1$_{S,C}$ and NightSOA2$_{S,C}$** --- We retrieve two SOA factors that are enhanced overnight and in
the early morning, denoted NightSOA1$_{S,C}$ and NightSOA2$_{S,C}$. Their factor profiles and time
series/diurnals closely resemble those of NightSOA1$_{S,E}$ and NightSOA2$_{S,E}$ (see Figs. S29c and S29d).
Similar to the DaySOA$_{S,C}$ factors, terpene oxidation products are evident. However, the composition is
weighted towards less oxygenated and more volatile terpene oxidation products, e.g., $C_{10}H_{16}O_2$ and
$C_{10}H_{16}O_3$, which likely partition to the particle phase at night when temperature decreases. In addition,
signals consistent with monoterpene-derived organonitrates are also evident, e.g., the $C_{10}H_{17}O_{6-8}N$ and
$C_{10}H_{15}O_{6-9}N$ series, which are consistent with night time oxidation of monoterpenes by $NO_3$ radicals
(Faxon et al., 2018; Zhang et al., 2018; Xu et al., 2015). The AMS ions in these two factors are
characterised by a strong $CO_2^+$ signal and also a relatively high $NO^+$ signal compared to
sum_DaySOAs$_{S,C}$. The ratio of $NO^+$/ $NO_2^+$ ratio is 4.55 and 8.24 for NightSOA1$_{S,C}$ and NightSOA2$_{S,C}$,
respectively, yielding $frac_{ON}$ for NightSOA1$_{S,C}$ and NightSOA2$_{S,C}$ of 0.798 and 1, indicating high
organonitrate content. These two factors correlate well with sum_NightSOAs$_{S,E}$, reaching $r^2$ of 0.975
and 0.897, following in general the same diurnal patterns, with NightSOA1$_{S,C}$ peaking from 22:00 to
05:00 and NightSOA1$_{S,C}$ peaking from 04:00 to 12:00.

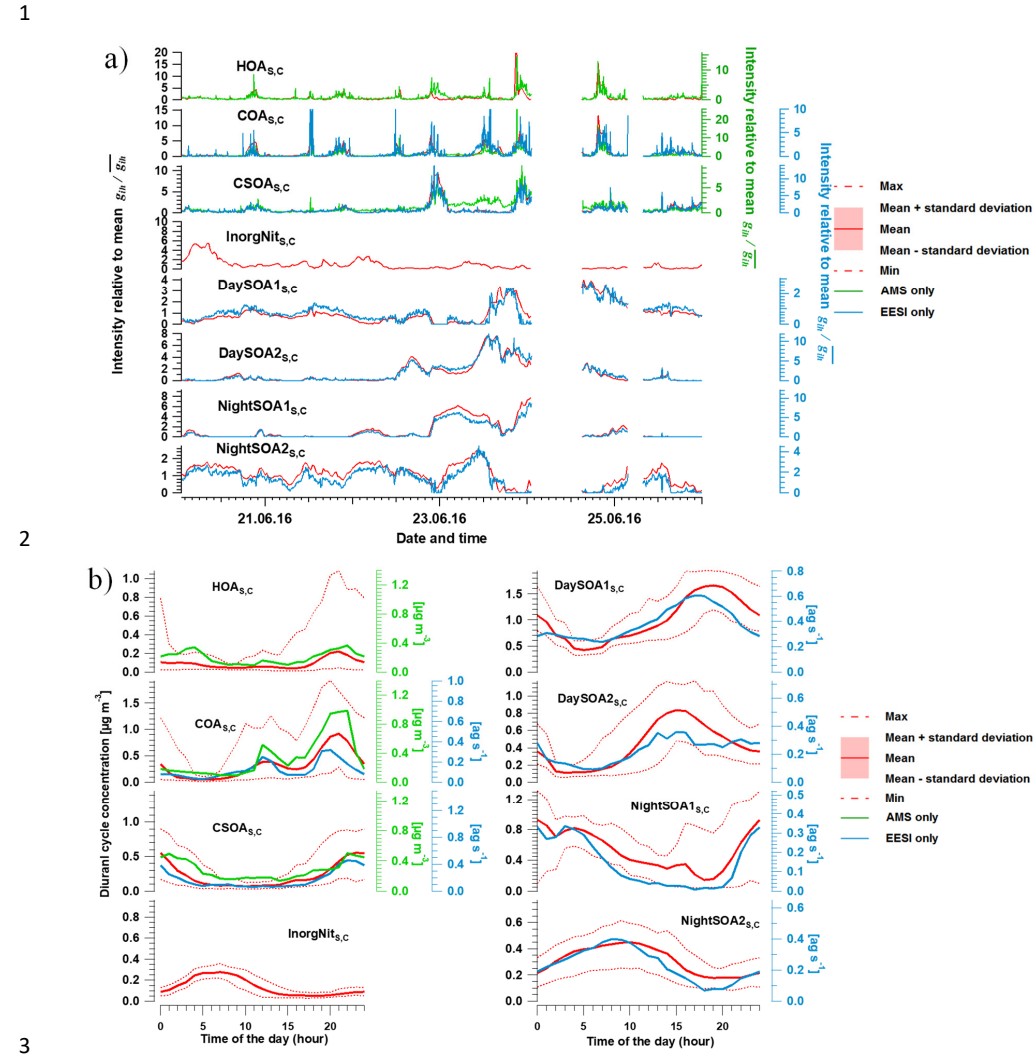



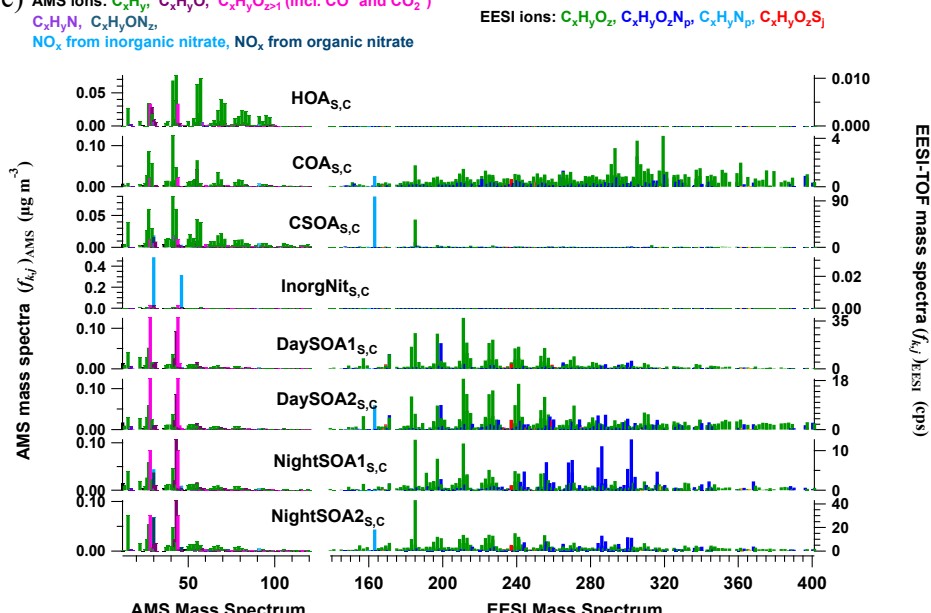

Figure 5. Mean factor time series (a), diurnal cycles (b) and factor profiles (c) from the 764 accepted
bootstrap runs from cPMF analysis. In a), the average factor time series are shown in red, and
corresponding AMS and/or EESI-TOF factors from standalone PMF are shown in green and blue,
respectively. Shaded areas represent the standard deviation across all accepted runs and are summarised
in Table S2. In b), the average diurnal cycles are displayed as red solid lines. Shaded areas denote the
standard deviation over the average diurnal from individual solutions over all 764 accepted runs.
Dashed lines denote the maximum and minimum mean diurnal observed within these 764 runs. For
comparison, the AMS and EESI-TOF PMF factor time series and diurnal cycles from the individual
dataset in Stefenelli et al. (2019) are shown in green and blue respectively for related factors. In c), the
average factor profiles are coloured by different ion families. Here, the AMS factor profiles are in the
unit of $\mu g\ m^{-3}$ (each factor sums to 1 $\mu g\ m^{-3}$), whereas the EESI-TOF spectra are in the unit of cps (each
factor sums to the total signal derived from 1 $\mu g\ m^{-3}$ of the factor). Note that the $NO^+$ and $NO_2^+$ signal
is divided into inorganic and organic contributions.
### 3.2.2   cPMF analysis: Zurich winter
Twelve factors were resolved from cPMF analysis of the Zurich winter campaign: $HOA_{W,C}$, $COA_{W,C}$,
$InorgNit_{W,C}$, $CSOA_{W,C}$, $SOA1_{W,C}$, $SOA2_{W,C}$, a more-aged biomass burning OA ($MABB_{W,C}$), two less-
aged biomass burning OAs ($LABB1_{W,C}$ and $LABB2_{W,C}$), two nitrogen-containing OA factors
($NitOA1_{W,C}$ and $NitOA2_{W,C}$), and a factor related to a specific local event ($EVENT_{W,C}$). Because no
significant chemical differences are apparent between $LABB1_{W,C}$ and $LABB2_{W,C}$ (see Figs. S30 and
S31), they are aggregated to a single $LABB_{W,C}$ factor for presentation. Therefore, there are 11 factors
presented below. The average time series and mass spectra of these factors among 308 accepted runs
are shown in Fig. 6. The factor profiles for $HOA_{W,C}$, $COA_{W,C}$, $InorgNit_{W,C}$, and $CSOA_{W,C}$ are constrained
as described previously. Similar to the summer dataset, uncertainties in the factor mass concentrations
are summarised in Table S2.





**HOA$_{W,C}$** --- This factor has a qualitatively a profile similar to the summer campaign, and the discussion
of the HOA$_{S,C}$ profile applies here as well. The HOA$_{W,C}$ time series correlates strongly with HOA$_{W,A}$ ($r^2$
of 0.913).
**COA$_{W,C}$** --- The COA$_{W,C}$ profile is characterised by long-chain fatty acids and alcohols e.g., coronaric
acid and/or its isomers at $m/z$ 319.2 ($[C_{18}H_{32}O_3]Na^+$), oleic acid and/or its isomers at $m/z$ 305.2
($[C_{18}H_{34}O_2]Na^+$), and 2-oxo-tetredecanoic acid and/or its isomers at $m/z$ 293.2 ($[C_{16}H_{30}O_3]Na^+$), and in
the AMS, a combination of alkyl fragments and slightly oxygenated ions from aliphatic acids from
cooking oils, including $C_5H_8O^+$, $C_6H_{10}O^+$ and $C_7H_{12}O^+$. These features are consistent with features
found by other studies (Qi et al., 2019; Stefenelli et al., 2019; Tong et al., 2021). The COA$_{W,C}$ time
series correlates with the corresponding single instrument analyses, exhibiting $r^2$ of 0.894, and 0.798,
with COA$_{W,A}$ and COA$_{W,E}$, respectively.
**InorgNit$_{W,C}$** --- As noted in Sect. 3.1.2, the NO$^+$/ NO$_2^+$ ratio of this factor (2.42) is higher than that of
pure NH$_4$NO$_3$ measured onsite (1.58), consistent with the presence of other inorganic nitrate sources
such as KNO$_3$. Also, the mean $CO_2^+/(NO^++NO_2^+)$ ratio is 0.0371, higher than the ratio of 0.0261 from
the constructed InorgNit$_{W,C}$ profile, probably due to 1) uncertainties in the constrained profile, and/or
2) a small amount of OA apportioned to this factor. The time series of this factor shows high correlations
with the AMS nitrate (NO$_3^-$), NO$^+$ and NO$_2^+$ time series, with $r^2$ of 0.739, 0.792 and 0.754, respectively.
Regarding the mass fraction, only 13.7% of the NO$^+$ signal and 13.2 % of the NO$_2^+$ signal are
apportioned to this factor. The considerable fractions of the NO$^+$ and NO$_2^+$ signal from inorganic nitrate
and organonitrates in other factors are estimated as discussed above (Kiendler-Scharr et al., 2016) and
will be interpreted later for the relevant factors (as summarised in Table S1).
**CSOA$_{W,C}$** --- Similar to CSOA$_{S,C}$, nicotine at $m/z$ 163.12 and levoglucosan at $m/z$ 185.042 were found
to be the two highest peaks in the EESI-TOF mass spectra, contributing 8.75 % and 4.56 % of the EESI-
TOF signal. The time series of this factor resolved from cPMF analysis correlates with CSOA$_{W,E}$ ( $r^2 =$
0.662). Similar to CSOA$_{W,C}$, the fragment of cigarette smoke tracer n-methyl pyrrolidine $C_5H_{10}N^+$ at
$m/z$ 84.081 is also found here. This is a minor factor, comprising 2.4 % of OA.
**SOA1$_{W,C}$ and SOA2$_{W,C}$** --- these two factors have qualitatively similar spectra but different temporal
patterns. SOA1$_{W,C}$ decreased gradually from 26 to 30 January, whereas SOA2$_{W,C}$ increased from 26
January and fluctuated at high level from 28 to 31 January and then decreased from 1 February on.
From the AMS perspective, both factors are characterised by high NO$^+$, NO$_2^+$ and CO$_2^+$ signal compared
to other organic ions. Organonitrates account for all NO$^+$ and NO$_2^+$ signals in SOA1$_{W,C}$, but contribute
nothing in SOA2$_{W,C}$. Aside from the NO$^+$ and NO$_2^+$ ions, these AMS spectra are similar to the profiles
of MO-OOA$_{W,A}$ and LO-OOA$_{W,A}$ which are characterised by high CO$_2^+$ signal. Major ions in the EESI-
TOF profile include $C_{10}H_{16}O_x$ (x = 3, 4, 5), $C_9H_{14}O_x$ (x = 3, 4), $C_8H_{12}O_x$ (x = 4, 5), $C_{10}H_{18}O_4$, and
$C_{10}H_{14}O_5$, which are also found in secondary biomass burning (three MABB$_{W,E}$ factors) and/or terpene
oxidation factors (SOA1$_{W,E}$ and SOA2$_{W,E}$ ) from Qi et al. (2019). However, the H:C ratio of these two
factors from the EESI-TOF component (1.578 and 1.588 for SOA1$_{W,C}$ and SOA2$_{W,C}$, respectively) is
less than that of DaySOA1$_{S,C}$ (1.650) and DaySOA2$_{S,C}$ (1.672), suggesting an increased contribution
from aromatic precursors.
**Biomass burning factors (LABB$_{W,C}$ and MABB$_{W,C}$)** --- We resolve a less-aged biomass burning
factor (LABB$_{W,C}$, which, as mentioned above, is the aggregate of two similar LABB factors), and a
more-aged biomass burning factor (MABB$_{W,C}$). Consistent with Qi et al. (2019), the EESI-TOF
component of LABB$_{W,C}$ is characterised by a large signal from $[C_6H_{10}O_5]Na^+$ (mainly levoglucosan)



(20.4 %), and MABB$_{W,C}$ by a smaller but notably non-zero one (6.21 %). In addition, 76.7 % and 11.9 %
of the total levoglucosan signal is apportioned to LABB$_{W,C}$, and MABB$_{W,C}$, respectively. The difference
in the fraction of total levoglucosan apportioned to these two factors suggests different degrees of ageing
of biomass burning-emitted OA. The AMS spectrum of the BBOA$_{W,A}$ factor is characterised by
$C_2H_4O_2^+$ and $C_3H_5O_2^+$, which are typical fragments of anhydrosugars, such as levoglucosan (Alfarra et
al., 2007; Lanz et al., 2007; Sun et al., 2011). These ions are also present in LABB$_{W,C}$ and MABB$_{W,C}$
and are higher in LABB$_{W,C}$ (1.91 % vs 0.879 % for $C_2H_4O_2^+$ and 0.978 % vs 0.323 % for $C_3H_5O_2^+$). In
addition, the ratio of $C_2H_4O_2^+$ to $CO_2^+$ is 0.396 and 0.092 for LABB$_{W,C}$ and MABB$_{W,C}$, respectively,
supporting the separation of these factors based on different degrees of ageing.
**EVENT$_{W,C}$** --- This factor is low throughout the campaign except for the nights of 28 and 29 January
from 00.00 to 07.00 UTC+2, where large peaks are observed. Therefore, it likely corresponds to a
specific event near the sampling location. The mass spectrum features ions at $m/z$ 174.08, 185.04 and
195.06, tentatively assigned to $[C_8H_{11}N_2O]Na^+$, $[C_6H_{10}O_5]Na^+$ and $[C_8H_{12}O_4]Na^+$ from the EESI-TOF
part and at $m/z$ 15.024 ($CH_3^+$), 27.027 ($C_2H_3^+$), 31.018 ($CH_3O^+$), and 43.018 ($C_2H_3O^+$) from the AMS
part. Qi et al. (2019) observed a very similar factor in standalone EESI-TOF PMF, which was tentatively
attributed to the Zurich gaming festival and/or plastic burning in a nearby restaurant. The factor includes
large contributions from $C_8H_{12}O_4$, which likely represents 1,2-cyclohexane dicarboxylic acid diisononyl
ester, a plasticiser for the manufacture of food packaging. In the AMS spectrum, large signals from $NO^+$
(7.36%) and $NO_2^+$ (2.03 %) are also observed, with 46.6 % of the $NO^+$ signal and 23.6% of the $NO_2^+$
signal assigned to organonitrates. Similar to Qi et al. (2019), the AMS spectrum is also dominated by
the ions in the $C_xH_yO_z^+$ group.
**NitOA1$_{W,C}$**--- this factor is characterised by a high signal of $C_5H_{10}N^+$ at $m/z$ 84.081, contributing 4.02 %
to the AMS intensity in this factor  (no other factor exceeds 0.16 %) while 97.0 % of the $C_5H_{10}N^+$ mass
is apportioned to this factor. This ion is considered to be a tracer of cigarette smoking (Struckmeier et
al., 2016), however, different from typical CSOA mass spectra, this factor also has high signal from
$CO_2^+$, suggesting a contribution from secondary formation processes. Similar to other OA factors, this
factor also has a considerable fraction of $NO^+$ and $NO_2^+$ signal, attributed entirely to organonitrates. For
the EESI-TOF component, this factor is characterised by $[C_8H_{11}N_2O]Na^+$, levoglucosan and
$[C_8H_{11}N_2O]Na^+$, $[C_6H_{10}O_5]Na^+$ and $[C_9H_{12}O_4]Na^+$ and $[C_{11}H_{14}O_4]Na^+$, suggesting this factor may also
be influenced by fresh biomass burning.
**NitOA2$_{W,C}$** --- this factor is characterised by a high fraction of total signal from the CHON group in the
EESI-TOF analysis (38.5 %). Among these ions, $[C_7H_{11}O_6N]Na^+$ at $m/z$ 228.048, $[C_{10}H_{15}O_6N]Na^+$ at
$m/z$ 268.079, and $[C_{10}H_{17}O_7N]Na^+$ at $m/z$ 286.090 are the three highest ions, contributing 1.65 %, 1,99 %,
and 1.98 %, respectively. There are also some typical ions with high intensity from biomass burning
ageing (Qi et al., 2019; Stefenelli et al., 2019), e.g., $[C_9H_{14}O_4]Na^+$ at $m/z$ 209.078, $[C_{10}H_{14}O_6]Na^+$ at $m/z$
253.068, and $[C_{10}H_{16}O_6]Na^+$ at $m/z$ 255.084, contributing 6.47 %, 2.85 %, and 4.39 %, respectively.
This may suggest a contribution from biomass burning activities. From the AMS perspective, this factor
is characterised by high $NO^+$ and $NO_2^+$ signal, in which all of the $NO^+$ and $NO_2^+$ signals are produced
from inorganic nitrates (see Table S1), with the other ions being qualitatively similar to OOA-type
spectra.



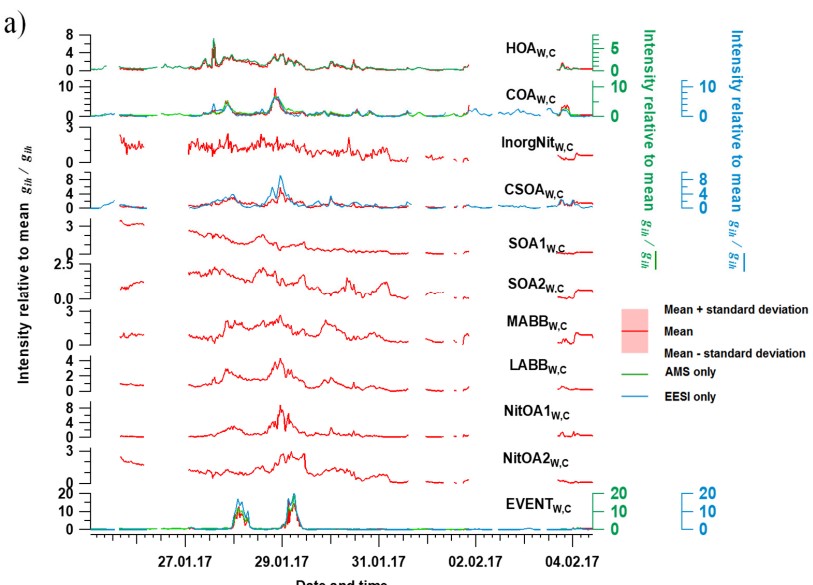

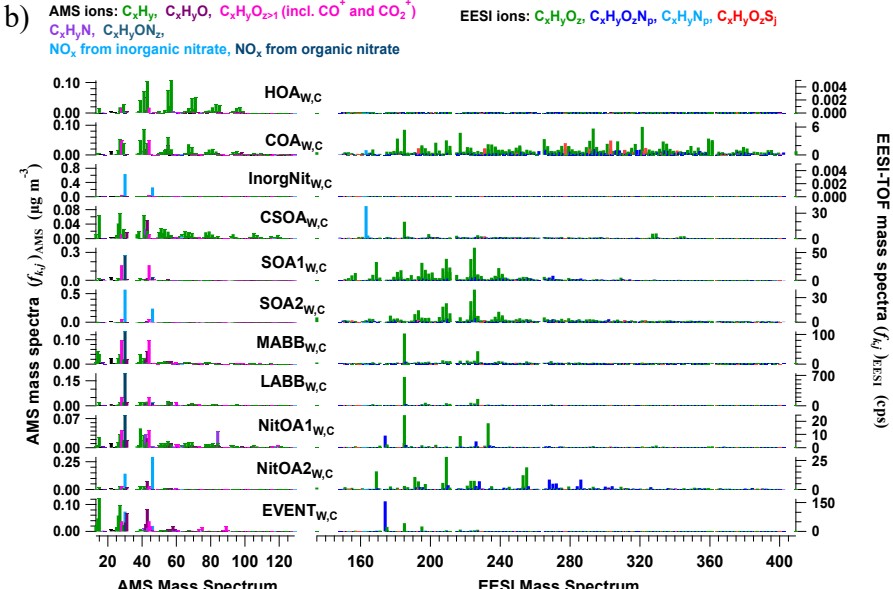

Figure 6. Average factor time series (a) and factor profiles (b), which are calculated as the mean of all accepted bootstrap runs (308 runs in total). In a), the average factor time series are shown in red, and corresponding AMS and/or EESI-TOF factors from standalone PMF are shown in green and blue, respectively. Shaded areas represent the standard deviation across all accepted runs, and are summarised in Table S2. In b), the average factor profiles are coloured by different ion families. Here, the AMS factor profiles are in the unit of $\mu g\ m^{-3}$ (each factor sums to 1 $\mu g\ m^{-3}$), whereas the EESI-TOF spectra are in the unit of cps (each factor sums to total signal derived from 1 $\mu g\ m^{-3}$ of the factor). Note that the $NO^+$ and $NO_2^+$ signal is divided into inorganic and organic contributions.





### 3.3 EESI-TOF sensitivity to resolved factors
AMS and EESI-TOF contributions to the factor profiles are intrinsically linked by cPMF. That is, for
each individual factor the two instrument profiles by definition describe the same OA fraction.
Therefore, the EESI-TOF sensitivity to a factor $AS_k$ can be calculated according to Eq. (10). Note that
this calculation depends on the assumptions that (1) both instruments are well-represented in the
solution; (2) the PMF solution is of high quality (i.e., factors are all meaningful and well-separated,
without significant mixing or splitting); (3) solution uncertainties are not so high as to preclude
quantitative interpretation of the results. Assumption (1) was discussed earlier in the context of
instrument weighting, and assumption (2) is supported by the interpretability of the factors as presented
in the previous section. By performing the cPMF analysis on a large number of runs combining
bootstrap analysis and $a$-value exploration, we can estimate uncertainties in the calculated sensitivities
imposed by the analysis model, as presented below, thereby addressing assumptions (2) and (3).
The datasets analysed here were taken from the first field deployments of the EESI-TOF. As a result,
operational protocols were not yet fully standardised across campaigns. Specifically, we lack reliable
on-site calibration with a chemical standard common to the two campaigns (this was attempted but the
measurements were evaluated to be unreliable during post-analysis due to operational problems).
Therefore, to enable comparison of relative factor sensitivities between the summer and winter
campaigns, we select COA as a reference. That is, we assume $AS_{COA} = AS_{COA_{S,C}} = AS_{COA_{W,C}}$ .We
choose COA because it is the only factor that both (1) appears in all four single-instrument datasets (i.e.,
summer and winter, AMS and EESI-TOF) and (2) compared to other factors, is less likely to
significantly change in composition between the campaigns (in contrast to, e.g., SOA in Zurich, which
is known to have significantly different precursors in summer and winter). Therefore, all sensitivities
below are reported as $(AS_k/\overline{AS_{COA}})$, in which $AS_k$ is calculated in every bootstrap run, and then
referenced to $\overline{AS}_{COA}$ (the mean $AS_{COA}$ calculated over all bootstrap runs). Here $k$ denotes a given factor
from the (summer or winter) cPMF solutions. Note that EESI-TOF sensitivities to HOA and InorgNit
are not discussed here, since they are undetectable by EESI-TOF (as configured for these campaigns;
see Sect. 2.2.2) and therefore constrained to be ~0.01 cps / (ug m$^{-3}$). The mean and standard deviation
of factor-dependent $AS_k/\overline{AS_{COA}}$ for the summer and winter datasets are shown in Fig. 7, with
histograms summarising all accepted runs shown in Fig. S32 and Fig. S33.
For ease of viewing, the factors in Fig. 7 are collected into related groups. We also calculate the $AS_k$'s
for several factor aggregations. First, five factors that are likely related to biomass burning (LABB$_{W,C}$,
MABB$_{W,C}$, NitOA1$_{W,C}$, NitOA2$_{W,C}$ and EVENT$_{W,C}$), are denoted as the "Sum_BB" factor. Additionally,
we separately aggregate the two DaySOA$_{S,C}$ and two NightSOA$_{S,C}$ factors, denoted "sum_DaySOAs$_{S,C}$"
and "sum_NightSOAs$_{S,C}$", respectively. As seen in Fig. 7 (as well as Fig. S34 and Table S3), the relative
uncertainty from the summer factors is systematically lower than for the winter factors within the
accepted solutions. This may indicate higher source apportionment quality and solution stability for the
former, but is also related to the sub-division of factors related to primary biomass burning-related
factors, as discussed later.
For COA$_{S,C}$ and COA$_{W,C}$, the mean relative sensitivities are 1 by definition, though uncertainties are
still calculated due to non-zero $a$-values, while the reference profile utilised for CSOA$_{W,C}$, ensures that
CSOA$_{W,C}$ CSOA$_{S,C}$ will have similar sensitivities. Interestingly, the distribution of the sensitivities of
COA$_{S,C}$, COA$_{W,C}$ and CSOA$_{W,C}$ in Fig. S32 and Fig. S33 is clearly multi-modal despite $a$-value
constraints, but the reason for this remains to be explored.





The next group of factors (LABB$_{w,c}$, MABB$_{w,c}$, NitOA1$_{w,c}$, NitOA2$_{w,c}$ and EVENT$_{w,c}$) includes non-
negligible contributions from levoglucosan (C$_6$H$_{10}$O$_5$), produced typically from biomass-burning(BB)-
related activities. Previous work has demonstrated that the EESI-TOF sensitivity to levoglucosan is
higher than that of many other compounds and bulk SOA from representative precursors (Lopez-
Hilfiker et al., 2019; Brown et al., 2021). Indeed, although the set of studied compounds is far from
comprehensive, the relative sensitivity of the EESI-TOF to levoglucosan is among the highest yet
recorded. Therefore, although the composition of the POA-influenced factors varies considerably, it is
possible that the levoglucosan content may have significant predictive value with respect to the overall
factor sensitivity. Figure 8 shows $AS_k$ as a function of the C$_6$H$_{10}$O$_5$ fraction for all factors for which the
C$_6$H$_{10}$O$_5$ signal is believed to result largely from levoglucosan. This analysis accounts for all factors
resolved from the cPMF of the winter dataset except CSOA$_{w,c}$, because CSOA$_{w,c}$ is dominated by the
signal from the protontated nicotine ([C$_{10}$H$_{14}$N$_2$]H$^+$) ion, which is both chemically different (reduced
nitrogen) and has a different ionisation pathway than other measured ions. The four summer SOA
factors are excluded as well, because the contribution from C$_6$H$_{10}$O$_5$ in these factors was previously
attributed to terpene and/or aromatic oxidation products (Stefenelli et al., 2019). An obvious qualitative
trend of increasing sensitivity with increasing levoglucosan fraction is evident with Pearson $r^2$ of 0.676,
indicating the overwhelming influence of the high sensitivity species levoglucosan on the factor
apparent sensitivity.
For the primary BB-related factors, the uncertainties are generally higher than for the other factors (see
Fig. S33 and Fig. S34b). In contrast, the aggregated BB factor (Sum_BB$_{w,c}$, and Sum_BB$_{w,c}$ =
MABB$_{w,c}$ + LABB$_{w,c}$ + NitOA1$_{w,c}$ + NitOA2$_{w,c}$ + EVENT$_{w,c}$) is less uncertain and has a narrower
sensitivity distribution. This suggests that the overall classification of signal as biomass burning-related
is robust, but the subdivision into more specific BB-related sources carries higher uncertainties.
Likewise, the relative sensitivities of sum_DaySOAs$_{s,c}$ and sum_NightSOAs$_{s,c}$ are less uncertain
compared to individual corresponding SOA factors in summer (as shown in Fig. S32 and Fig. S34a).
This contrast suggests that coarse classifications of factors may have higher precision, but provide less
information, whereas fine classifications of factors may have higher uncertainties, but potentially
provide more information from each factor. It also suggests that, at least for these datasets, factor mixing
occurs primarily between factors with closely related sources. Despite their higher uncertainties, the
finest classification levels explored here still appear to be meaningful. We also note that both datasets
investigated here are of relatively short duration, and factor separation may improve in longer datasets.
The final group of factors in Fig. 7 corresponds to SOA. The relative sensitivities of the SOA factors in
winter are shown to be lower than any of the SOA factors resolved during summer. This is consistent
with expectations regarding the seasonal differences in the dominant SOA precursors and the expected
$AS_k$ of the resulting SOA. At this site, SOA precursors are expected to be dominated by monoterpenes
in summer, and biomass burning (increasing the contribution of phenols, naphthalenes, and other
aromatics) in winter, with traffic making a lesser contribution in both seasons (Daellenbach et al., 2016;
Qi et al., 2020). This is supported by analysis of the characteristics of the retrieved factors as discussed
above (Qi et al., 2019; Stefenelli et al., 2019). Previous studies have shown differences in the EESI-
TOF bulk sensitivity to SOA from different precursors, with terpene-derived SOA generally exhibiting
higher sensitivity than SOA from light aromatics (Lopez-Hilfiker et al., 2019; Wang et al., 2021). Figure
9 shows the $AS/\overline{AS_{COA}}$ for two DaySOA$_{s,c}$ and NightSOA$_{s,c}$ factors in summer, as well as the
sum_DaySOAs$_{s,c}$ and sum_NightSOAs$_{s,c}$, which are the aggregates of the individual DaySOA$_{s,c}$ and
NightSOA$_{s,c}$ factors (sum_DaySOAs$_{s,c}$ = DaySOA1$_{s,c}$ + DaySOA2$_{s\_c}$; and sum_NightSOAs$_{s,c}$ =
NightSOA1$_{s,c}$ + NightSOA2$_{s,c}$), respectively, and two SOA$_{w,c}$ factors in winter as a function of their





H:C ratio calculated from the EESI-TOF component. A trend of increasing sensitivity with increasing
H:C ratio is observed for the summer SOAs, as well as the winter SOAs ($SOA1_{w,C}$ and $SOA2_{w,C}$).
For the SOA factors, we compare $AS_k$ retrieved to $AS_k$ predicted using a molecular formula-based
parameterisation trained with laboratory SOA measurements, as described in Sect. 2.2.3 (Wang et al.,
2021). No parameterisations presently exist for POA factors, so these are excluded from the comparison,
although to allow comparison between campaigns the model is used to calculate a reference value for
$AS_{COA}$. Figure 10 compares the $AS_k$ values based on model predictions against values determined from
cPMF. For summer SOAs, the LMN (limonene)-based parameterisation is applied as a surrogate for
terpene oxidation products. Regarding the winter SOAs, three scenarios (cresol, LMN and TMB) are
applied, as the winter SOAs in Zurich are mainly related to oxidation of biomass burning emissions,
which include monoterpenes, phenols, naphthalenes, and other aromatics (Bruns et al., 2016; Kelly et
al., 2018; Rouvière et al., 2006). In Fig. 10, 1:1, 1:2, 1:4, and 1:8 lines are provided to guide the eye,
although a 1:1 correspondence is not expected because the models are not trained on primary COA. The
figure shows a monotonic increase in model sensitivity predictions with increasing cPMF-derived
sensitivities, with the sole exception of $SOA2_{w,C}$. Specifically, the summer-derived points fall mainly
between the 1:1 and 1:2 lines, while for $SOA1_{w,C}$, the model predictions are roughly a factor of 2 lower
relative to the cPMF results. This offset may reflect differences in the appropriateness of the selected
precursor surrogate. The $SOA2_{w,C}$ factor is a slight outlier, probably because the $AS_k$ for this factor is
more uncertain than the others (and not fully captured by the error bars in Fig. 7) due to the high
contribution from inorganic nitrate (~80 % of mass) in its factor profile. Given the limitations of the
multi-variate parameterisation (see Sect. 2.2.3) and the several orders of magnitude variation in EESI-
TOF sensitivities to individual compounds, the qualitative agreement between $AS_k$ values independently
retrieved from multivariate parameterisation and cPMF provide support for both methods.

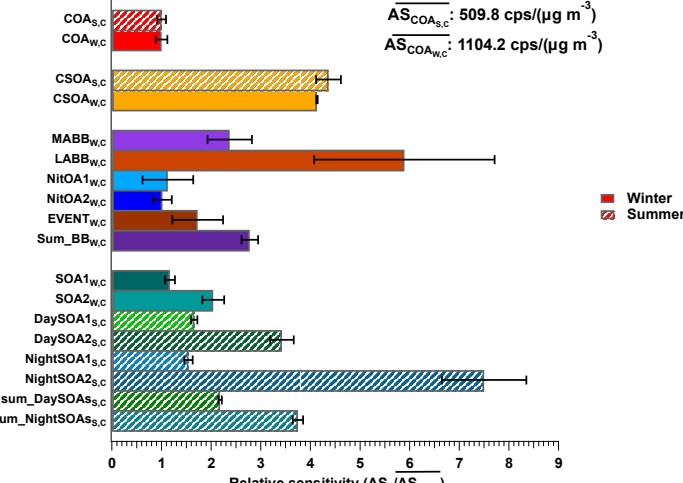

Figure 7. Comparison of $AS_k / \overline{AS_{COA_C}}$ of different factors resolved from the cPMF on the summer and
winter datasets. Mean values are shown as bars, and error bars indicate the standard deviation over all
accepted bootstrap runs. The following factor aggregations are also shown: $Sum\_BB_{w,C} = MABB_{w,C}$
$+ LABB_{w,C} + NitOA1_{w,C} + NitOA2_{w,C} + EVENT_{w,C}$; $sum\_DaySOAs_{s,C} = DaySOA1_{s,C} +$
$DaySOA2_{s\_C}$; and $sum\_NightSOAs_{s,C} = NightSOA1_{s,C} + NightSOA2_{s,C}$.




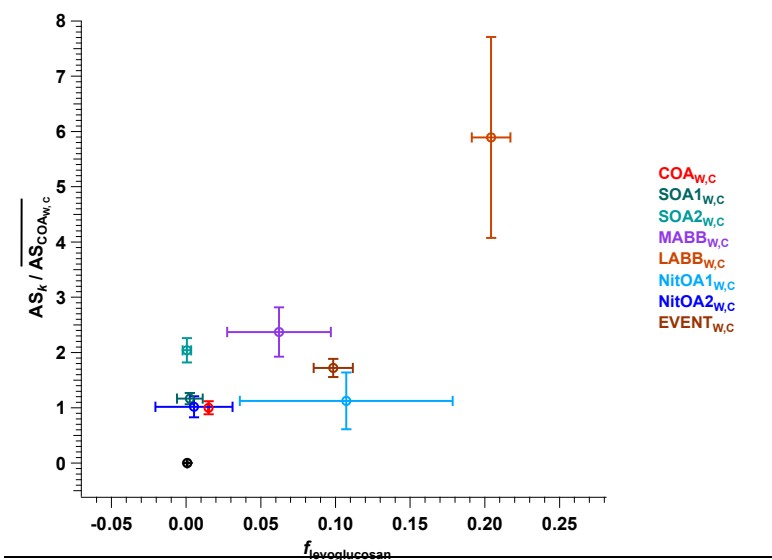

Figure 8. Relative apparent sensitivity $AS_k/\overline{AS_{COA_{W,C}}}$ as a function of levoglucosan fraction for all

factors resolved from the cPMF of the winter dataset except $CSOA_{W,C}$. Error bars denote standard

deviation.

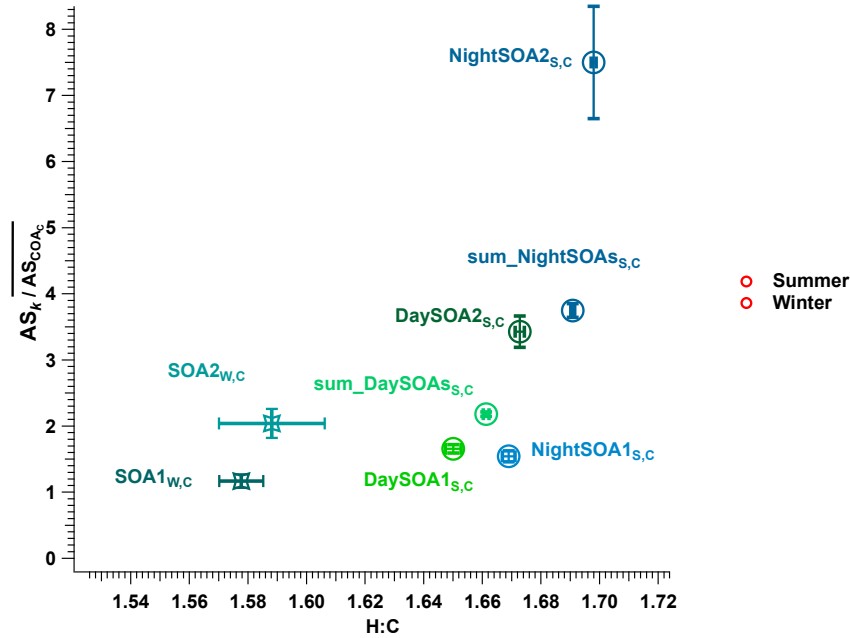

Figure 9. $AS_k/\overline{AS_{COA_C}}$ of SOA factors retrieved from the summer and winter datasets as a function of

the H:C ratio. Error bars denote standard deviation across all accepted runs.

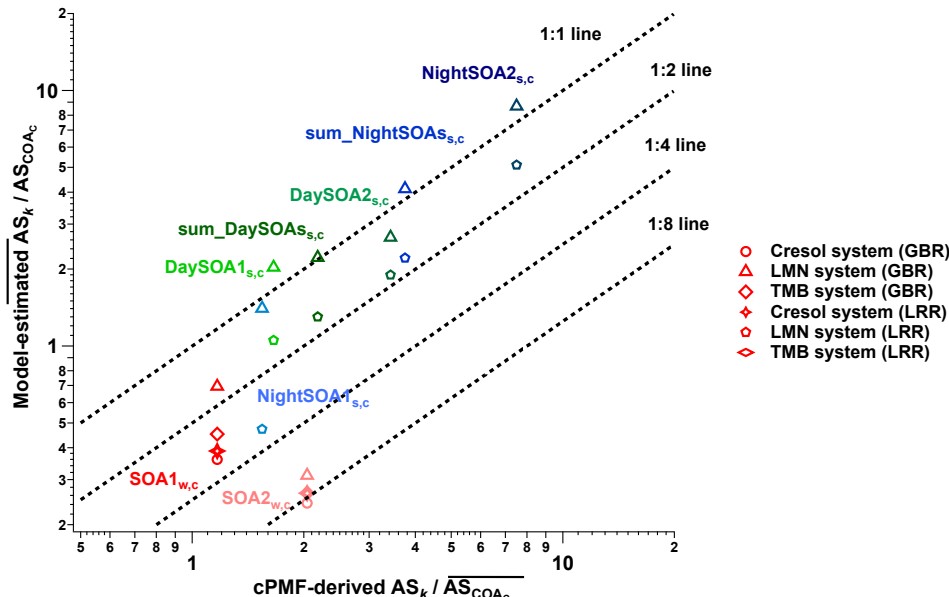

Figure 10. The estimated relative apparent sensitivity to COA ($AS_k/\overline{AS_{COA_C}}$) from the gradient boosting regression (GBR) and linear ridge regression (LRR) models as a function of cPMF-derived $AS_k/\overline{AS_{COA_C}}$. The symbols indicate the different oxidation-precursor system (LMN for SOA produced from oxidation of limonene by ozone, cresol and TMB for SOA produced from oxidation of $o$-cresol and 1,3,5-trimethylbenzene by OH radicals, respectively).

## 4. Atmospheric implications

The application of factor-dependent sensitivities can qualitatively and quantitatively affect the source apportionment results. Figures 11a and 11b compare the source apportionment results from cPMF on the summer and winter datasets using the calculated factor sensitivities $(AS_k)$ (i.e., direct outputs of the cPMF analysis) vs. using a single bulk sensitivity $(AS_{bulk})$ for all factors, where the latter is calculated as the ratio of the total OA measured by the EESI-TOF (cps) to that measured by the AMS ($\mu$g m$^{-3}$). Figures 12a and 12b compare the total OA concentrations returned from the cPMF using $AS_k$ and $AS_{bulk}$ to the total OA measured by the AMS. Table S3 summarises the retrieved $AS_k$ values for each factor (note that although the relative $AS_k$ are believed to be intrinsic properties of the factors, the absolute sensitivities are instrument- and tuning-dependent, and will vary between campaigns).

In the Zurich summer campaign, the bulk OA sensitivity $AS_{bulk_{S,C}}$ (1254.0 cps /( $\mu$g m$^{-3}$)) is higher than that of $AS_{COA_{S,C}}$ (509.8 cps /( $\mu$g m$^{-3}$)). Four factors (HOA$_{S,C}$, COA$_{S,C}$, DaySOA1$_{S,C}$ and NightSOA1$_{S,C}$) are underestimated, whereas three factors (CSOA$_{S,C}$, DaySOA2$_{S,C}$ and NightSOA2$_{S,C}$) are overestimated when $AS_{bulk_{S,C}}$ is used. Using the calculated $AS_k$, the contribution of COA$_{S,C}$ to total OA more than doubles, from 4.5 % to 11.7 % as shown in Fig. 11a). Similarly, the application of $AS_k$ increases the contributions of DaySOA1$_{S,C}$ and NightSOA1$_{S,C}$ from 22.7 % to 35.2 %, and from 10.3 % to 17.1 %, respectively. Among the overestimated factors, the largest decrease post-correction is found for NightSOA2$_{S,C}$, the contribution of which decreases by approximately a factor of three (from 29.7 %



to 10.3%). Smaller post-correction decreases are observed for the contributions of CSOA$_{S,C}$ (12.9 % to
7.7 %) and DaySOA2$_{S,C}$ (19.9 % to 14.9 %). If factor-dependent sensitivities were ignored,
NightSOA2$_{S,C}$ would be the largest contributor to total OA, followed by DaySOA1$_{S,C}$ whereas the full
analysis indicates that DaySOA1$_{S,C}$ is the largest contributor.
Similar to the summer campaign, application of $AS_k$ significantly affects the source apportionment
results in winter. CSOA$_{W,C}$, MABB$_{W,C}$, and LABB$_{W,C}$ are shown to be overestimated, while HOA$_{W,C}$,
COA$_{W,C}$, SOA1$_{W,C}$, NitOA1$_{W,C}$, NitOA2$_{W,C}$ and EVENT$_{W,C}$ are underestimated. If factor-dependent
sensitivities were not considered, LABB$_{W,C}$ and MABB$_{W,C}$ would appear to be the dominant
contributors to total OA (35.7 % and 18.2 % respectively) due to their high levoglucosan content.
However, the full cPMF analysis indicates the LABB$_{W,C}$ and MABB$_{W,C}$ contributions to be 14.9 % and
14.4 %, respectively, whereas accounting for $AS_k$ increases the contribution of SOA1$_{W,C}$ from 12.7 %
to 22.0 %, making it the largest contributor.
For both the summer and winter datasets, calculation of total OA from cPMF results using factor-
specific $AS_k$ significantly outperforms that using a single $AS_{bulk}$. This is evident from an increased $r^2$
(0.966 vs 0.821) for summer. However, the $r^2$ is similar between the two approaches in winter (0.947
vs 0.943). The difference after applying $AS_k$ and $AS_{bulk}$ in $r^2$ might be related to the extent which the
contribution from factors with high $AS_k$ and low $AS_k$ to total OA changes over the time during the
campaign, which can vary in different datasets.
Box-and-whisker diagrams of factor contributions to total OA with/without applying $AS_k$ values for
summer and winter are presented in Fig. 13. In the Zurich summer campaign, the box plots of the
corrected contributions of all six factors fall completely outside of the interquartile range (IQR) of the
uncorrected results, suggesting that the use of a single $AS_{bulk}$ would lead to significant biases. In contrast,
the winter campaign exhibits a lack of overlap between the $AS_k$ and $AS_{bulk}$-derived results for eight
factors (HOA$_{W,C}$, COA$_{W,C}$, CSOA$_{W,C}$, SOA1$_{W,C}$, SOA2$_{W,C}$, NitOA1$_{W,C}$, NitOA2$_{W,C}$ and EVENT$_{W,C}$) ,
whereas two factors overlap (SOA2$_{W,C}$ and MABB$_{W,C}$,). This may result from statistical uncertainties
in bootstrap analysis coupled with a less robust division between certain factors, yielding a wide
distribution, e.g., MABB$_{W,C}$, and/or $AS_k$ values that are similar to $AS_{bulk}$ (2271.1 cps /( µg m$^{-3}$)), e.g.,
SOA2$_{W,C}$ (2253.2 cps /( µg m$^{-3}$)), and MABB$_{W,C}$ (2619.0 cps /( µg m$^{-3}$)).

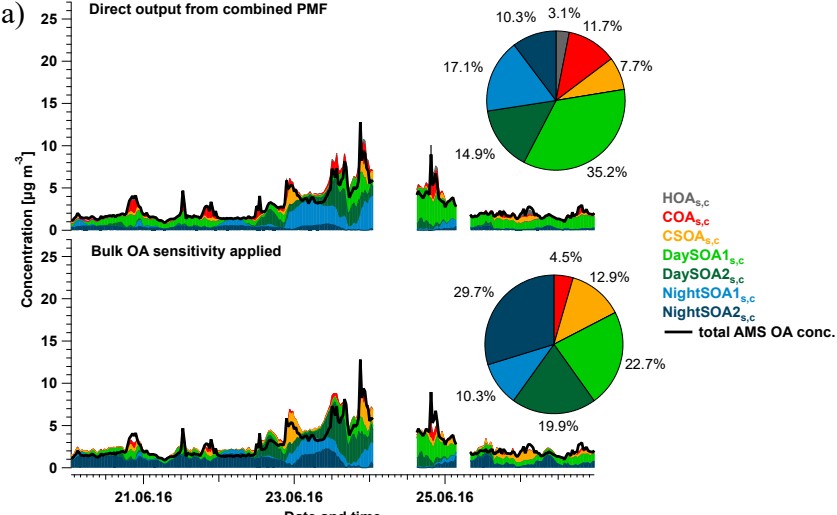

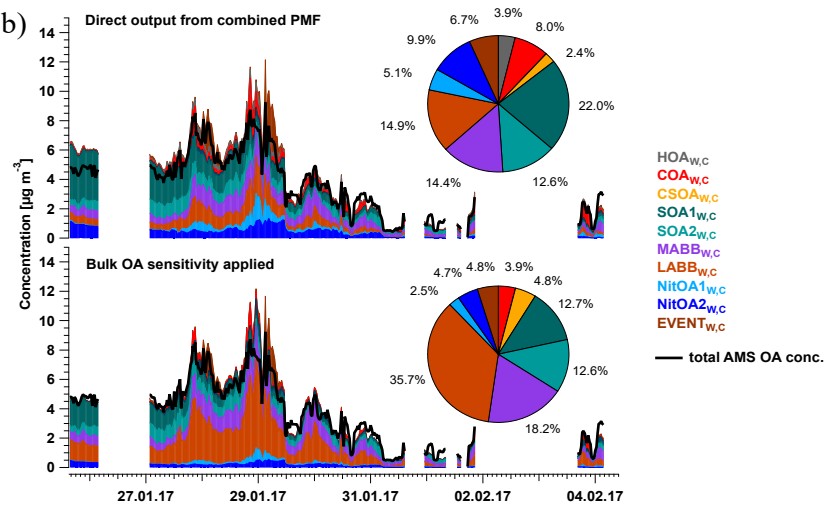

Figure 11. Comparison of source apportionment results between direct output from cPMF (i.e., accounting for factor-dependent sensitivities) and application of a single bulk OA sensitivity, applied to the Zurich summer (a) and winter (b) datasets. Stack plots of factor time series directly from combined PMF and factor time series calculated from bulk OA sensitivity compared with total AMS OA concentration are shown in the upper and lower panel, respectively in each subfigure, together with the corresponding factor contribution shown in the pie chart. Note that here the contribution of the InorgNit factor and the contributions of $NO^+$ and $NO_2^+$ from inorganic nitrate in each factor are excluded to account only for the total OA.



a)

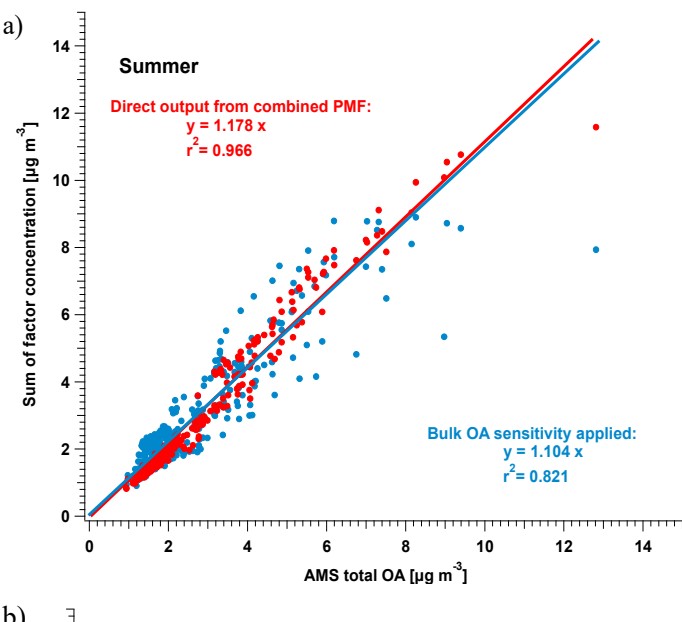

b)

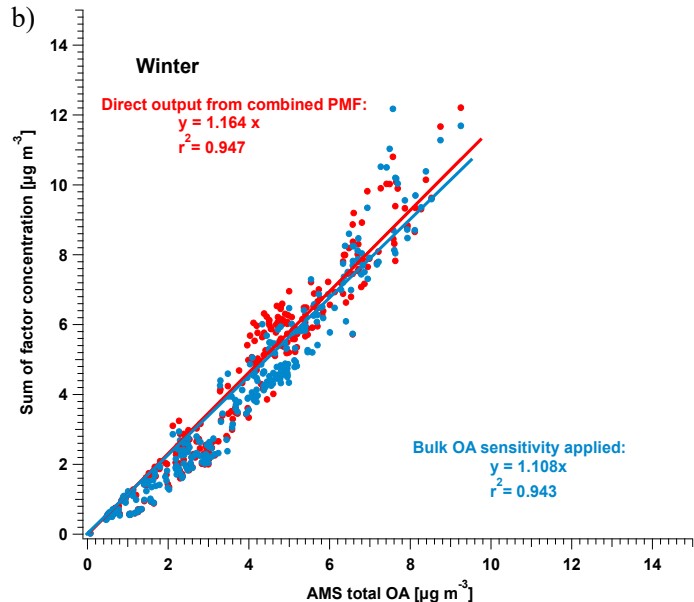

Figure 12. Comparison between the sum of factor concentrations in each time point with (in red) and
without (in blue) taking the factor-dependent sensitivity into account and total OA measured by AMS
for summer in a) and winter in b). A linear fit is conducted based on the Levenberg-Marquardt least
orthogonal distance method. Note that here the contribution of the InorgNit factor and the contributions
of $NO^+$ and $NO_2^+$ from inorganic nitrate in each factor are excluded.



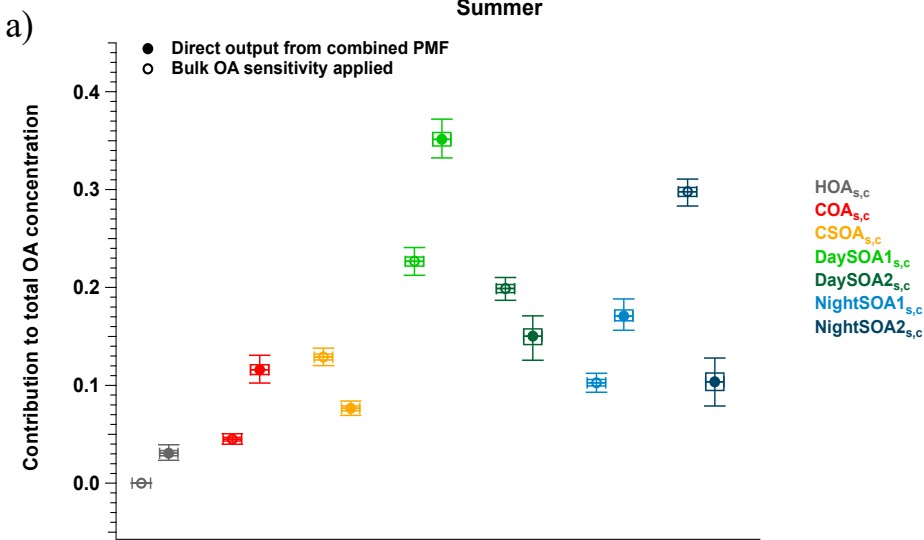

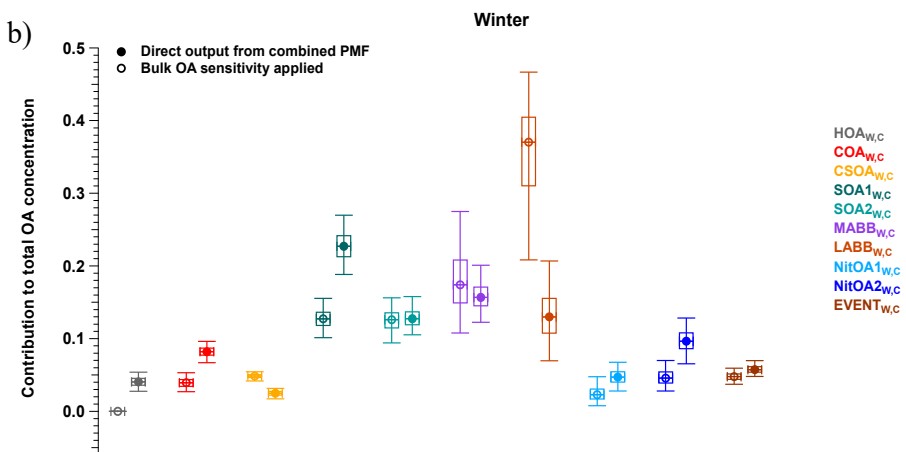

Figure 13. Box-and-whisker diagrams of factor contribution to total OA with/without applying the
factor dependent sensitivities, for summer in a) and winter in b) within accepted solutions. For each pair
of factors, the contribution without factor-dependent sensitivity applied is shown in the left box (open
symbols), whereas the contribution corrected by factor-dependent sensitivity is shown in the right box
(full symbols). The plot shows the mean (open/filled circle), median (horizontal bar), interquartile range
(rectangle), and minimum/maximum values (whiskers). Note that here the contribution of InorgNit
factor and contribution of $NO^+$ and $NO_2^+$ from inorganic nitrate in each factor are excluded.





**5.    Conclusion**
We address the longstanding challenges in achieving quantitative source apportionment of SOA sources
by conducting a positive matrix factorisation (PMF) analysis of a dataset combining measurements from
an aerosol mass spectrometer (AMS) and an extractive electrospray ionisation time-of-flight mass
spectrometer (EESI-TOF). This approach combines the strengths of the two instruments, namely the
quantification ability of the AMS and the chemical resolution of the EESI-TOF. We demonstrate the
utility of this approach by PMF analysis of combined EESI-TOF/AMS datasets collected during
summer and winter in Zurich, Switzerland. The results retain the chemical resolution of the standalone
EESI-TOF PMF, while additionally providing quantitative factor time series and the EESI-TOF bulk
sensitivity to different OA factors.
We present a general procedure to conduct source apportionment on a combined dataset, including a
new metric for ensuring both instruments are well-represented in the solution, a method for optionally
constraining factor profiles for one or both instruments, and a protocol for uncertainty analysis. The
balancing metric references residual distributions obtained from cPMF to those of optimised single-
instrument PMF solutions to avoid bias due to differing instrument characteristics or error models.
Factor profile constraints require the construction of a reference profile, which may be challenging in a
multi-instrument dataset. We therefore provide methods for reference profile construction for cases
when (1) a single reference profile exists combining data from both instruments; (2) reference profile
exist independently for each instrument; and (3) a factor is detectable by one instrument but not the
other. To explore the solution stability and the uncertainties, a protocol for combined bootstrap
analysis/constraint exploration is developed.
The cPMF method intrinsically provides factor-dependent sensitivities (cps / (ug/m$^3$) for the EESI-TOF.
To account for organonitrate content, the AMS ions NO$^+$ and NO$_2^+$ are included in the cPMF analysis.
Organic and inorganic contributions to these ions are estimated on a factor-by-factor basis using the
method of Kiendler-Scharr et al. (2016).
For practical reasons, sensitivities between winter and summer campaigns are compared using cooking-
related OA (COA) as a common reference. The retrieved factor sensitivities range from approximately
1.3 to 7.5 times the sensitivity of COA. The relative sensitivities of SOA factors are precursor-
dependent, and qualitatively consistent with trends observed in lab measurements of SOA from single
precursors (Lopez-Hilfiker et al., 2019). The SOA sensitivities estimated using our cPMF approach also
agree with the sensitivities predicted by multi-variate regression models (Wang et al. 2021), which
further demonstrates that SOA sensitivities are precursor- and/or source-dependent. Comparison of
source apportionment results using factor-dependent sensitivities to uncorrected results show
substantial differences, highlighting the importance of quantitative analysis. For example, before
applying factor-dependent sensitivities, the contribution of a daytime SOA factor is underestimated by
about 30 % (22.7 % before vs 35.2 % after), whereas the contribution of a nighttime SOA factor is
almost overestimated by a factor of 3 in the summer campaign (29.7 % before vs 10.3 % after). As for
the winter campaign, the contribution of less-aged biomass burning factor to total OA in Zurich winter
dataset is 35.7 %, making it a major factor in winter without considering its factor-dependent sensitivity.
However, this factor is significantly overestimated by more than a factor of 2 (35.7 %, before vs 14.9 %
after). In contrast, the SOA1 factor in winter is underestimated, with its contribution increasing from
12.7% to 22.0 %.



The cPMF method presented herein is can be utilised as-is not only for the AMS/EESI-TOF combination, but to any dataset comprising data from multiple instruments. As such, it provides a promising strategy for utilising instruments with high chemical resolution but semi-quantitative performance (i.e., a linear but hard-to-calibrate response to mass) within the framework of a quantitative source apportionment.

*Data Availability.* The data presented in the text and figures will be available at the Zenodo Online repository (https://zenodo.org) upon final publication.

*Competing interests.* The authors declare that they have no conflict of interest.

*Author contributions.* GS and LQ conducted the campaigns in summer and winter in Zurich, respectively. YT performed the whole analysis. DSW performed the multi-variate model for machine learning parameterisation of sensitivities. FC developed the weighting and constraining functions in SoFi. JGS conceived and supervised the project. All authors currently working at PSI contributed to the data interpretation. All authors contributed to the manuscript revision.

*Acknowledgements.* We gratefully acknowledge the contribution from Dr. Anna Tobler and Mr. Gang Chen for coordinating the workstation for the computationally-intensive bootstrap analysis.

*Financial support.* This project has received funding from the Swiss National Science Foundation (grant no. BSSGI0_155846) and the European Union's Horizon 2020 Research and Innovation Program under the Marie Skłodowska-Curie grant agreement no. 701647.



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
