# Peer review of "Quantification of primary and secondary organic aerosol"

_Atmospheric Measurement Techniques, 2022_

## Referee Comment (RC1)

**Review report Tong et al. AMTD 2022**

In this study, Tong et al. used the capability of Positive Matrix Factorisation (PMF) analysis to combine data from multiple sources in one dataset for analysis. They used the very detailed molecular composition information obtained from an Extractive Electro Spray Ionisation mass spectrometer (EESI-TOF) together with data from simultaneous Aerosol Mass Spectrometer (AMS) measurements, which provide almost no molecular information but instead enables the quantification of the measured OA signals. As each factor obtained in this combined PMF (cPMF) contains both AMS and EESI-TOF signals, apparent sensitivities could be derived for the EESI-TOF data. This revealed that without considering the compound specific sensitivity for this instrument, the contribution of specific sources (i.e., the identified factors) may be strongly over- or underestimated and thus alter the interpretation of the measurement data.

The underlying principle (combining variables from different instruments) is a basic feature of PMF in general, but the authors provide a clever way of using an established method in a new way to improve the interpretation of EESI-TOF data. As this instrument is becoming more and more important for atmospheric measurements, this manuscript provides very valuable suggestions for the analysis and interpretation, especially for source apportionment. But the implication of the derived apparent sensitivity are also very relevant.

I recommend publication in AMT after some minor issues are addressed.

General Comments:

1) In general, the authors do a good job linking their new analysis results to the already published ones for the two data sets used here. However, it would be easier for the reader if a few more details would be repeated in the main manuscript and not just given by reference.

   a. It remains unclear why some of the factors were constrained. Yes, the previous studies provide explanations for their decisions. But constraining a factor in the single instrument PMF to enhance its separation so that a lower factor number could be used, may no longer be necessary in the combined data set. Did the authors test if the factor constraining was really necessary in cPMF? How did they determine if a factor had to be constrained in cPMF?

   b. How was it determined if a factor was "mixed"?

   c. It would be beneficial to emphasise to the reader a bit earlier, that the chose solution for single instrument PMF was based on the considerations done for the two previous studies and that also the reference spectra are related to that.

   d. Were the same AMS and EESI-TOF instruments used in both studies? Do the authors expect changes in the instrument behaviour (e.g. change in Pieber effect for AMS)? The change in the $NO^+$ and $NO_2^+$ ratio for the AMS calibration with Ammonium Nitrate seems to suggest some instrumental differences.

2) The authors claim that they did not observe any relevant fragmentation or clustering with solvent molecules in the EESI-TOF. I find that hard to believe since other studies reported strong fragmentation for specific ions (Bell et al., 2021). Also, how could it be determined that no clusters with the solvent occurred? Especially with acetonitrile, the danger would be to interpret the N from the solvent molecule as a nitrated organic compound.

3) I wonder how computationally intensive this is. There are 2 single instrument PMF runs (with bootstrap, rotations etc). Then the exploratory cPMF run. Then at least 2 more bootstrap cPMF to constrain the additional parameters like $C_{EESI}$ (or the 2D-scan for the a values). Noting that two very short dataset were chosen for this proof-of-concept study, I really wonder how feasible this is for a 2 or 3-month long campaign data set.

4) The authors show that for each data set a new set of parameters (a- constraints, $C_{EESI}$, $AS_k$) have to be determined. However, it would be good to really clearly state that once more in the conclusions. Lately I have come across publication which took data set specific parameters from older studies (e.g., for instrument calibrations) and blindly applied them to their data sets. While that is of course the shortcoming of these people and not the original authors, we can try to emphasise that it is really the method that is being presented and that every new data set needs its own careful exploration of the parameter space to provide reliable data interpretation.

5) This is just a suggestion: The manuscript is very long with a lot of important details about how to conduct this new version of PMF analysis. A long manuscript is not per se a problem – especially if a "new" analysis method is introduced. But the authors should consider if some of the more technical aspects could be shifted to the Supplement Material – or be presented as an Appendix. That would put a bit more emphasis on the interpretation of the cPMF results like the changes in factor contributions depending on the apparent sensitivity, which is currently a little bit hidden under all the (important) technical details.

Specific comments:

1) P4 L18f: in addition to the PMF studies Lee et al. (2014) could be mentioned here as another study investigating the sources and formation processes of OA. They used FIGAERO-CIMS spectra obtained in dedicated chamber experiments to reconstruct ambient measurements as linear combinations of them.

2) P4 L38: typo: wind speed (WD) -> (WS)

3) P5 L14: typo: "by the massed base method" -> with the mass based method

4) P6 L9: What was the schedule for the switching between direct sampling and background measurements? I.e. how long between each background measurement? Were there strong changes observed between adjacent background measurements?

5) P6 L25ff: For AMS PMF analysis, signals with SNR = 0.2 – 2 are usually considered as "weak" and downweighed by a factor of 2-3. Only signals with SNR < 0.2 are removed. Why did the authors choose to remove signals with SNR <2?

6) P7 L23 & Fig 10: The authors do not explain anything about the two chosen methods (GBR vs LRR). Later in section 3.3 they also do not provide any more information about what these two methods mean, why they were used, or what the different values mean. As the paper is already very long as it is and only a qualitative comparison is conducted anyway, I recommend that the authors decide which of the methods they want to use here. To me the trends seem identical with just a general offset between LRR and GBR.

7) P15&16: Here the similarities of the single instrument PMF with the new data set are described and compared to the original ones (although only for AMS). The information presented here is not really used in the rest of the manuscript. This could be a section that could be moved to the supplement and replaced by a sentence stating that the new factors are similar enough to the old ones so that all the interpretations from the previous studies are still valid (see SI for details).

8) Table 1: out of curiosity, how many "common ions" are there in the two EESI-TOF data sets? I.e., how many of the 892 ions identified for the winter data were also present in the summer data?

9) P18 L22: It is not clear to how the authors handled the HOA and InorgNit factor reference spectra for the EESI-TOF part of the combined dataset. What was the "same intensity" to which all ions were set? Why was $0.01$ cps $(ug\,m^{-3})^{-1}$ chosen? Why did they not just set the values to 0 (or a very small number, e.g. 1e-6)?

10) P19: where the factor profiles constrained for the $C_{EESI}$ analysis in section 3.1.3?

11) P19 L29f: The authors state that the summer data follows the "expected" trend of the overlap values with $C_{EESI}$. But why is that trend expected? And what does it mean that the winter data does not follow the same trend (i.e. having the low values for p=7 and $C_{EESI} = 0.5$)? Could it be connected to the change in EESI background due to using a different solvent?

12) P20: Do the authors have any idea why the $C_{EESI}$ value was so different for the two data set? I assume the AMS sensitivity can be considered as constant. Which changes may have occurred to the EESI to cause this value to change by 2 orders of magnitude (2 -> 0.05)?

13) P26 L9: Will the world may agree on the time for lunch, dinner time is very culture specific (just ask an Italian and a Finn ;-). It may be more specific to give the time of day in the description of the COA factor and then clarify that those times correspond to lunch and dinner in Switzerland.

14) P26 & Fig S28, S29: It is very good that the authors provide the factor mass spectra for comparison. But the chosen visualisation may not be the best. As this is HR data, sticks may overlap and not be distinguishable (especially for EESI-TOF). The authors could try using a modified Kroll diagram (OSC vs carbon number) with showing the signal intensity of the base case as the size of the symbols. Then a colour code could be used to show the difference to the compared factor. That would also highlight if certain groups of compounds are different (e.g. dimers).

15) P27 L14. Hereafter the notation "Sum_BB" etc is used. It is clear what the authors mean, but it may enhance readability if the Greek Σ were used instead of Sum_ (e.g., Σ BB). Especially in Fig 7 and Fig 10, this would highlight at these values are something different from the after bars/markers.

16) P29 L22: It feels a bit odd that one of the arguments to constrain individual factor profiles was to prevent factor splitting. But now the authors recombine two factors which they deem to have been split artificially. Since I am not a fan of overly constrained PMF, I do not want to recommend constraining LABB. But it makes me wonder if using the same constraints in cPMF as in single PMF is the right approach here.

17) P30 L 1 & 29: The authors have used correlations for almost everything to express similarity. But now for the factors they use "qualitatively similar". What does that mean here? Is there a reason why the authors do not use a mathematical measure for the similarity of their factors (e.g., uncentered R or contrast angle)? How can they be qualitatively similar if one contains ON and the other does not?

18) P30 L 1ff: Also, how meaningful is it to compare the factor mass spectrum of the constrained factors here? How much could e.g. HOA really vary with the set constraints? E.g. how big are the differences of HOA for summer and winter?

19) P30 L 9 "consistent with other studies" If the authors use the PMF factor from Qi et al. to create a constrained factor, they cannot claim that this factor than is in good agreement with Qi et al. (unless the constraint was so loose that a strong variation was truly possible)

20) Section 3.3: I find it difficult to follow in this section when the authors are talking about the Fig S32 and S33 and when they are using the "scaled" values in Fig 34 in their arguments. I also do not understand what the benefit is of Fig 34. The spread of the $AS_k$ values are already visible in S32&33. For me the additional scaling was more confusing than enlightening.

21) p33 L43: how is COAs,c multimodal in Fig S32?

22) p 34 L 7ff I find the predictive value of levoglucosane/$C_6H_{10}O_5$ very limited. It only works for BB factors in which the ion associated with levoglucosane is not disturbed by anything. If the origine of the factors have not yet been interpreted, this value may be misleading.

23) P35 L1f & Fig 9: I find the correlation of H:C and relative $AS_k$ rather poor. 5 markers fall into the square H:C 1.58 – 1.66 and AS 1-2 without any clear trend in my eyes. Also, why was H:C chosen? How would this look with O:C or OSc?

24) Section 4: In my opinion the authors could put a bit more emphasis on this part of the manuscript. The implications for the interpretation of the factor contributions for the EESI-TOF data are big while the AMS part as not affected as much. To me, that is an important message for any PMF analysis of instruments which lack detailed sensitivity information (so also FIGAERO-CIMS). Already for one instrument applied at the same location but in two different seasons, such big differences are observed. How careful should we then be when comparing factors obtained for different locations.

25) Fig 11-13: Although I would like to see a bit more emphasis on the Atmospheric Implications section, I do think that the authors should decide if they prefer Fig 11 or Fig 13 as they do show the same information. One of them could go to the supplement information. Also Fig 12 is not providing that much exiting visual information. I take from Fig 12 that for AMS pmf factors, the sensitivity does not matter (as could be expected).

26) P39 Fig 9 The summer and winter symbol are both circles in the legend. I think you need a square for winter.

27) P 42 L13f: "optionally constraining factor profiles for one or both instruments" I did not find the place where a factor was constrained for one of the instruments but not for the other. As I understood the method, that is not possible. One a-value is applied for a factor containing both EESI-TOF and AMS ions. Hence a factor is either constrained (to a certain degree) or not.

References

Bell, D., Wu, C., Bertrand, A., Graham, E., Schoonbaert, J., Giannoukos, S., Baltensperger, U., Prevot, A., Riipinen, I., El Haddad, I. and Mohr, C.: Particle-phase processing of α-pinene NO3 secondary organic aerosol in the dark, Atmos. Chem. Phys. Discuss., 1–28, doi:10.5194/acp-2021-379, 2021.

Lee, B. H., Lopez-Hilfiker, F. D., Mohr, C., Kurtén, T., Worsnop, D. R. and Thornton, J. A.: An iodide-adduct high-resolution time-of-flight chemical-ionization mass spectrometer: Application to atmospheric inorganic and organic compounds, Environ. Sci. Technol., 48(11), 6309–6317, doi:10.1021/es500362a, 2014.

---

## Referee Comment (RC3)

**Review of "Quantification of primary and secondary organic aerosol sources by combined factor analysis of extractive electrospray ionisation and aerosol mass spectrometer measurements (EESI-TOF and AMS)"**

**Brief summary**

In "Quantification of primary and secondary organic aerosol sources by combined factor analysis of extractive electrospray ionisation and aerosol mass spectrometer measurements (EESI-TOF and AMS)" the authors take on the challenging task of not only performing positive matrix factorization (PMF) source apportionment on a relatively new measurement method, EESI-TOF-MS, but also combine that dataset with AMS data to present a new application of PMF they call "combined PMF (cPMF)". In great detail the authors outline the procedure they followed to perform cPMF. They present a PMF factor-based approach to estimating sensitivities from the EESI-TOF and describe how they use bootstrapping to estimate uncertainties for the PMF factors. After reading the paper I have an appreciation for the challenging of appropriately preparing and evaluating the data input to and produced from the cPMF analysis. I suspect any researcher utilizing this technique in the future would have to apply a similar level of rigor. This paper is appropriate for the journal and should be published with minor revisions.

While thorough, rigorous, and interesting, this paper is very long and has a very long supplement. I think it is useful to consider how very long papers can impact the accessibility of a manuscript. For many scientists, including myself, this is a very challenging task in addition to the great effort of doing quality research. I acknowledge the supplement is long mostly because of the 34 supplemental figures, but one way to decrease the length of the main manuscript is to partition some of the information that is highly detailed—but is not absolutely necessary to understand the main points of the manuscript—to sections in the supplement for the advanced reader.

**Major comments**

**Major comment 1:** At the end of section 2.2.2 the apparent sensitivity application informed by the AMS mass is introduced and then said it will be discussed later. Section 2.2.3 basically does the same thing. Can it be briefly mentioned here that two methods of applying EESI-TOF sensitivities were tested in the study and they'll be discussed and evaluated in a different section? In its current state the authors review the methods then say they will discuss them further later.

**Major comment 2:** (page 8, line 13) I'm unclear about what residuals are used and how the residuals are used as a reference to help retrieve a balanced solution from the AMS/EESI-TOF joint PMF. Are the residuals the total residual (basically the Q value) of the "best" stand-alone AMS and EESI-TOF PMF runs? What does it mean to use these residuals "as a reference to retrieve a balanced solution"? Instead of "(step 4)" could you replace that with "(procedure described in step 4)"? It's unclear if, when reading this, the reader should skip to step 4.

**Major comment 3:** Can the authors consider changing the title of section 2.3 to "combined PMF method" or something similar? It would describe the section better as this section is a set of instructions for researchers who would want to apply an identical or similar method to their datasets.

**Major comment 4:** I'm finding the presentation of section 2.3 hard to follow. There's description of some unique things that were done for the PMF analysis in this paper and there's description of PMF generally that have been published many times. I have some suggestions for clarification.

(1) Change figure 2 to figure 1.

- (2) Combine the ideas from (page 8, lines 1-9) and (page 9, lines 6-24) with each other and use it to introduce section 2.3. Trim it down considerably. Can say "We used PMF and in one sentence this is what it is and what it does. We combined the mass spectral time series from the EESI-TOF and AMS to create the input matrix for the PMF analysis. We performed PMF on the combined input matrix and we call this "combined PMF" (cPMF). A conceptual schematic is shown in Figure 1." Two additional things to keep that the authors mentioned that I thought was useful was the respective units of the AMS and EESI components and the inclusion of NO+ and NO2+.
- (3) Remove (page 10) lines 7-19 and reference either the Paatero papers or Ulbrich's review.
- (4) More or less remove (page 9) lines 11-19 and equation 5 for the same reason.
- (5) In a new paragraph make the major points "We present an overview of the cPMF through a series of steps listed below. Details corresponding to each step are outlined in subsequent sections. The overall procedure is outlined in Figure 2, with the main steps as follows: ....".
- (6) Title subsections as 2.3.# where # = 1-6 (representing each step in cPMF) and include the relevant information to perform that step in that section. For example, the first subsection would be Section 2.3.1 with a title of "Step 1: Conventional PMF". In this section you'll say PMF was performed on the AMS and EESI-TOF datasets independently. You'll also explain how constraints on factor profiles were applied. Section 2.3.2 with a title "Step 2: Creating input matrix for cPMF". Maybe consider combining steps 2 and 3.

It's clear all the elements of the analysis are detailed in the subsections, but it's not clear at what points analyses are performed. For instance, (page 11) lines 1-21 describe the calculation of the apparent sensitivity for the EESI-TOF, but it's unclear at what step(s) in the cPMF method this is done.

**Major comment 5:** I apologize, I might have missed some information; why is 1 ug m-3 used as a reference value?

**Major comment 6:** In subsections 2.3.3 and 2.3.4 the authors do a very nice job of explaining how they treated and evaluated the data and solutions for the cPMF. I think I personally would have to actually go through the process to fully understand all the details.

**Major comment 7:** Consider putting the text between the title of section 3.2 and 3.2.1 as a supplemental section. I was finally excited to see some results with the title "cPMF results", but instead started reading more details of analysis.

**Major comment 8:** The authors have demonstrated the application of cPMF to a multi-season, complex mass spectrometry dataset from two instruments. Despite the thorough and rigorous development and evaluation of the method some curiosities and uncertainties still persisted like the contribution of high sensitivity species in contributing to a factor profile, multi-modality of sensitivity values in the COA and CSOA factors, and increasing uncertainty when sub-dividing factors like in the case of the aggregate BB factor. These uncertainties contribute in the overall uncertainty to a relatively complex data processing procedure. Can the authors briefly provide any laboratory experiments, calibrations, or "ideal" datasets in the conclusions section where this cPMF method could be applied in the future as test cases for improving the interpretability and quality of the cPMF analysis?

**Minor comments**

(**page 3, line 3**) "The corresponding decrease in chemical resolution, particularly for the multifunctional and/or highly oxygenated SOA components molecules of which SOA is comprised..."

Reviewer: Please remove the words indicated above.

(page 3, line 47) "...gas-phase concentrations measured by a Vocus proton transfer reaction-mass spectrometer (Vocus-PTR-MS) (Wang et al., 2021)"

Reviewer: I didn't see the Wang et al. study listed in the references. Please list the following article in the references section "Constraining the response factors of an extractive electrospray ionization mass spectrometer for near-molecular aerosol speciation" Wang, et al. (2021)

(**page 4, line 22**) "The present study is the first application of cPMF to a joint EESI-TOF/AMS dataset, and the first attempt at quantitative EESI-TOF-driven source apportionment."

Reviewer: Please remove words indicated above.

(page 5, line 20) "...from highresolution mass spectral analysis..."

Reviewer: Replace "highresolution" with "high resolution" or "high-resolution".

(page 16, line 2) Is "(2.54)" a ratio value? If so can you change it to read (NO+/NO2+ = 2.54)? It's unclear what (2.54) means as is.

(**page 30, line 2**) Replace "This factor has a qualitatively a profile similar to the summer campaign..." with "This factor qualitatively has a profile similar to the summer campaign...".

(**page 33, paragraph 2**) I think using the COA profile as a reference is an appropriate solution to a tough problem.

(**Figure 8**) Figure 8 has a line through the x-axis label. I assume this is a formatting/review feature accidentally carried into the PDF from Word?

(Figure 13) Can you set the maximum value for the y-axis in panel a = 0.5? In the caption can you note what defines the box and whiskers (i.e. are the boxes  $25^{\text{th}}$  and  $75^{\text{th}}$  percentiles?)

(page 43, line 1) "The cPMF method presented herein is can be utilised as-is not only for the AMS/EESI-TOF combination..."

Reviewer: Please delete the misplaced "is" in the sentence above.

---

## Author Comment (AC1)

**Response to RC1**

We thank the reviewer for the helpful comments. Below we provide a detailed point-by-point response to the issues raised by the reviewer. Reviewer comments provided in *italics* and our responses follow in normal text. Changes to the manuscript are denoted in blue font. When our responses reference other comments, we use the formalism R#C#, such that R1GC5 and R1SC5 would refer to General Comment 5 by Reviewer 1, and Specific Comment 5 by Reviewer 1. When indicating the page and line, we use the formalism P#L#, such that P10L5 would refer to Page 10 Line 5

**General Comment #1**

*In general, the authors do a good job linking their new analysis results to the already published ones for the two data sets used here. However, it would be easier for the reader if a few more details would be repeated in the main manuscript and not just given by reference.*

1) *It remains unclear why some of the factors were constrained. Yes, the previous studies provide explanations for their decisions. But constraining a factor in the single instrument PMF to enhance its separation so that a lower factor number could be used, may no longer be necessary in the combined data set. Did the authors test if the factor constraining was really necessary in cPMF? How did they determine if a factor had to be constrained in cPMF?*

2) *How was it determined if a factor was "mixed"?*

3) *It would be beneficial to emphasise to the reader a bit earlier, that the chose solution for single instrument PMF was based on the considerations done for the two previous studies and that also the reference spectra are related to that*

4) *Were the same AMS and EESI-TOF instruments used in both studies? Do the authors expect changes in the instrument behaviour (e.g. change in Pieber effect for AMS)? The change in the NO+ and NO2+ ratio for the AMS calibration with Ammonium Nitrate seems to suggest some instrumental differences*

**Response:**

Thanks for this comment. We address the issue raised point-by-point:

1. We attempted both unconstrained and constrained runs, empirically arriving at the constraint strategy described herein. Although we have only investigated two datasets, our results suggest that factors appearing only in one instrument (here HOA, InorgNit) are difficult to separate without constraints. This may be because their contribution to $Q$ is reduced relative to the single instrument case, similar to previous studies showing that low-mass factors are difficult to resolve accurately from unconstrained PMF (Ulbrich et al., 2009). For InorgNit and HOA, specifically, these factors often require constraints in standalone AMS PMF; with no additional tracers for these factors coming from the EESI-TOF but other unrelated variables added in, it would be very surprising if constraints were not required. COA is a more complex case, and whether or not it requires constraints is likely to rely on the characteristics of the specific dataset (e.g., fractional contribution to total signal, degree of temporal correlation with other factors

2. A factor is determined as mixed in any of the following cases: 1) the mass spectrum from one and/or both instrument in factor shows chemical evidence of influences from other sources; 2) increasing the

number of factors, splits a single factor into two factors with similar mass spectral features but different temporal features (e.g., the several MABB factors in Qi et al. (2019)); and/or 3) inconsistency between the AMS and EESI-TOF components of the factor profile.

3. This part is moved to Text S2, and we emphasise this point in the Text S2 as:

To determine the $F^*_{overlap}$, the EESI-TOF-only PMF was re-run on only the period when both AMS and EESI-TOF were operating based on the same configuration and mass spectra in Stefenelli et al. (2019) and Qi et al. (2019)

4. Although the same instruments were used in the two studies, measurements were conducted approximately half a year apart, so there is no guarantee of identical instrument tuning and performance. Further, the EESI-TOF used a different working solution in the two studies: 1:1 water:methanol in summer and 1:1 water:acetonitrile in winter. As a result, all analysis and corrections were performed separately for the two campaigns. We clarify this for the Pieber effect (Pieber et al., 2016), specifically, as follows (P5 L26-28):

The contribution of nitrate ions to $CO_2^+$ was estimated separately in each campaign from their respective $NH_4NO_3$ calibrations.

We further note that the differences in ambient $NO^+/NO_2^+$ values observed in the summer and winter campaigns are not explainable by instrument differences and are attributable instead to $NO_3$ from different inorganic cations, e.g., $NH_4NO_3$ vs. $KNO_3$

**General Comment #2**

*The authors claim that they did not observe any relevant fragmentation or clustering with solvent molecules in the EESI-TOF. I find that hard to believe since other studies reported strong fragmentation for specific ions (Bell et al., 2021). Also, how could it be determined that no clusters with the solvent occurred? Especially with acetonitrile, the danger would be to interpret the N from the solvent molecule as a nitrated organic compound.*

**Response:**

The extent of clustering observed in the EESI-TOF depends strongly on instrument settings, i.e., the collision energy in the quadrupole guides. The settings in these field campaigns appear to have led to relatively energetic collisions, such that clustering with solvent molecules is disfavoured. This is supported by comparisons of the ambient data with selected chamber measurements (e.g., terpene oxidation), where we identified ions giving the largest signal from clusters and verified that they were not significant in the field data. It is possible that some minor signals from cluster ions remain, but they do not appear to greatly influence the data.

Fragmentation due to the transfer of excess energy during ionisaiton has not been observed in the EESI-TOF and is not expected. However, molecular decomposition, possibly from hydrolysis reactions in the ESI droplets, has been observed in the EESI-TOF for certain molecular classes We cannot rule out the possibility that such reactions affect the data, although these are not expected to affect the majority of the signal (Bell et al., 2022).

**General Comment #3**

*I wonder how computationally intensive this is. There are 2 single instrument PMF runs (with bootstrap, rotations etc). Then the exploratory cPMF run. Then at least 2 more bootstrap cPMF to constrain the additional parameters like CEESI (or the 2D-scan for the a values). Noting that two very short dataset were chosen for this proof-of-concept study, I really wonder how feasible this is for a 2 or 3-month long campaign data set.*

**Response:**

This is an excellent point, and we agree that the computational costs are worth considering. For our study, the cPMF on summer data and winter data 1) for 2D-scan costed us 3 days and 1 week, respectively, and 2) for bootstrap costed about 1 and 2 weeks, respectively. Nevertheless, we consider the cPMF outputs to be of sufficiently high value to be worth the computational investment. In terms of dataset size, SoFi/ME-2 are capable of handling datasets of the scale mentioned by the reviewer, although of course the computational time will be considerable. With that said, we would strongly support efforts to make PMF solvers faster and/or more compatible with large datasets. We also note that the trend of increasing chemical resolution in atmospheric measurements (and thus dataset size/complex) suggests that the need for such software improvements will become increasingly urgent in the future.

**General Comment #4**

*The authors show that for each data set a new set of parameters (a- constraints, CEESI, ASk) have to be determined. However, it would be good to really clearly state that once more in the conclusions. Lately I have come across publication which took data set specific parameters from older studies (e.g., for instrument calibrations) and blindly applied them to their data sets. While that is of course the shortcoming of these people and not the original authors, we can try to emphasise that it is really the method that is being presented and that every new data set needs its own careful exploration of the parameter space to provide reliable data interpretation.*

**Response:**

Thanks for this suggestion. We state it once again in the second paragraph of the conclusion part in P34 L12:

Note that while these methods provide a general procedure for cPMF analysis, the specific parameters employed (i.e., the number of factors ($p$), instrument weighting parameter ($C_{inst}$), and the factors to be constrained and the tightness of constraints ($a$ value ranges)) are dataset-specific and should be determined independently for each new analysis.

**General Comment #5**

*This is just a suggestion: The manuscript is very long with a lot of important details about how to conduct this new version of PMF analysis. A long manuscript is not per se a problem – _especially if a "new" _analysis method is introduced. But the authors should consider if some of the more technical aspects could be shifted to the Supplement Material – _or be presented as an Appendix. That would put a bit more emphasis on the interpretation of the cPMF results like the changes in factor contributions depending on the apparent sensitivity, which is currently a little bit hidden under all the (important) technical details.*

**Response:**

We appreciate the reviewer's suggestion. We have chosen to keep section 2.3, which describes the cPMF method in the main text, as the introduction of this method constitutes a central advance of the paper. However, we have moved the (old) section 3.1, which presented the details of the application of the cPMF to the present datasets (e.g., discussion of individual instrument solutions, construction of dataset-specific profiles, exploration of the solution space in terms of $C$ and $p$, selection of the base case) to the supplement as section S2. We have also moved the discussion of the method used to organonitrate contributions to $NO^+$ and $NO_2^+$ (old section 3.2) to the SI as section S3.

**Specific Comment #1**

*P4 L18f: in addition to the PMF studies Lee et al. (2014) could be mentioned here as another study investigating the sources and formation processes of OA. They used FIGAERO-CIMS spectra obtained in dedicated chamber experiments to reconstruct ambient measurements as linear combinations of them.*

**Response:**

We have added the results of Lee et al. (2020) as follows P3 L18):

Another source apportionment study from Lee et al. (2020) using FIGAERO-CIMS spectra successfully distinguished ambient SOA formation and ageing pathways in two forested regions.

**Specific Comment #2**

*P4 L38: typo: wind speed (WD) -> (WS)*

**Response:**

Now the typo is corrected in P4 L38:

Gas-phase species, e.g., nitrogen dioxide ($NO_2$), nitrogen oxide (NO) and sulfur dioxide ($SO_2$) and meteorological data, e.g., temperature (T), relative humidity (RH), radiation, wind speed (WS) and wind direction (WD) are recorded by the monitoring station.

**Specific Comment #3**

*P5 L14: typo: "by the massed base method" -> with the mass based method*

**Response:**

Now the typo is corrected in P5 L14:

At the beginning and end of the both campaigns, the instrument was calibrated for ionisation efficiency (IE) using 400 nm $NH_4NO_3$ particles using the mass-based method (Jimenez et al., 2003; Canagaratna et al., 2007).

**Specific Comment #4**

*P6 L9: What was the schedule for the switching between direct sampling and background measurements? I.e. how long between each background measurement? Were there strong changes observed between adjacent background measurements?*

**Response:**

The detailed operation and evaluation of instrument performance is discussed in Stefenelli et al. (2019) and Qi et al. (2019). To avoid making the paper too lengthy, we include information about the measurement cycle here but refer the reader to the original manuscripts for more detailed discussion. During both campaigns the EESI-TOF alternated between direct sampling for 8 min and background sampling for 3 min. Adjacent background periods were similar. The transition period from direct sampling to background sampling and from the background sampling to direct sampling was excluded in both campaigns (Qi et al., 2019; Stefenelli et al., 2019). The revised text reads P6 L15:

The EESI-TOF alternates between direct sampling (8 min) and sampling through a particle filter (3 min) to provide a measurement of instrument background (including spray). No major changes between adjacent background measurements were observed in either campaign (Qi et al., 2019; Stefenelli et al., 2019).

**Specific Comment #5**

*P6 L25ff: For AMS PMF analysis, signals with SNR = 0.2 – 2 are usually considered as "weak" and downweighed by a factor of 2-3. Only signals with SNR < 0.2 are removed. Why did the authors choose to remove signals with SNR <2?*

**Response:**

For the AMS, ions with low SNR were indeed treated as suggested by the reviewer, consistent with the recommendations of Paatero and Hopke (2003). This information has been added to the manuscript as follows (P5 L24):

Ions with signal-to-noise ratio (SNR) smaller than 0.2 were excluded in the further analysis, whereas ions with an SNR between 0.2 and 2 were downweighted by a factor of 2 (Paatero and Hopke, 2003)

For the EESI-TOF, we have empirically found that "weak" ions can be affected by fluctuations in the electrospray and/or instrumental background. The more strict SNR threshold was chosen in Stefenelli et al. (2019) and Qi et al. (2019) to compensate for this, and we retain their approach for cross-comparability here.

**Specific Comment #6**

*P7 L23 & Fig 10: The authors do not explain anything about the two chosen methods (GBR vs LRR). Later in section 3.3 they also do not provide any more information about what these two methods mean, why they were used, or what the different values mean. As the paper is already very long as it is and only a qualitative comparison is conducted anyway, I recommend that the authors decide which of the*

*methods they want to use here. To me the trends seem identical with just a general offset between LRR and GBR.*

**Response:**

We agree with the reviewer that the performance of the two models is similar, with the main difference being an offset. For simplicity, we have removed the LRR results and include only the GBR, because it is expected to perform better at handling possible interactions in the feature space.

The updated Figure 10 (now Figure 8) and the caption is shown below:

[Figure]

Figure 8. The estimated relative apparent sensitivity to COA ($AS_k/\overline{AS_{\mathrm{COA_C}}}$) from the gradient boosting regression (GBR) model as a function of cPMF-derived $AS_k/\overline{AS_{\mathrm{COA_C}}}$). The symbols indicate the different oxidation-precursor system (LMN for SOA produced from oxidation of limonene by ozone, cresol and TMB for SOA produced from oxidation of *o*-cresol and 1,3,5-trimethylbenzene by OH radicals, respectively).

**Specific Comment #7**

*P15&16: Here the similarities of the single instrument PMF with the new data set are described and compared to the original ones (although only for AMS). The information presented here is not really used in the rest of the manuscript. This could be a section that could be moved to the supplement and replaced by a sentence stating that the new factors are similar enough to the old ones so that all the interpretations from the previous studies are still valid (see SI for details)*

**Response:**

We have moved this part of presentation and corresponding discussion to Text S2.1, including the figures comparing the result from this conventional PMF to the published result in Stefenelli et al. (2019) and Qi et al. (2019). This discussion is referenced in the manuscript as (P15, L32):

We re-ran the conventional PMF on the summer and the winter data, obtaining results similar to Stefenelli et al. (2019) and Qi et al. (2019), as discussed in Text S2 in the supplement.

**Specific Comment #8**

*Table 1: out of curiosity, how many "common ions" are there in the two EESI-TOF data sets? I.e., how many of the 892 ions identified for the winter data were also present in the summer data?*

**Response:**

There are 257 ions that are both found in PMF input matrices for both the summer and winter datasets. Here we attach the common ion list.

| m/z | ion identity | m/z | ion identity | m/z | ion identity |
|---|---|---|---|---|---|
| 148.07327 | C7H11ONNa | 227.12538 | C10H20O4Na | 282.05841 | C10H13O7NNa |
| 150.05255 | C6H9O2NNa | 228.08424 | C8H15O5NNa | 282.1312 | C12H21O5NNa |
| 151.07295 | C7H12O2Na | 229.03188 | C7H10O7Na | 283.07883 | C11H16O7Na |
| 152.03181 | C5H7O3NNa | 229.06825 | C8H14O6Na | 283.1152 | C12H20O6Na |
| 153.05222 | C6H10O3Na | 229.10464 | C9H18O5Na | 283.15158 | C13H24O5Na |
| 155.03148 | C5H8O4Na | 230.02711 | C6H9O7NNa | 283.18799 | C14H28O4Na |
| 157.01074 | C4H6O5Na | 233.07843 | C11H14O4Na | 284.07407 | C10H15O7NNa |
| 157.04713 | C5H10O4Na | 234.11006 | C11H17O3NNa | 284.18323 | C13H27O4NNa |
| 163.03656 | C7H8O3Na | 234.14645 | C12H21O2NNa | 285.05807 | C10H14O8Na |
| 163.07295 | C8H12O2Na | 235.09409 | C11H16O4Na | 285.09448 | C11H18O7Na |
| 163.12297 | C10H15N2 | 236.08932 | C10H15O4NNa | 285.13086 | C12H22O6Na |
| 165.05222 | C7H10O3Na | 238.10498 | C10H17O4NNa | 286.05334 | C9H13O8NNa |
| 166.04745 | C6H9O3NNa | 238.14136 | C11H21O3NNa | 286.08972 | C10H17O7NNa |
| 166.08385 | C7H13O2NNa | 239.05261 | C9H12O6Na | 286.1261 | C11H21O6NNa |
| 167.03148 | C6H8O4Na | 239.08899 | C10H16O5Na | 287.16177 | C16H24O3Na |
| 167.06786 | C7H12O3Na | 240.12064 | C10H19O4NNa | 289.17743 | C16H26O3Na |
| 169.04713 | C6H10O4Na | 241.03188 | C8H10O7Na | 290.17267 | C15H25O3NNa |
| 170.04237 | C5H9O4NNa | 241.06825 | C9H14O6Na | 291.19308 | C16H28O3Na |
| 170.07877 | C6H13O3NNa | 241.10464 | C10H18O5Na | 292.15192 | C14H23O4NNa |
| 176.06821 | C8H11O2NNa | 242.02711 | C7H9O7NNa | 293.13593 | C14H22O5Na |
| 177.05222 | C8H10O3Na | 242.09988 | C9H17O5NNa | 293.17233 | C15H26O4Na |
| 177.08859 | C9H14O2Na | 243.04752 | C8H12O7Na | 293.20871 | C16H30O3Na |
| 178.08385 | C8H13O2NNa | 243.08391 | C9H16O6Na | 294.09482 | C12H17O6NNa |
| 179.06786 | C8H12O3Na | 243.12029 | C10H20O5Na | 294.1312 | C13H21O5NNa |
| 179.10425 | C9H16O2Na | 244.04277 | C7H11O7NNa | 294.20395 | C15H29O3NNa |
| 180.06311 | C7H11O3NNa | 244.07916 | C8H15O6NNa | 295.1152 | C13H20O6Na |

| | | | | | |
|---|---|---|---|---|---|
| 181.04713 | C7H10O4Na | 244.11554 | C9H19O5NNa | 295.15158 | C14H24O5Na |
| 181.08353 | C8H14O3Na | 245.11482 | C13H18O3Na | 295.18799 | C15H28O4Na |
| 182.07877 | C7H13O3NNa | 245.1512 | C14H22O2Na | 296.14685 | C13H23O5NNa |
| 183.09917 | C8H16O3Na | 247.09409 | C12H16O4Na | 297.09448 | C12H18O7Na |
| 184.05803 | C6H11O4NNa | 247.16685 | C14H24O2Na | 297.13086 | C13H22O6Na |
| 185.04204 | C6H10O5Na | 248.08932 | C11H15O4NNa | 297.16724 | C14H26O5Na |
| 185.07843 | C7H14O4Na | 248.12572 | C12H19O3NNa | 297.20364 | C15H30O4Na |
| 187.05769 | C6H12O5Na | 249.10973 | C12H18O4Na | 298.1261 | C12H21O6NNa |
| 189.08859 | C10H14O2Na | 249.14612 | C13H22O3Na | 298.16248 | C13H25O5NNa |
| 190.08385 | C9H13O2NNa | 251.05261 | C10H12O6Na | 299.11011 | C12H20O7Na |
| 191.10425 | C10H16O2Na | 251.08899 | C11H16O5Na | 299.18289 | C14H28O5Na |
| 192.0995 | C9H15O2NNa | 251.12538 | C12H20O4Na | 300.069 | C10H15O8NNa |
| 193.04713 | C8H10O4Na | 252.04787 | C9H11O6NNa | 300.10538 | C11H19O7NNa |
| 193.08353 | C9H14O3Na | 253.03188 | C9H10O7Na | 300.14175 | C12H23O6NNa |
| 194.04237 | C7H9O4NNa | 253.06825 | C10H14O6Na | 301.05301 | C10H14O9Na |
| 195.02638 | C7H8O5Na | 253.10464 | C11H18O5Na | 302.08463 | C10H17O8NNa |
| 195.06277 | C8H12O4Na | 253.17741 | C13H26O3Na | 302.12103 | C11H21O7NNa |
| 195.09917 | C9H16O3Na | 254.13628 | C11H21O4NNa | 303.06866 | C10H16O9Na |
| 197.00566 | C6H6O6Na | 255.04752 | C9H12O7Na | 303.10504 | C11H20O8Na |
| 197.04204 | C7H10O5Na | 255.08391 | C10H16O6Na | 303.14142 | C12H24O7Na |
| 197.07843 | C8H14O4Na | 255.12029 | C11H20O5Na | 304.10028 | C10H19O8NNa |
| 198.07368 | C7H13O4NNa | 256.07916 | C9H15O6NNa | 304.13666 | C11H23O7NNa |
| 199.05769 | C7H12O5Na | 256.11554 | C10H19O5NNa | 307.1152 | C14H20O6Na |
| 199.09409 | C8H16O4Na | 257.06317 | C9H14O7Na | 307.15158 | C15H24O5Na |
| 200.01656 | C5H7O6NNa | 257.09955 | C10H18O6Na | 307.18799 | C16H28O4Na |
| 200.05293 | C6H11O5NNa | 258.05841 | C8H13O7NNa | 308.18323 | C15H27O4NNa |
| 200.08932 | C7H15O4NNa | 258.09482 | C9H17O6NNa | 309.13086 | C14H22O6Na |
| 201.03696 | C6H10O6Na | 258.1312 | C10H21O5NNa | 310.16248 | C14H25O5NNa |
| 203.01622 | C5H8O7Na | 261.09448 | C9H18O7Na | 311.14651 | C14H24O6Na |
| 203.05261 | C6H12O6Na | 265.14102 | C13H22O4Na | 313.08939 | C12H18O8Na |
| 204.06311 | C9H11O3NNa | 266.06351 | C10H13O6NNa | 313.12576 | C13H22O7Na |
| 205.08353 | C10H14O3Na | 267.04752 | C10H12O7Na | 313.16217 | C14H26O6Na |
| 206.07877 | C9H13O3NNa | 267.1203 | C12H20O5Na | 314.19379 | C14H29O5NNa |
| 206.11514 | C10H17O2NNa | 267.15668 | C13H24O4Na | 317.17233 | C17H26O4Na |
| 207.02638 | C8H8O5Na | 268.07916 | C10H15O6NNa | 321.13086 | C15H22O6Na |
| 207.06277 | C9H12O4Na | 268.18832 | C13H27O3NNa | 321.16724 | C16H26O5Na |
| 207.09917 | C10H16O3Na | 269.02679 | C9H10O8Na | 321.20364 | C17H30O4Na |
| 208.09441 | C9H15O3NNa | 269.06317 | C10H14O7Na | 325.08939 | C13H18O8Na |
| 209.04204 | C8H10O5Na | 269.09955 | C11H18O6Na | 325.12576 | C14H22O7Na |
| 209.07843 | C9H14O4Na | 269.13593 | C12H22O5Na | 325.16217 | C15H26O6Na |
| 209.11482 | C10H18O3Na | 270.05841 | C9H13O7NNa | 325.19855 | C16H30O5Na |
| 210.11006 | C9H17O3NNa | 270.09482 | C10H17O6NNa | 325.23492 | C17H34O4Na |
| 211.05769 | C8H12O5Na | 271.07883 | C10H16O7Na | 326.15741 | C14H25O6NNa |

| | | | | | |
|---|---|---|---|---|---|
| 211.09409 | C9H16O4Na | 271.1152 | C11H20O6Na | 326.19379 | C15H29O5NNa |
| 215.05261 | C7H12O6Na | 271.15158 | C12H24O5Na | 329.24509 | C20H34O2Na |
| 215.08899 | C8H16O5Na | 273.1825 | C16H26O2Na | 336.17813 | C16H27O5NNa |
| 217.06825 | C7H14O6Na | 275.12537 | C14H20O4Na | 337.16217 | C16H26O6Na |
| 220.05803 | C9H11O4NNa | 275.16177 | C15H24O3Na | 337.19855 | C17H30O5Na |
| 220.09441 | C10H15O3NNa | 276.12064 | C13H19O4NNa | 337.23492 | C18H34O4Na |
| 221.07843 | C10H14O4Na | 276.15701 | C14H23O3NNa | 339.2142 | C17H32O5Na |
| 223.02132 | C8H8O6Na | 277.17743 | C15H26O3Na | 340.17307 | C15H27O6NNa |
| 223.05769 | C9H12O5Na | 279.08392 | C12H16O6Na | 340.24582 | C17H35O4NNa |
| 223.09409 | C10H16O4Na | 279.1203 | C13H20O5Na | 345.20364 | C19H30O4Na |
| 224.08932 | C9H15O4NNa | 279.15668 | C14H24O4Na | 345.24002 | C20H34O3Na |
| 225.03696 | C8H10O6Na | 279.19308 | C15H28O3Na | 345.2764 | C21H38O2Na |
| 225.07333 | C9H14O5Na | 280.07916 | C11H15O6NNa | 349.16217 | C17H26O6Na |
| 225.10973 | C10H18O4Na | 280.15192 | C13H23O4NNa | 351.2142 | C18H32O5Na |
| 226.10498 | C9H17O4NNa | 281.09955 | C12H18O6Na | 373.23492 | C21H34O4Na |
| 227.05261 | C8H12O6Na | 281.13593 | C13H22O5Na | 389.26624 | C22H38O4Na |
| 227.08899 | C9H16O5Na | 281.17233 | C14H26O4Na | | |

**Specific Comment #9**

*P18 L22: It is not clear to how the authors handled the HOA and InorgNit factor reference spectra for the EESI-TOF part of the combined dataset. What was the "same intensity" to which all ions were set? Why was 0.01 cps (ug m-3)-1 chosen? Why did they not just set the values to 0 (or a very small number, e.g., 1e-6)?*

**Response:**

The approach taken in this paper actually aligns with the reviewer's suggestion. We found that setting the values to 0 caused instabilities in the ME-2 solver, for reasons unknown. Therefore, to propose a generalised strategy that can be applied regardless of differences in the measurement units between instruments, we based this "same intensity" on the factor sensitivity, $AS_k$. This is based on Eq. 11, repeated here for clarity:

$$\frac{\left(f_{k,j}\right)_{j=all,ref}}{1\ \mu g\ m^{-3}} = \begin{cases} \dfrac{\left(f_{k,j}\right)_j}{\sum_j\left(f_{k,j}\right)_j}, & j \in AMS, ref \\[3mm] AS_k \cdot \dfrac{\left(f_{k,j}\right)_j}{\sum_j\left(f_{k,j}\right)_j}, & j \in EESI, ref \end{cases} \tag{11}$$

For the case where the EESI-TOF is insensitive to the profile, we calculate the profile as follows:

$$\frac{\left(f_{k,j}\right)_{j=all,ref}}{1\ \mu g\ m^{-3}} = \begin{cases} \dfrac{\left(f_{k,j}\right)_j}{\sum_j\left(f_{k,j}\right)_j}, & j \in AMS, ref \\[3mm] AS_k \cdot \dfrac{1/n_{EESI}}{\sum_j\left(f_{k,j}\right)_j}, & j \in EESI, ref \end{cases} \tag{S1}$$

Here $n_{EESI}$ denotes the number of ions in the EESI-TOF dataset and assume $AS_k = 0.01$ cps (ug m$^{-3}$). This value is chosen to reliably yield and appropriately small numbers (again, regardless of instrument measurement units) while avoiding the 0-based solver instability. This value for $AS_k$ is approximately 4-5 orders of magnitude lower than the lowest factor sensitivities, and thus implies species well below the EESI-TOF detection limit. This latter equation has been added to the SI, as Text S1 (Eq. S1).

We refer to the new text in the manuscript as follows (P12, L13):

In the case that a factor is undetectable by the EESI-TOF (e.g., non-oxygenated hydrocarbons comprising traffic-related factors), a value of $AS_k$ is assumed that fixes the EESI-TOF contribution near zero, as discussed in the Supplement in Text S1.

The new Supplement section (Text S1) is as follows:

**Text S1. Profile construction for factors to which the EESI-TOF is insensitive**

In the Sect. 2.3.2, Eq. (11) proposes a generalised strategy for constructing reference factor profiles, that can be applied regardless of differences in the measurement units between instruments. Here we discuss the special case of a factor measured by the AMS but to which the EESI-TOF is insensitive, In this case, all variables in the EESI-TOF component of the profile are set to a low value based on an assumed $AS_k = 0.01$ cps (ug m$^{-3}$), which is orders of magnitude lower than the $AS_k$ of detectable factors. This approach is preferred to simply setting the EESI-TOF variables to zero, as this was empirically observed to create instabilities in the ME-2 solver. The full profile is then calculated as follows:

$$\frac{(f_{k,j})_{j=all,ref}}{1\ \mu g\ m^{-3}} = \begin{cases} \dfrac{(f_{k,j})_j}{\Sigma_j(f_{k,j})_j}, & j \in AMS, ref \\[2ex] AS_k \cdot \dfrac{1/n_{EESI}}{\Sigma_j(f_{k,j})_j}, & j \in EESI, ref \end{cases} \tag{S1}$$

Here $n_{EESI}$ denotes the number of ions in the EESI-TOF dataset and as noted above we assume $AS_k = 0.01$ cps (ug m$^{-3}$).

**Specific Comment #10**

*P19: where the factor profiles constrained for the $C_{EESI}$ analysis in section 3.1.3?*

**Response:**

Note that this comment refers to section 3.1.3, which is now moved to the supplement as Text S2.3. Yes, factor profiles were constrained during the $C_{EESI}$ analysis, as noted in the original text ("the $a$ values of all constrained factor profiles were set to zero during this initial exploration"). We now additionally present the anchor profiles for all constrained factors in the new Fig. S5.

[Figure]

Figure S5. Normalised reference factor profiles for all constrained factors in (a) summer and (b) winter, coloured by different ion families.

**Specific Comment #11**

*P19 L29f: The authors state that the summer data follows the "expected" trend of the overlap values with $C_{EESI}$. But why is that trend expected? And what does it mean that the winter data does not follow the same trend (i.e. having the low values for $p=7$ and $C_{EESI} = 0.5$)? Could it be connected to the change in EESI background due to using a different solvent?*

**Response:**

Because $C_{EESI}$ is a weighting parameter, the expectation is that the residuals ($e_{ij}/s_{ij}$) of the EESI-TOF monotonically decrease with increasing $C_{EESI}$, while the AMS residuals, if they are perturbed at all, would move in the opposite direction. This means that for a given $p$, $F^*_{overlap}$ is expected to monotonically decrease with increasing $C_{EESI}$, reach a minimum at some dataset-specific value of $C_{EESI}$, and then monotonically increase with increasing $C_{EESI}$. Such

behaviour is observed for summer (Fig. S6a). However, Fig. S6b shows 2 local minima for $F^*_{\text{overlap}}$ as a function of $p$ and $C_{EESI}$. The precise reason for this is unknown but is likely related to the general complexities of the PMF solution space, in which the possibility of multiple local minima is well-established.

**Specific Comment #12**

*P20: Do the authors have any idea why the $C_{EESI}$ value was so different for the two data set? I assume the AMS sensitivity can be considered as constant. Which changes may have occurred to the EESI to cause this value to change by 2 orders of magnitude (2 -> 0.05)?*

**Response:**

The reasons for this are not entirely clear, but likely relate to one or more of the following: 1) different instrument configurations, which can change the sensitivity of EESI-TOF to different ions, 2) different solvents used in two campaigns, which causes the different background value, and 3) different numbers of ions resolved by EESI-TOF.

**Specific Comment #13**

*P26 L9: Will the world may agree on the time for lunch, dinner time is very culture specific (just ask an Italian and a Finn ;-). It may be more specific to give the time of day in the description of the COA factor and then clarify that those times correspond to lunch and dinner in Switzerland.*

**Response:**

We agree, and now note that the lunch time corresponds to approximately 11:30-13:30 and dinner to 18:30-20:30 in the manuscript.

**Specific Comment #14**

*P26 & Fig S28, S29: It is very good that the authors provide the factor mass spectra for comparison. But the chosen visualisation may not be the best. As this is HR data, sticks may overlap and not be distinguishable (especially for EESI-TOF). The authors could try using a modified Kroll diagram (OSC vs carbon number) with showing the signal intensity of the base case as the size of the symbols. Then a colour code could be used to show the difference to the compared factor. That would also highlight if certain groups of compounds are different (e.g. dimers).*

**Response:**

We thank the referee for this suggestion.

We have added Kroll diagrams for Figure S32 and Figure S33 (previous Fig. S29 and S30), as the referee suggested. We also add scatter plots of the PMF vs. cPMF factors where 1:1 correspondence can be established. These are shown below.

We also show below the modified Kroll diagrams suggested by the reviewer (i.e., marker size as base case signal, color as difference to compared factor). Although interesting, we find the side-by-side Kroll diagrams simpler to understand in illustrating the strong similarity between corresponding factors and use these in the supplement.

[Figure]

[Figure]

Figure S32. Comparison of four summer SOA factors (DaySOA1$_{S,E}$, DaySOA2$_{S,E}$, NightSOA1$_{S,E}$ and NightSOA2$_{S,E}$) resolved from EESI-TOF-only PMF analysis to the corresponding factors (DaySOA1$_{S,C}$, DaySOA2$_{S,C}$, NightSOA1$_{S,C}$ and NightSOA2$_{S,C}$) resolved from the combined PMF analysis, shown in a), b), c) and d), respectively. Each subfigure contains the direct comparison of corresponding factors, and modified Kroll diagram sized by the ion intensities of the corresponding factor.

[Figure]

Figure S33. Comparison of two LABB_W,C factors resolved from combined dataset in Zurich winter. Direct EESI part mass spectra comparison and modified Kroll diagram sized by the ion intensities are shown in a) and direct AMS part mass spectra comparison and modified Kroll diagram sized by the ion intensities without $NO^+$ and $NO_2^+$ are shown in b).

Figures suggested by the referee but not included in the manuscript.

[Figure]

We also make the comparison between two LABB factors.

[Figure]

**Specific Comment #15**

*P27 L14. Hereafter the notation "Sum_BB" etc is used. It is clear what the authors mean, but it may enhance readability if the Greek Σ were used instead of Sum_ (e.g., Σ BB). Especially in Fig 7 and Fig 10, this would highlight at these values are something different from the after bars/markers.*

**Response:**
We agree, and have modified the manuscript accordingly. Fig 7, Fig 9 and Fig 10 (now Figs. 5, 7, and 8) are also updated, as shown below.

[Figure]

Figure 5. Comparison of $AS_k/\overline{AS_{COA_C}}$ of different factors resolved from the cPMF on the summer and winter datasets. Mean values are shown as bars, and error bars indicate the standard deviation over all accepted bootstrap runs. The following factor aggregations are also shown: $\Sigma BB_{W,C}$ = $MABB_{W,C}$ + $LABB_{W,C}$ + $NitOA1_{W,C}$ + $NitOA2_{W,C}$ + $EVENT_{W,C}$; $\Sigma DaySOAs_{S,C}$ = $DaySOA1_{S,C}$ + $DaySOA2_{S\_C}$; and $\Sigma NightSOAs_{S,C}$ = $NightSOA1_{S,C}$ + $NightSOA2_{S,C}$.

[Figure]

Figure 7. $AS_k/\overline{AS_{COA_C}}$ of SOA factors retrieved from the summer and winter datasets as a function of the H:C ratio. Error bars denote standard deviation across all accepted runs. Spearman correlation is 0.833, as indicated in the top-left corner.

[Figure]

Figure 8. The estimated relative apparent sensitivity to COA ($AS_k/\overline{AS_{COA_C}}$) from the gradient boosting regression (GBR) model as a function of cPMF-derived $AS_k/\overline{AS_{COA_C}}$). The symbols indicate the different oxidation-precursor system (LMN for SOA produced from oxidation of limonene by ozone, cresol and TMB for SOA produced from oxidation of *o*-cresol and 1,3,5-trimethylbenzene by OH radicals, respectively).

**Specific Comment #16**

*P29 L22: It feels a bit odd that one of the arguments to constrain individual factor profiles was to prevent factor splitting. But now the authors recombine two factors which they deem to have been split artificially. Since I am not a fan of overly constrained PMF, I do not want to recommend constraining LABB. But it makes me wonder if using the same constraints in cPMF as in single PMF is the right approach here.*

**Response:**

There are three issues raised here, which we address separately (1) general motivation for applying factor constraints; (2) recombination of LABB factors; (3) relationship between cPMF and individual PMF constraints.

We are not sure from where the referee gained the impression that avoiding factor splitting was a motivation for constraining factor profiles, but would be happy to clarify the misleading text if identified. Indeed, constraints do not address factor splitting but rather the common problem of rotational ambiguity leading to mixed and/or unresolvable factors, as noted in the first paragraph of section 2.3.2.

In P20 L21, we state: "Because no significant chemical differences are apparent between LABB1$_{W,C}$ and LABB2$_{W,C}$, they are aggregated to a single LABB$_{W,C}$ for presentation." This practice of factor recombination is a well-known technique in unconstrained PMF, and indeed is performed here on unconstrained factors. It is not related to the use or lack thereof of constraints on other factors. As a result, we are unclear regarding the referee's concern.

The question of using the same constraints in single PMF vs. cPMF is an excellent point, and one that we fully agree deserves more investigation. There are two questions: (1) to what extent do factors of the "same type" (e.g., cooking-related aerosol) from individual PMF analyses by separate instruments represent the same aerosol fraction; and (2) even assuming they represent identical fractions, how to construct the joint profile in the current case when uncertainties in the quantification of one or both instruments exist. We attempt to address (1) by varying $a$ values within the bootstrap analysis. This allows the cPMF to adapt to the case that the two instruments describe similar-but-not-quite-identical fractions of the aerosol. We consider this theoretically robust, although practical tests on additional datasets and/or synthetic data would be illuminating. Regarding (2), we now suggest that in future work, variation of the assumed $AS_k$ values used to construct the reference profiles (Eq. 11) could be varied, as follows (P14, L21):

Since the constrained factors use reference profiles constructed with an estimated $AS_k$ (see Eq. (11)), this combined bootstrap/constraint analysis allows recalculation of $AS_k$ within PMF for any factor with a non-zero $a$ value. As a result, the final reported solution is the average of all accepted bootstrap runs, with uncertainties in factor profiles and time series taken as the standard deviation. To minimise the effect of estimated $AS_k$ on constrained factors, we suggest that in the future this method could be improved by initialisation of constrained factor profiles with randomised $AS_k$ within a predefined range, in conjunction with the existing $a$-value/bootstrap routine.

**Specific Comment #17**

*P30 L 1 & 29: The authors have used correlations for almost everything to express similarity. But now for the factors they use "qualitatively similar". What does that mean here? Is there a reason why the authors do not use a mathematical measure for the similarity of their factors (e.g., uncentered R or contrast angle)? How can they be qualitatively similar if one contains ON and the other does not?*

**Response:**

In P30 L1, we refer to the presence of a characteristic set of features in the HOA profile, specifically the large contribution from the $C_nH_{2n+1}^+$, and $C_nH_{2n-1}^+$ series, consistent with *n*-alkanes and branched alkanes. This is now clarified in the manuscript (P21 L1) as:

This factor is dominated by the $C_nH_{2n+1}^+$, and $C_nH_{2n-1}^+$ series, consistent with *n*-alkanes and branched alkanes, with lower $CO^+$ and $CO_2^+$ content than the HOA$_{S,C}$. The HOA$_{W,C}$ time series correlates strongly with HOA$_{W,A}$ ($r^2$ of 0.913).

In P30 L29, the comment on profile similarity between SOA1$_{W,C}$ and SOA2$_{W,C}$ was included mistakenly, and has been removed.

**Specific Comment #18**

*P30 L 1ff: Also, how meaningful is it to compare the factor mass spectrum of the constrained factors here? How much could e.g. HOA really vary with the set constraints? E.g. how big are the differences of HOA for summer and winter?*

**Response:**

All cPMF factor profiles are taken as the average of the bootstrap/$a$-value randomization results. For $HOA_{W,C}$, this includes $a$ values as high as 0.9. As such, this is a loose constraint (recall $a = 1$ is unconstrained) and comparison of the profiles is meaningful.

**Specific Comment #19**

*P30 L 9 "consistent with other studies" If the authors use the PMF factor from Qi et al. to create a constrained factor, they cannot claim that this factor than is in good agreement with Qi et al. (unless the constraint was so loose that a strong variation was truly possible)?*

**Response:**

We agree, and have rephrased the text as follows (P21 L8):

"These are key features of the constrained reference profile ($0 \leq a \leq 0.3$) (Qi et al., 2019) and COA factors found in other studies (Stefenelli et al, 2019; Tong et al., 2021)."

**Specific Comment #20**

*Section 3.3: I find it difficult to follow in this section when the authors are talking about the Fig S32 and S33 and when they are using the "scaled" values in Fig 34 in their arguments. I also do not understand what the benefit is of Fig 34. The spread of the $AS_k$ values are already visible in S32&33. For me the additional scaling was more confusing than enlightening.*

**Response:**

We have deleted Figure S34 and now solely use the other figures to discuss the spread of $AS_k$ values.

**Specific Comment #21**

*p33 L43: how is COAs,c multimodal in Fig S32?.*

**Response:** Although the overall width of the distribution of $COA_{S,C}$ is relatively narrow compared to most other factors, a close inspection indicates that is comprised of ~3 discrete peaks, centered at $AS_k = $ ~510 cps/($\mu$g m$^{-3}$) (around 1 in Figure S35). Similar results are observed for $COA_{W,C}$. This has been clarified in the text as follows (P24 L44):

"Interestingly, the distribution of the sensitivities, of $COA_{S,C}$, $COA_{W,C}$, and $CSOA_{W,C}$ in Figs. S32 and S33 is clearly multi-modal despite $a$ value constraints (although the overall $COA_{S,C}$ and $COA_{W,C}$ distributions remain relatively narrow), but the reason for this is unknown."

**Specific Comment #22**

*p 34 L 7ff I find the predictive value of levoglucosane/C6H10O5 very limited. It only works for BB factors in which the ion associated with levoglucosane is not disturbed by anything. If the origine of the factors have not yet been interpreted, this value may be misleading.*

**Response:**

We agree that the $C_6H_{10}O_5$ content is not predictive in a quantitative sense. However, we consider the analysis useful in illustrating the effect that this single ion, which appears with moderate-to-high intensity in several primary factors, exerts on $AS_k$. The statement has been rephrased as follows (P25 L7);

"Therefore, despite the variation in composition of the POA-influenced factors, the effect of the $C_6H_{10}O_5$ content on the overall factor sensitivity is often considerable for cases where this ion is strongly influenced by levoglucosan."

**Specific Comment #23**

*P35 L1f & Fig 9: I find the correlation of H:C and relative ASk rather poor. 5 markers fall into the square H:C 1.58 – 1.66 and AS 1-2 without any clear trend in my eyes. Also, why was H:C chosen? How would this look with O:C or OSc?*

**Response:**

We select H:C for comparison due to two reasons. First, it was identified as a major predictor of molecular sensitivity in the study of Wang et al. (2021), in preference to quantities such as O:C and OSc. Second, the correlation of factor sensitivities with H:C in the present study is much stronger than with O:C or OSc, with Spearman's rank correlation of 0.833, -0.167 and -0.452 for $AS_k$ vs H:C, $AS_k$ vs O:C and $AS_k$ vs OSc, respectively. This has been added to the manuscript as follows (P26 L2):

Consistent with Wang et al. (2021), H:C is found to be a better predictor of $AS_k$ than either O:C or OSc, yielding Spearman's rank correlation of 0.833 for $AS_k$ vs. H:C, -0.167 for $AS_k$ vs. O:C, and -0.452 for $AS_k$ vs. OSc.

The $AS_k$ vs H:C, O:C and OSc plots are shown below:

[Figure]

**Specific Comment #24**

*Section 4: In my opinion the authors could put a bit more emphasis on this part of the manuscript.*
*The implications for the interpretation of the factor contributions for the EESI-TOF data are big while*
*the AMS part as not affected as much. To me, that is an important message for any PMF analysis of*

*instruments which lack detailed sensitivity information (so also FIGAERO-CIMS). Already for one instrument applied at the same location but in two different seasons, such big differences are observed. How careful should we then be when comparing factors obtained for different locations.*

**Response:**

We fully agree that the implications for interpretation of standalone source apportionment results from instruments such as the EESI-TOF and FIGAERO-CIMS are an important outcome of the study, and now highlight this in the Conclusions as follows (P34 L7):

These considerable differences in the source contributions between the uncorrected EESI-TOF and cPMF results highlight the challenges in interpreting standalone source apportionment results for instruments where ion-specific sensitivity information is not readily available, such as EESI-TOF or FIGAERO-CIMS. Although the time trends of such analyses are likely robust, interpretation of the relative composition requires caution. Therefore, if such interpretation is desired, it is advised to employ analysis strategies such as cPMF that are capable of integrating quantitative measurements from reference instruments.

In the present study, we consider the differences between the summer and winter results to be driven by real differences in the factor composition, as discussed in the manuscript. Because of this contrast in factor composition, we do not yet have enough information to assess the consistency of factors across different locations, or consistency of $AS_k$'s for related factors retrieved across different studies. As a result, we cannot comment on this point.

**Specific Comment #25**

*Fig 11-13: Although I would like to see a bit more emphasis on the Atmospheric Implications section, I do think that the authors should decide if they prefer Fig 11 or Fig 13 as they do show the same information. One of them could go to the supplement information. Also Fig 12 is not providing that much exiting visual information. I take from Fig 12 that for AMS pmf factors, the sensitivity does not matter (as could be expected).*

**Response:**

Although there is some overlap in information between Fig.11 (now 9) and Fig. 13 (now 11), the first focuses on temporal variation in composition, while the second relates to the uncertainties in the cPMF results (i.e., differences in composition across solutions). Therefore, we consider both figures to be important (and complementary) and retain both in the main text.

Regarding Fig. 12 (now 10), our cPMF implementation implicitly treats the AMS as the reference instrument. As such, it is very unlikely that the total AMS OA concentration changes, as this would require a large and systematically positive or negative increase in AMS residuals.

**Specific Comment #26**

*P39 Fig 9 The summer and winter symbol are both circles in the legend. I think you need a square for winter.*

**Response:**

The legend has been corrected, and the revised figure is shown in response to RC1SC #15 and #23.

[Figure]

**Specific Comment #27**

*P 42 L13f: "optionally constraining factor profiles for one or both instruments" I did not find the place where a factor was constrained for one of the instruments but not for the other. As I understood the method, that is not possible. One a-value is applied for a factor containing both EESI-TOF and AMS ions. Hence a factor is either constrained (to a certain degree) or not.*

**Response:**

This was not stated clearly in the original text. We intended to refer to the case where a factor is detectable by only one instrument (e.g., our treatment of HOA and InorgNit), as opposed to the case where the factor is detectable by both instruments but has a constrained profile in only one (which the reviewer correctly notes was not addressed). We have clarified and emphasise the text as follows in the Abstract (P1 L28):

"…a method for optionally constraining the profiles of factors that are detectable by one or both instruments,…"

**Reference**

Bell, D. M., Wu, C., Bertrand, A., Graham, E., Schoonbaert, J., Giannoukos, S., Baltensperger, U., Prevot, A. S. H., Riipinen, I., El Haddad, I., and Mohr, C.: Particle-phase processing of α-pinene NO3 secondary organic aerosol in the dark, Atmos. Chem. Phys., 22, 13167-13182, 10.5194/acp-22-13167-2022, 2022.

Canagaratna, M. R., Jayne, J. T., Jimenez, J. L., Allan, J. D., Alfarra, M. R., Zhang, Q., Onasch, T. B., Drewnick, F., Coe, H., Middlebrook, A., Delia, A., Williams, L. R., Trimborn, A. M., Northway, M. J., DeCarlo, P. F., Kolb, C. E., Davidovits, P., and Worsnop, D. R.: Chemical and microphysical characterization of ambient aerosols with the aerodyne aerosol mass spectrometer, Mass Spectrom. Rev., 26, 185-222, https://doi.org/10.1002/mas.20115, 2007.

Jimenez, J. L., Jayne, J. T., Shi, Q., Kolb, C. E., Worsnop, D. R., Yourshaw, I., Seinfeld, J. H., Flagan, R. C., Zhang, X., Smith, K. A., Morris, J. W., and Davidovits, P.: Ambient aerosol sampling using the Aerodyne Aerosol Mass Spectrometer, J. Geophys. Res. Atmos., 108, https://doi.org/10.1029/2001JD001213, 2003.

Lee, B. H., D'Ambro, E. L., Lopez-Hilfiker, F. D., Schobesberger, S., Mohr, C., Zawadowicz, M. A., Liu, J., Shilling, J. E., Hu, W., Palm, B. B., Jimenez, J. L., Hao, L., Virtanen, A., Zhang, H., Goldstein, A. H., Pye, H. O. T., and Thornton, J. A.: Resolving Ambient Organic Aerosol Formation and Aging Pathways with Simultaneous Molecular Composition and Volatility Observations, Acs Earth Space Chem, 4, 391-402, 10.1021/acsearthspacechem.9b00302, 2020.

Paatero, P., and Hopke, P. K.: Discarding or downweighting high-noise variables in factor analytic models, Anal. Chim. Acta, 490, 277-289, https://doi.org/10.1016/S0003-2670(02)01643-4, 2003.

Pieber, S. M., El Haddad, I., Slowik, J. G., Canagaratna, M. R., Jayne, J. T., Platt, S. M., Bozzetti, C., Daellenbach, K. R., Frohlich, R., Vlachou, A., Klein, F., Dommen, J., Miljevic, B., Jimenez, J. L., Worsnop, D. R., Baltensperger, U., and Prevot, A. S. H.: Inorganic Salt Interference on CO2+ in Aerodyne AMS and ACSM Organic Aerosol Composition Studies, Environ. Sci. Technol., 50, 10494-10503, https://doi.org/10.1021/acs.est.6b01035, 2016.

Qi, L., Chen, M. D., Stefenelli, G., Pospisilova, V., Tong, Y. D., Bertrand, A., Hueglin, C., Ge, X. L., Baltensperger, U., Prevot, A. S. H., and Slowik, J. G.: Organic aerosol source apportionment in Zurich using an extractive electrospray ionization time-of-flight mass spectrometer (EESI-TOF-MS) - Part 2: Biomass burning influences in winter, Atmos. Chem. Phys., 19, 8037-8062, https://doi.org/10.5194/acp-19-8037-2019, 2019.

Stefenelli, G., Pospisilova, V., Lopez-Hilfiker, F. D., Daellenbach, K. R., Hüglin, C., Tong, Y., Baltensperger, U., Prévôt, A. S. H., and Slowik, J. G.: Organic aerosol source apportionment in Zurich using an extractive electrospray ionization time-of-flight mass spectrometer (EESI-TOF-MS) – Part 1: Biogenic influences and day–night chemistry in summer, Atmos. Chem. Phys., 19, 14825-14848, https://doi.org/10.5194/acp-19-14825-2019, 2019.

Ulbrich, I. M., Canagaratna, M. R., Zhang, Q., Worsnop, D. R., and Jimenez, J. L.: Interpretation of organic components from Positive Matrix Factorization of aerosol mass spectrometric data, Atmos. Chem. Phys., 9, 2891-2918, https://doi.org/10.5194/acp-9-2891-2009, 2009.

Wang, D. S., Lee, C. P., Krechmer, J. E., Majluf, F., Tong, Y., Canagaratna, M. R., Schmale, J., Prévôt, A. S. H., Baltensperger, U., Dommen, J., El Haddad, I., Slowik, J. G., and Bell, D. M.: Constraining the response factors of an extractive electrospray ionization mass spectrometer for near-molecular aerosol speciation, Atmos. Meas. Tech. Discuss., 2021, 1-24, https://doi.org/10.5194/amt-2021-125, 2021.

---

## Author Comment (AC2)

**Response to RC3**

We thank the reviewer for the helpful comments. Below we provide a detailed point-by-point response to the issues raised by the reviewer. Reviewer comments provided in *italics* and our responses follow in normal text. Changes to the manuscript are denoted in blue font. When our responses reference other comments, we use the formalism R#C#, such that R1GC5 and R1SC5 would refer to General Comment 5 by Reviewer 1, and Specific Comment 5 by Reviewer 1. When indicating the page and line, we use the formalism P#L#, such that P10L5 would refer to Page 10 Line 5.

**Major Comment #1**

*At the end of section 2.2.2 the apparent sensitivity application informed by the AMS mass is introduced and then said it will be discussed later. Section 2.2.3 basically does the same thing. Can it be briefly mentioned here that two methods of applying EESI-TOF sensitivities were tested in the study and they'll be discussed and evaluated in a different section? In its current state the authors review the methods then say they will discuss them further later.*

**Response:**

We appreciate the reviewer's suggestion and have clarified this in the manuscript as follows.

In Section 2.2.2 (P7 L11, following Eq. 3), we discuss the method based on cPMF outputs:

"Equation (3) is used to determine the apparent factor-specific sensitivities from cPMF outputs by defining the AMS contribution to the factor profile (μg m$^{-3}$) as $Mass_x$ and the EESI-TOF contribution (cps) as $I_x$."

In Section 2.2.3 (P8 L2, following Eq. 4), we present the GBR-based method used for comparison:

"The factor-specific sensitivities derived from cPMF (Eq. 3) and from the GBR model (Eq. 4) are compared in Sect. 3.2."

**Major Comment #2**

*I'm unclear about what residuals are used and how the residuals are used as a reference to help retrieve a balanced solution from the AMS/EESI-TOF joint PMF. Are the residuals the total residual (basically the Q value) of the "best" stand-alone AMS and EESI-TOF PMF runs? What does it mean to use these residuals "as a reference to retrieve a balanced solution"? Instead of "(step 4)" could you replace that with "(procedure described in step 4)"? It's unclear if, when reading this, the reader should skip to step 4.*

**Response:**

We have clarified several issues here. First, the general reference to "residuals" is specifically to scaled residual distributions ($e_{ij}/s_{ij}$), as discussed in detail in Section 2.3.3. Second, we have clarified what is meant by using the residuals "as a reference to retrieve a balanced" solution and the associated reference to step 4. The revised text reads (P8L27):

"1) PMF analyses are conducted on the standalone EESI-TOF and AMS datasets with synchronised time resolution, including constraints on factor profiles as necessary. Residual distributions from the optimised solutions are used later in step 3 as a criterion for assessing relative instrument weight."

**Major Comment #3**

*Can the authors consider changing the title of section 2.3 to "combined PMF method" or something similar? It would describe the section better as this section is a set of instructions for researchers who would want to apply an identical or similar method to their datasets.*

**Response:**

Thanks for this suggestion. We change the title of this subsection to "Combined Positive Matrix Factorisation (cPMF) Method".

**Major Comment #4**

*I'm finding the presentation of section 2.3 hard to follow. There's description of some unique things that were done for the PMF analysis in this paper and there's description of PMF generally that have been published many times. I have some suggestions for clarification.*

1) *Change figure 2 to figure 1.*
2) *Combine the ideas from (page 8, lines 1-9) and (page 9, lines 6-24) with each other and use it to introduce section 2.3. Trim it down considerably. Can say "We used PMF and in one sentence this is what it is and what it does. We combined the mass spectral time series from the EESI-TOF and AMS to create the input matrix for the PMF analysis. We performed PMF on the combined input matrix and we call this "combined PMF" (cPMF). A conceptual schematic is shown in Figure 1." Two additional things to keep that the authors mentioned that I thought was useful was the respective units of the AMS and EESI components and the inclusion of NO+ and NO2+*
3) *Remove (page 10) lines 7-19 and reference either the Paatero papers or Ulbrich's review*
4) *More or less remove (page 9) lines 11-19 and equation 5 for the same reason.*
5) *In a new paragraph make the major points "We present an overview of the cPMF through a series of steps listed below. Details corresponding to each step are outlined in subsequent sections. The overall procedure is outlined in Figure 2, with the main steps as follows: ..."*
6) *Title subsections as 2.3.# where # = 1-6 (representing each step in cPMF) and include the relevant information to perform that step in that section. For example, the first subsection would be Section 2.3.1 with a title of "Step 1: Conventional PMF". In this section you'll say PMF was performed on the AMS and EESI-TOF datasets independently. You'll also explain how constraints on factor profiles were applied. Section 2.3.2 with a title "Step 2: Creating input matrix for cPMF". Maybe consider combining steps 2 and 3.*

*It's clear all the elements of the analysis are detailed in the subsections, but it's not clear at what points analyses are performed. For instance, (page 11) lines 1-21 describe the calculation of the apparent sensitivity for the EESI-TOF, but it's unclear at what step(s) in the cPMF method this is done*

**Response:**

We appreciate the reviewer's suggestions, and have made the following improvements to the manuscript:

1. We condensed the introduction to PMF and cPMF, roughly following the lines suggested by the reviewer in points 1 and 2.

2. We clarified the old Figure 1 (now Figure 2) and linked it more closely to the related text by labelling the corresponding steps in boxes.

3. Sect. 2.3.0 to 2.3.4 are numbered to align with steps of the same number in the new Fig. 2 and related text. In Sect. 2.3.0, the basic principle of PMF, and $a$-value approach is introduced, we also mentioned its capability in cPMF analysis. Sect 2.3.1 demonstrates how the combined dataset is constructed. In Sect. 2.3.2, we introduced the general method to construct the reference profile for cPMF analysis. It Sect. 2.3.3, the instrument weighting method and corresponding evaluation process are introduced. Finally, in Sect. 2.3.4, we introduced the case-specific criteria for solution selection and bootstrap analysis.

4. We decided to keep Eq. (5) and Eq. (6) in the manuscript, because Eq. (5) is the basic principle of PMF and Eq. (6) introduces the concept of scaled residual scaled residual ($e_{ij}/s_{ij}$). Although well known, both define symbols and quantities that are critical for later discussion. Equation (7) has been deleted as suggested.

**Major Comment #5**

*I apologize, I might have missed some information; why is 1 ug m-3 used as a reference value?*

**Response:**

This is used for convenience in representing the factor profiles. It is similar to the traditional presentation of AMS-only PMF data (dimensionless factor profiles normalized such that the sum is 1), while providing an easy means of visually separating profiles of factors to which the EESI-TOF is not sensitive vs. those to which it is. It also simplifies some of the governing equations used to separate EESI-TOF and AMS components of the cPMF solutions.

**Major Comment #6**

*In subsections 2.3.3 and 2.3.4 the authors do a very nice job of explaining how they treated and evaluated the data and solutions for the cPMF. I think I personally would have to actually go through the process to fully understand all the details.*

**Response:**

We appreciate the reviewer's support and agree this is a complex analysis. We have tried to make the discussion easier to follow by moving dataset-specific details to the Supplement, as discussed in response to RC1GC5.

**Major Comment #7**

*Consider putting the text between the title of section 3.2 and 3.2.1 as a supplemental section. I was finally excited to see some results with the title "cPMF results", but instead started reading more details of analysis.*

**Response:**

We have moved this section, which introduces the method from Kiendler-Scharr et al. (2016) to estimate the contribution from organonitrate and inorganic nitrate to individual factors, to the SI as Text S3.

**Major Comment #8**

*The authors have demonstrated the application of cPMF to a multi-season, complex mass spectrometry dataset from two instruments. Despite the thorough and rigorous development and evaluation of the method some curiosities and uncertainties still persisted like the contribution of high sensitivity species in contributing to a factor profile, multi-modality of sensitivity values in the COA and CSOA factors, and increasing uncertainty when sub-dividing factors like in the case of the aggregate BB factor. These uncertainties contribute in the overall uncertainty to a relatively complex data processing procedure. Can the authors briefly provide any laboratory experiments, calibrations, or "ideal" datasets in the conclusions section where this cPMF method could be applied in the future as test cases for improving the interpretability and quality of the cPMF analysis?*

**Response:**

We certainly agree with the reviewer on the need for continued evaluation of cPMF results and performance. In our view, the most useful strategy at present is simply the application of cPMF to additional datasets. We are currently in the process of doing this for both laboratory measurements and ambient datasets from different geographical areas, although consider it premature to comment on this in the current manuscript.

Another useful approach would be the replacement of the EESI-TOF with another instrument having high chemical resolution but semi-quantitative outputs, such as the FIGAERO-CIMS. This is noted in the Conclusions.

**Minor Comment #1**

*P3 L3: "The corresponding decrease in chemical resolution, particularly for  multifunctional and/or highly oxygenated SOA components ..."*

Reviewer: Please remove the words indicated above.

**Response:**

These words are removed and the revised text reads (P3L4):

The corresponding decrease in chemical resolution, particularly for multifunctional and/or highly oxygenated SOA components (e.g., multifunctional acids, peroxides, organonitrates, organosulfates, oligomers), limits the resolution of SOA source apportionment.

**Minor Comment #2**

*P3 L47: "...gas-phase concentrations measured by a Vocus proton transfer reaction-mass spectrometer (Vocus-PTR-MS) (Wang et al., 2021)"*

Reviewer: I didn't see the Wang et al. study listed in the references. Please list the following article in the references section **"Constraining the response factors of an extractive electrospray ionization mass spectrometer for near-molecular aerosol speciation" Wang, et al. (2021)**

**Response:**
The reference list has been updated to include:
Wang, D. S., Lee, C. P., Krechmer, J. E., Majluf, F., Tong, Y., Canagaratna, M. R., Schmale, J., Prévôt, A. S. H., Baltensperger, U., Dommen, J., El Haddad, I., Slowik, J. G., and Bell, D. M.: Constraining the response factors of an extractive electrospray ionization mass spectrometer for near-molecular aerosol speciation, Atmos. Meas. Tech., 14, 6955-6972, https://doi.org/10.5194/amt-14-6955-2021, 2021.

**Minor Comment #3**

*P4 L22: "The present study is the first application of cPMF to a joint EESI-TOF/AMS dataset, and the  quantitative EESI-TOF-driven source apportionment."*

Reviewer: Please remove words indicated above.

**Response:**
Now these words are deleted, and the sentence is corrected in P4 L25 as:
The present study is the first application of cPMF to a joint EESI-TOF/AMS dataset, and the first quantitative EESI-TOF-driven source apportionment.

**Minor Comment #4**

*P5 L20: "...from highresolution mass spectral analysis..."*

Reviewer: Replace "highresolution" with "high resolution" or "high-resolution".

**Response:**
We now use "high-resolution" throughout the manuscript.

**Minor Comment #5**

*P16 L2: Is "(2.54)" a ratio value? If so can you change it to read (NO+/NO2+ = 2.54)? It's unclear what (2.54) means as is.*

**Response:**
Yes, it is the ratio of $NO^+/NO_2^+$. Now we have changed this here and throughout out the manuscript wherever appropriate.

**Minor Comment #6**

*P30 L2: Replace "This factor has a qualitatively a profile similar to the summer campaign..." with "This factor qualitatively has a profile similar to the summer campaign...".*

**Response:**

This text was modified in response to RC1SC17, where the use of "qualitatively" was found unclear. The revised text (P21 L1) reads:

"This factor is dominated by the $C_nH_{2n+1}^+$, and $C_nH_{2n-1}^+$ series, consistent with *n*-alkanes and branched alkanes, with lower $CO^+$ and $CO_2^+$ content than the $HOA_{S,C}$. The $HOA_{W,C}$ time series correlates strongly with $HOA_{W,A}$ ($r^2$ of 0.913)."

**Minor Comment #7**

*P33 Paragraph2: I think using the COA profile as a reference is an appropriate solution to a tough problem.*

**Response:**

We appreciate the reviewer's support. In the future, we intend to use the sensitivity of a reference compound (levoglucosan) from on-site calibrations to facilitate cross-campaign comparisons.

**Minor Comment #8**

*Figure 8: Figure 8 has a line through the x-axis label. I assume this is a formatting/review feature accidentally carried into the PDF from Word?*

**Response:**

We have corrected this formatting error. The figure appears below.

[Figure]

Figure 6. Relative apparent sensitivity $AS_k/\overline{AS_{COA_{W,C}}}$ as a function of levoglucosan fraction for all factors resolved from the cPMF of the winter dataset except $CSOA_{W,C}$. Error bars denote standard deviation.

**Minor Comment #9**

*Figure 13: Can you set the maximum value for the y-axis in panel a = 0.5? In the caption can you note what defines the box and whiskers (i.e. are the boxes 25th and 75th percentiles?)*

**Response:**

Now the maximum value for y-axis for summer is set to 0.5. We have clarified the caption and also expanded the explanation by adding the 25th and 75th percentiles. The revised text reads:

"The box-and-whisker diagram shows the mean (open/filled circle), median (horizontal bar), interquartile range (rectangle, the 25th percentile is the lower edge and the 75th is the upper edge), and minimum/maximum values (whiskers)."

[Figure]

[Figure]

Figure 13. Box-and-whisker diagrams of factor contribution to total OA with/without applying the factor dependent sensitivities, for summer in a) and winter in b) within accepted solutions. For each pair of factors, the contribution without factor-dependent sensitivity applied is shown in the left box (open symbols), whereas the contribution corrected by factor-dependent sensitivity is shown in the right box (full symbols). The box-and-whisker diagram shows the mean (open/filled circle), median (horizontal bar), interquartile range (rectangle, the 25[th] percentile is the lower edge and the 75[th] is the upper edge), and minimum/maximum values (whiskers). Note that here the contribution of InorgNit factor and contribution of $NO^+$ and $NO_2^+$ from inorganic nitrate in each factor are excluded.

**Minor Comment #10**

*P43 L1: "The cPMF method presented herein is can be utilised as-is not only for the AMS/EESI- TOF combination..."*

*Reviewer: Please delete the misplaced "is" in the sentence above.*

**Response:**

The typo has been corrected.

**Reference**

Kiendler-Scharr, A., Mensah, A. A., Friese, E., Topping, D., Nemitz, E., Prevot, A. S. H., Aijala, M., Allan, J., Canonaco, F., Canagaratna, M., Carbone, S., Crippa, M., Dall Osto, M., Day, D. A., De Carlo, P., Di Marco, C. F., Elbern, H., Eriksson, A., Freney, E., Hao, L., Herrmann, H., Hildebrandt, L., Hillamo, R., Jimenez, J. L., Laaksonen, A., McFiggans, G., Mohr, C., O'Dowd, C., Otjes, R., Ovadnevaite, J., Pandis, S. N., Poulain, L., Schlag, P., Sellegri, K., Swietlicki, E., Tiitta, P., Vermeulen, A., Wahner, A., Worsnop, D., and Wu, H. C.: Ubiquity of organic nitrates from nighttime chemistry in the European submicron aerosol, Geophys. Res. Lett., 43, 7735-7744, https://doi.org/10.1002/2016gl069239, 2016.